# Scalable Evaluation and Neural Models for Compositional Generalization

**Giacomo Camposampiero**
IBM Research – Zurich, ETH Zurich
giacomo.camposampiero1@ibm.com

**Pietro Barbiero**
IBM Research – Zurich
pietro.barbiero@ibm.com

**Michael Hersche**
IBM Research – Zurich
michael.hersche@ibm.com

**Roger Wattenhofer**
ETH Zurich
wattenhofer@ethz.ch

**Abbas Rahimi**
IBM Research – Zurich
abr@zurich.ibm.com

## Abstract

Compositional generalization—a key open challenge in modern machine learning—requires models to predict unknown combinations of known concepts. However, assessing compositional generalization remains a fundamental challenge due to the lack of standardized evaluation protocols and the limitations of current benchmarks, which often favor efficiency over rigor. At the same time, general-purpose vision architectures lack the necessary inductive biases, and existing approaches to endow them compromise scalability. As a remedy, this paper introduces: 1) a rigorous evaluation framework that unifies and extends previous approaches while reducing computational requirements from combinatorial to constant; 2) an extensive and modern evaluation on the status of compositional generalization in supervised vision backbones, training more than 5000 models; 3) Attribute Invariant Networks, a class of models establishing a new Pareto frontier in compositional generalization, achieving a 23.43% accuracy improvement over baselines while reducing parameter overhead from 600% to 16% compared to fully disentangled counterparts. Our code is available at github.com/IBM/scalable-compositional-generalization.

## 1 Introduction

Robustness to distributional shifts is a long-standing research topic in machine learning, spanning areas such as domain adaptation [1, 2], transfer learning [3], and out-of-distribution (OOD) generalization [4]. In this work, we focus on a specific type of OOD generalization, that is, compositional generalization. Informally, this type of generalization requires the model to generalize to *unknown* combinations of *known* concepts. For instance, given a dataset of pairs (input, {concepts labels}), a model trained on the samples {( 🍎, {apple, yellow}), ( 🍌, {banana, green})} should be able to correctly predict {apple, green} for the test input image 🍏. Achieving and even assessing this type of generalization remains, however, challenging. To date, a standardized evaluation methodology for assessing compositional generalization has not been established. This renders the evaluation of existing results hard and prevents a consistent quantification of progress in the field. Furthermore, current state-of-the-art (SOTA) evaluation frameworks often trade off efficiency with shallowness, usually testing compositional generalization under weaker constraints to avoid a combinatorial complexity explosion in the number of target attributes [5, 6]. On the other hand, general-purpose SOTA vision architectures typically lack inductive biases for compositionality and disentanglement, resulting in limited generalization in both supervised and unsupervised settings [5, 7, 8]. Existing methods for incorporating such biases often rely on factorizing the prediction task across multiple specialized models [9], which significantly reduces scalability and limits practicality even in simple domains.

39th Conference on Neural Information Processing Systems (NeurIPS 2025).

To tackle these issues, this work makes the following main contributions:

1. In Section 3, we present a universal evaluation framework for compositional generalization in supervised learning. This evaluation strategy unifies prior strategies, improves efficiency by reducing complexity from combinatorial to constant, and introduces a controllable degree of freedom in the evaluation that significantly affects model performance and enables a principled hierarchy of evaluation difficulty.

2. We extensively validate the proposed benchmarking method against alternative methodologies and previous works. As part of the validation, we train more than 5000 SOTA vision models, representing the most extensive and up-to-date evaluation on compositional generalization for supervised models.

3. Motivated by the results in Section 3, we introduce in Section 4 a new class of neural architectures, Attribute Invariant Networks (AINs), that favor compositional generalization by construction. We show that AINs achieve a new Pareto frontier in the scalability-generalization tradeoff, significantly improving compositional generalization of standard architectures with only a minimal parameter overhead.

## 2   Background

**Formalizing compositional generalization.** As Ren et al. [10], we focus on the setting presented in Figure 1, considering an arbitrary set of $N$ observations $\mathbf{X} \in \mathbb{R}^{N \times D}$ and their corresponding downstream task labels $\mathbf{Y} \in \mathbb{R}^{N \times E}$. We assume that the observations are generated from a finite set of discrete factors $\mathbf{F}$, in turn divided into $I$ task-relevant factors $\mathbf{G} \subseteq \mathbf{F}$ (for which there exists a causal relationship with the downstream task) and $J$ task-irrelevant factors $\mathbf{O} = \mathbf{F} \setminus \mathbf{G}$. Each observation $\mathbf{x} \in \mathbf{X}$ is produced by a deterministic function $\mathcal{G}_x : (\mathbf{F}, \epsilon_x) \to \mathbb{R}^D$, mapping $\mathbf{F}$ and $\epsilon_x$ ($\perp$ noise accounting for minor input variations, e.g., pixel-wise variations)

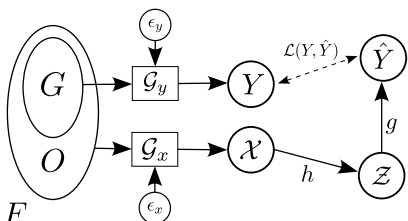

Figure 1: Theoretical setup.

into the data manifold. Labels $\mathbf{y}$ are generated by a deterministic function $\mathcal{G}_y : (\mathbf{G}, \epsilon_y) \to \mathbb{R}^E$, encoding the semantics of the task and parametrized by $\mathbf{G}$ and $\epsilon_y$ ($\perp$ noise accounting for label issues).

**Definition 2.1** (Compositional generalization). Compositional generalization emerges when the supports of the training and testing distributions over compositions of factors, $P_{train}(\mathbf{G})$ and $P_{test}(\mathbf{G})$, are disjoint. More formally, $\text{supp}[P_{train}(\mathbf{G})] \cap \text{supp}[P_{test}(\mathbf{G})] = \emptyset$. The production rules $\mathcal{G}_x$ and $\mathcal{G}_y$ are assumed to be consistent across training and testing.

**Example 2.2.** Consider the Shapes3D sample contained in the yellow corner of Figure 2a. The generative factors of the dataset are $\mathbf{F} = $ (size, shape, object hue, floor hue, wall hue). Here, $\mathbf{x}$ is the $64 \times 64$ RGB image, while $\mathbf{y} = $ (small, cube, blue object). Hence, $\mathbf{G} = $ (size, shape, object hue) and $\mathbf{O} = \mathbf{F} \setminus \mathbf{G} = $ (floor hue, wall hue). $\mathbf{x}$ represents a compositional generalization test example iff $P_{train}$(small, cube, blue object) $= 0$.

**Representation learning.** We consider the classical representation learning framework (Fig. 1). A backbone model $h : \mathbb{R}^D \to \mathbb{R}^S$ is used to extract a compressed representation of the input ($\mathbf{z}$), and a classification head $g : \mathbb{R}^S \to \mathbb{R}^E$ is used on top of the extracted representations to solve the downstream task. The output is predicted by the composition of these two functions, $\hat{y} = (g \circ h)(\mathbf{x}) = f(\mathbf{x})$. This model is trained by minimizing the empirical risk $\hat{R} = 1/N \sum_{i=1}^{N} \mathcal{L}(\mathbf{Y}_i, f(\mathbf{X}_i))$.

## 3   A novel framework for evaluating compositional generalization

We propose a general and scalable framework to evaluate compositional generalization and disentanglement in fully supervised settings (where the generative factors $\mathbf{G}$ are known and labeled) with variable degrees of difficulty (dictated by the similarity between train and test observations). Additionally, we define a dataset-agnostic, efficient procedure (dubbed orthotopic evaluation) for decomposing $\mathbf{G}$ in two splits ($train$ and $test$), such that $P_{train}$ and $P_{test}$ satisfy Definition 2.1.

## 3.1 Method

**An extended definition of compositional generalization.** For any $\mathbf{y}_1 \in P_{train}, \mathbf{y}_2 \in P_{test}$, Definition 2.1 constrains the number of task-relevant factors shared between the two observations, $\kappa = \max_{\mathbf{y}_1,\mathbf{y}_2} \sum_{i=1}^{I} \mathbb{I}_{\mathbf{y}_{1,i}=\mathbf{y}_{2,i}}$, to be $\kappa \leq I - 1$. However, the definition does not define any lower bound for $\kappa$, which can take any value in $[0, I-1]$. This ambiguity makes it unsuitable for benchmarking purposes, as it could potentially result in inconsistent evaluation protocols and hinder direct comparability across works. Previous works, in fact, already fell victim to this: some works [7, 8] considered the scenario where $\kappa = 1$, whereas others [5, 6] focused on $\kappa = I - 1$. To solve this, we propose a more complete definition of compositional generalization.

**Definition 3.1** (Compositional Generalization (complete)). Consider two data distributions, $P_{train}$ and $P_{test}$. For any $\mathbf{y}_2 \in P_{test}$ and its closest sample in the training data $\mathbf{y}_1 \in P_{train}$, compositional generalization emerges when $c \leq \kappa \leq I - 1$. In particular, $c \in [0, I]$, dubbed here as *compositional similarity index*, is a hyper-parameter of the evaluation setup.

Intuitively, $c$ influences the degree of similarity between known and unknown compositions of concepts and, indirectly, the volume of the concept space excluded from the training data. Consider, for instance, Figure 2a. When $c = I - 1 = 2$, a corner of the hypercube (yellow orthotope) is excluded from the training data and used for testing; all the samples in this volume share at most two task-relevant factors with the closest samples observed during training. If $c = 1$, on the other hand, both the yellow and green orthotopes are excluded from training and used for testing; hence, some test samples share two attributes with the closest training examples (green orthotopes), and some share only one (yellow corner). The characterization of $c$ is, to some extent, equivalent to the concept distance metric introduced by Okawa et al. [11]. Other than $c$, we also identify the existence of additional degrees of freedom in the evaluation, such as the size of the excluded volumes or their position in the space (see Appendix A.1), but we consider them fixed in the context of this work.

**A new ladder for compositional generalization difficulty.** We hypothesize that the newly introduced compositional similarity index directly influences the difficulty of the evaluation task. Intuitively, larger $c$ values increase the similarity between known and unknown combinations of concepts, rendering test samples "almost in-distribution". In addition, larger $c$ values increase the overall combinations of concepts accessible during training, in turn enhancing data variety (known to be vital for training robust and generalizing neural networks [12, 13]). Based on these observations, we postulate the existence of a ladder of compositional evaluation difficulty, presented in Figure 2b. In particular, we outline a discrete spectrum of evaluation complexity levels corresponding to

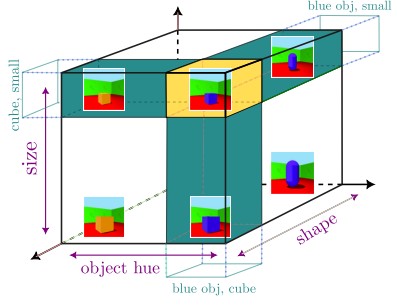

(a) Orthotopic evaluation intuition.

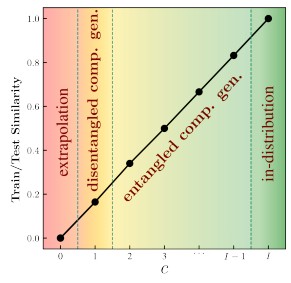

(b) Ladder of compositional evaluation difficulty.

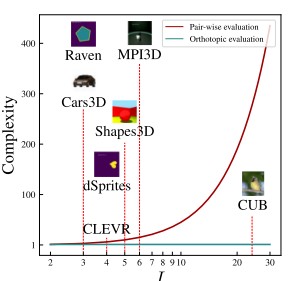

(c) Orthotopic evaluation efficiency.

Figure 2: **Orthotopic evaluation for compositional generalization.** Intuitively, orthotopic OOD split generation works by iteratively projecting and pruning the data in every $c$-dimensional attribute's subspace. We exemplify this in **(a)** for the dataset Shapes3D, where we consider only $I = 3$ attributes for simplicity. We highlight the disentangled compositional split ($c = 1$, green+yellow) and the entangled compositional split ($c = 2$, yellow). **(b)** pictures the proposed ladder of compositional evaluation difficulty, showing the dependence between the $c$ parameter and the similarity between train and test generative factors. This delineates a ladder of different difficulties of compositional evaluation, spanning from extrapolation to in-distribution regimes. Finally, **(c)** shows the computational advantage of the proposed evaluation technique compared to the naïve pair-wise evaluation strategy.

different $c$ values, spanning from extrapolation (generalization to unseen attribute values, $c = 0$) to compositional generalization ($1 \leq c \leq I - 1$) and, eventually, in-distribution generalization ($c = I$).

We further distinguish between two different types of generalization within the compositional case: *disentangled compositional generalization* ($c = 1$) and *entangled compositional generalization* ($2 \leq c \leq I - 1$). The difference between the two is subtle. In entangled compositional generalization, models can still generalize despite entangling the information of a subset of $n$ task-relevant generative factors in their latent space, as long as the evaluation is performed for $c \geq n$. Consider the illustrative example in Figure 2a. If the model learns entangled representations of the {shape, size} concepts, it may still succeed on a test instance with $c = 2$ (i.e., the sample in the yellow volume is used for testing) because the attributes entangled by the model are observed during training (e.g., in the small orange cube, top-left). However, when $c < n$, the specific combination of attributes is, by construction, absent from the training set, rendering generalization impossible for models relying on entangled representations. Disentangled compositional generalization ($c = 1$), on the other hand, strictly operates in the latter condition. This allows us to strictly evaluate both compositionality (that is, the ability to combine known concepts) and disentanglement (that is, the ability to separate distinct factors of variation independently). For more details, see Appendix A.2.

**Scalable compositional evaluation procedure.**   How can we translate the principles previously introduced into an algorithm that generates compositional evaluation splits? A naïve procedure, already explored to some extent in previous works [7, 8] and referred to here as *pair-wise evaluation*, allows for evaluating compositional generalization by considering only $c$ factors at a time. While effective, this protocol is not scalable: it requires training and evaluating a different model for each $s \in \mathcal{S}$, where $\mathcal{S}$ is the set of all $c$-dimensional subspaces of the original $I$-dimensional task-relevant concept manifold. That is, the evaluation complexity is $\Theta(I^c)$. In this work, we propose an alternative procedure, dubbed *orthotopic evaluation*, whose complexity is $\Theta(1)$. The pseudo-code is included in Algorithm 1 and encompasses different stages. Firstly, the set of possible combinations of $c$ factors in $\mathbf{G}$ (representing all the possible $c$-dimensional subspaces of the attributes' hyperspace) is computed (Line 2). Then, the dataset is iteratively projected and pruned in each one of these $c$-dimensional subspaces (Lines 3-5), as intuitively visualized in Figure 2a. The pruning operation (Line 5) is based on thresholds dynamically computed based on the percentage of the excluded orthotope's support along each dimension, summarized in the pseudo-code by the primitive `exclusion` (see Appendix A.3 for an example). Compared to pair-wise, orthotopic evaluation allows for flexibly setting the operating value of $c$ (in the range $[1, I - 1]$) while requiring only a single training run per model. Figure 2c shows an efficiency comparison between the two evaluation strategies, comparing the number of training runs required for different numbers of task-relevant generative factors.

---

**Algorithm 1** Orthotopic split generation.

---

1: **procedure** SPLITDATASET($\mathbf{X}$, $\mathbf{G}$, $c$)
2:     $\mathcal{S} = \binom{\mathbf{G}}{c+1} = \{\mathbf{S} \subseteq \mathbf{G} : |\mathbf{S}| = c + 1\}$          ▷ Get all combinations of $c$ factors in $\mathbf{G}$.
3:     **for each** $s \in \mathcal{S}$ **do**                              ▷ Iterative orthotope pruning.
4:         $\mathbf{X}_{proj} = \mathbf{proj}_s \mathbf{X}_{train}$                      ▷ Project $\mathbf{X}$ onto the $c$-dimensional subspace $s$.
5:         $\mathbf{X}_{train} = \mathbf{X}_{train} \setminus (\mathbf{X}_{proj} \cap \mathrm{exclusion}(s))$      ▷ Pruning operation in the subspace $s$.
6:     **return** $\mathbf{X}_{train}, \mathbf{X}_{train}^{\complement}$

---

### 3.2   Experimental setup

**Datasets and OOD splits.**   We consider different vision datasets that encompass a broad spectrum of complexity and visual diversity, ranging from purely synthetic to real images. Importantly, we only use datasets for which the annotations of the generative factors $\mathbf{F}$ are available. In particular, we experiment with dSprites [14], I-RAVEN [15], Shapes3D [16], CLEVR [17], Cars3D [18], and MPI3D [19]. More details on the datasets can be found in Appendix B.1. To create the compositional OOD splits, we fix all the degrees of freedom of the proposed orthotopic framework and only study compositional generalization for different values of the $c$ parameter. In particular, we define attribute-wise thresholds for the values of all task-relevant generative factors in each dataset such that the percentage of excluded attribute-pairs is constant ($\sim 60\%$). This is tightly mirrored in the size of the training and testing splits, as the data samples are equally distributed among different combinations.

**Models.** Inspired by Schott et al. [5], we experiment with a wide range of models that encompass different types of architectural and data inductive biases previously proposed to facilitate generalization. Besides MLPs [5], we evaluate convolutional neural networks (CNNs) such as Residual Networks (ResNets) [20], Densely Connected Convolutional Networks (DenseNets) [21], Wide Residual Networks [22], and the most recent iteration of CNN architectures, ConvNeXt [23]. We also consider vision transformers (that incorporate the attention mechanism, enabling global dependencies modeling and contextual understanding), such as the original Vision Transformer (ViT) [24] and the most recent Swin Transformer [25]. Since SOTA approaches in deep learning often rely on pre-training on large corpora prior to fine-tuning on specific target tasks, we also evaluate this different form of (data) bias. To this end, we fine-tune pre-trained versions of some of the previously mentioned models: PyTorch's RN-101, RN-152, and DN-121 pre-trained on ImageNet-1k. More details on the models used in our experiments can be found in Appendix B.2. On an orthogonal direction compared to the previous *monolithic* models, we additionally investigate *explicitly disentangled* (ED) architectures. The core idea in these models is to factorize the forward and backward passes by instantiating independent sub-networks for each factor in **G**. This corresponds to the strongest inductive bias towards disentanglement that can be implemented in fully-supervised models from an architectural perspective. We introduce an improved version of the Shared Architecture [9], which attains on-par or superior performance on the evaluated dataset compared to the original model (as shown in Appendix B.3). Specifically, our implementation constructs concept representations endowed with a similarity-preserving structure, induced via a kernel-based initialization scheme [26].

**Evaluation.** We use the exact-match score (multi-label accuracy) as the main evaluation metric. All the models are trained using a standard cross-entropy loss on all attribute labels. The model selection is based on a $10\%$ held-out split of the training data (testing in-distribution generalization). We also investigated alternative selection metrics specifically designed to improve the performance on compositional generalization splits; however, they did not perform better compared to the standard in-distribution validation, as shown in Appendix B.4. All experiments were conducted on compute nodes with an AMD EPYC 7763 64-Core CPU, 2TB RAM, and an NVIDIA A100-SXM4 GPU (80GB), running on Red Hat Enterprise Linux 9.4, CUDA 12.4, and PyTorch 2.3.0+cu121.

### 3.3 Results and discussion

In this section, we aim to address the following research questions on the orthotopic evaluation framework introduced in Section 3.

- **(R1)** To what extent does the similarity between training and testing observations, characterized by the $c$ parameter, influence compositional generalization? Furthermore, do the empirical results align with the postulated ladder of compositional evaluation difficulty?

- **(R2)** How does orthotopic evaluation compare to less efficient methodologies (e.g., pair-wise attribute evaluation) and previous works more generally?

$c$ **significantly influences the evaluation of compositional generalization (R1).** Figure 3a reports the results of the evaluation of the full range of $c$ values ($0 \leq c \leq I$) on the different datasets and models considered in this work. A general pattern appears from the results: $c$ distinctly impacts the measured compositional generalization across all the studied model families and datasets. Notably, this behavior is entirely attributable to the similarity between training and testing concepts, as the training data size (number of observations) and variety (number of concept combinations) are fixed. This observation entails several important implications for the empirical research on compositional generalization. First, the specific value of $c$ used in prior studies must be retrospectively taken into account to properly contextualize and interpret their findings. Second, in light of its significant impact on the measured generalization performance, $c$ should be systematically considered as part of future empirical investigations of compositional generalization.

**Empirical results support the proposed ladder of evaluation difficulty (R1).** Figure 3a also provides an empirical validation of the ladder of compositional evaluation difficulty proposed in Section 3.1. Starting from the two extremes of the considered intervals, we observe that on all datasets not a single model could properly generalize in the extrapolation domain ($c = 0$), while (almost) every model achieved perfect accuracy in-distribution ($c = I$). In between, compositional generalization and $c$ are in the majority of cases positively correlated. This suggests that neural networks can

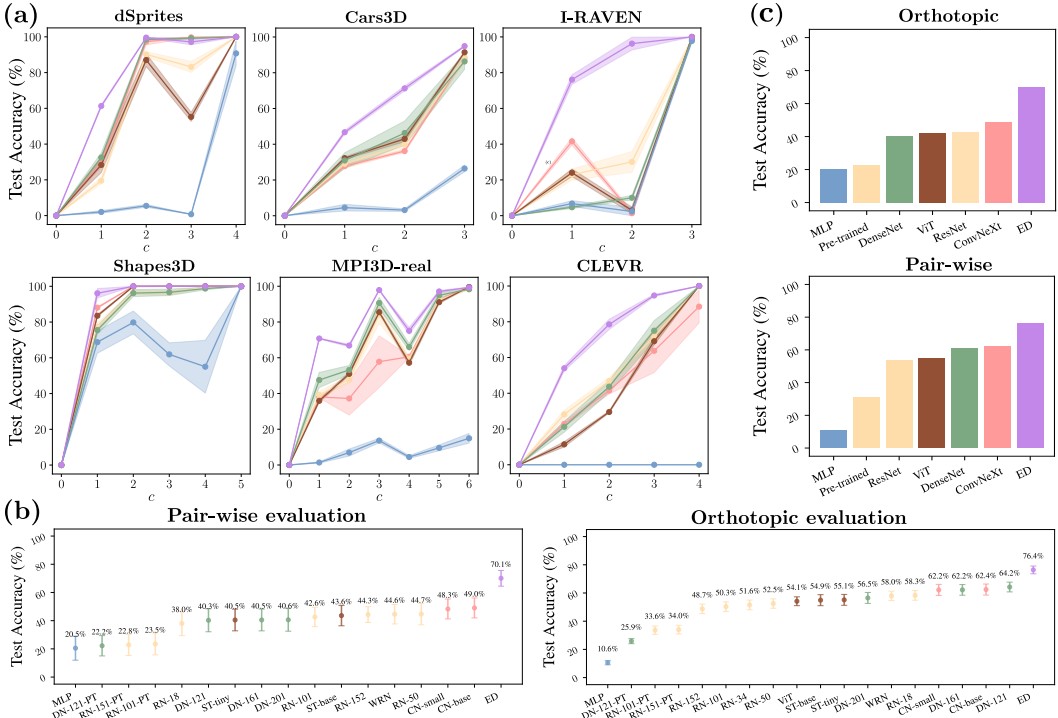

Figure 3: **Orthotopic evaluation results.** In **(a)**, we report the test accuracy (on the compositional generalization split) for different values of the compositional similarity index $c$. The results are collected for six well-known representation learning datasets and grouped into the six major families of models considered in this work. The uncertainty (SEM) is reported over different models in the family (various model sizes, pre-training, etc.) and random seeds (3). **(b)** on the other hand compares the results obtained with orthotopic evaluation (represented now at the granularity of single models, averaged across different datasets and seeds, for $c = 1$) with the results obtained with a more precise but inefficient evaluation technique, pair-wise evaluation. In order to extract a more general picture, we group the same results by model family in **(c)**. The results reported in this figure are obtained from a large-scale analysis encompassing more than 5000 training runs.

leverage the increasing level of similarity between training and testing observations. However, as the pressure towards learning disentangled representations increases (smaller $c$), the performances of these models drop sensibly. For some datasets (e.g., dSprites and Shapes3D), it is possible to observe a clear distinction between the disentangled ($c = 1$) and entangled compositional generalization ($1 < c < I$) regimes; the majority of the models, in fact, perfectly generalize on the latter ($\sim 100\%$ test accuracy), but struggle on the former. However, this difference is not as pronounced in other datasets, a sign that sometimes models could even struggle recombining information of entangled attributes. Overall, the consistent correlation between test accuracy and compositional complexity provides empirical support for the proposed ladder of compositional evaluation difficulty. We also provide some evidence that, realistically, the same observation would hold in more noisy, real-world datasets in Appendix E.

**Orthotopic and pair-wise evaluation share the same overall trends (R2).** Figure 3b reports a comparison between the orthotopic evaluation and pair-wise evaluation frameworks. Pair-wise represents the naïve baseline where only a pair of attributes is considered in the creation of the test split. From the results, we can observe that the test accuracy in orthotopic evaluation is consistently lower (11.6% on average) than that measured in the pair-wise evaluation (while the in-domain test accuracy is comparable, as shown in Supplementary Figure 25). This is, to some extent, expected since in the orthotopic framework we exclude more observations from the training data (40% vs. 10%) and the number of different task-relevant factors can be much larger ($I - 1$ vs. 2). Except for this, we can observe in Figure 3c that the main trends are consistent across the two frameworks. Models with no architectural inductive bias are the worst (MLP), pre-trained models generally perform worse than

models trained from scratch, some architectures are consistently better than others (e.g., ConvNeXt), and ED architectures are significantly better than any other monolithic model. We observe similar results when the number of optimization steps is substantially increased (up to $240\times$) to investigate the emergence of grokking phenomena [27, 28] (see Appendix C.5). Modifying the models by replacing the original activation functions with learnable ones, intended to promote mappings closer to isomorphic bijections, also does not improve the outcomes (see Appendix C.4). The full results for the two frameworks are included in Appendices E.1 and C.2, respectively.

**Orthotopic evaluation results are mostly consistent with the results from previous works (R2).** Comparing the overall results presented in Figure 3 with the closest previous work from Schott et al. [5], we observe similar results on dSprites and Shapes3D. On the other hand, we measure significantly better performances on MPI3D-real. Pointing out the exact origin of this discrepancy is hard, since this work differs from theirs in many aspects: different problem formulation (classification vs. regression), data pre-processing, and investigated models. The remaining datasets cannot be directly compared to any previous work, since this is the first work that uses them in the context of compositional generalization. As Montero et al. [8], we also observe that it is consistently harder for most of the models to generalize well for specific attribute combinations (e.g., shape and size) when the results are broken down to this granularity (as shown in Appendix C.3).

## 4 Improving architectural scalability with Attribute Invariant Networks

The results presented in Section 3 provide a modern and comprehensive perspective on disentanglement and compositionality in vision architectures. In this section, we leverage some of the insights gained from this analysis to devise a new architectural paradigm (dubbed *Attribute Invariant Networks*) which achieves a new Pareto-optimality in the scalability-performance tradeoff for compositional generalization tasks. In particular, we observe that explicitly disentangled (ED) architectures consistently outperform monolithic models, achieving an average improvement of 21.38% over the strongest baseline (ConvNeXt) and making it the *de facto* strongest baseline by a margin for what concerns compositional generalization. However, ED architectures suffer from a significant limitation: requiring a separate set of weights for each additional attribute, they are hardly scalable to any real-world scenario. Hence, we aim to answer the following question: Is it possible to achieve levels of compositional generalization comparable to ED without incurring its significant parameter overhead (i.e., retaining a model size which is closer to monolithic architectures)?

### 4.1 Method

The success of deep learning in vision tasks was initially driven by the insight that shift-invariance is a fundamental symmetry in the image domain. Following a similar principle, we argue that achieving compositional generalization requires identifying the key symmetries involved and design principles that guarantee these invariances by construction.

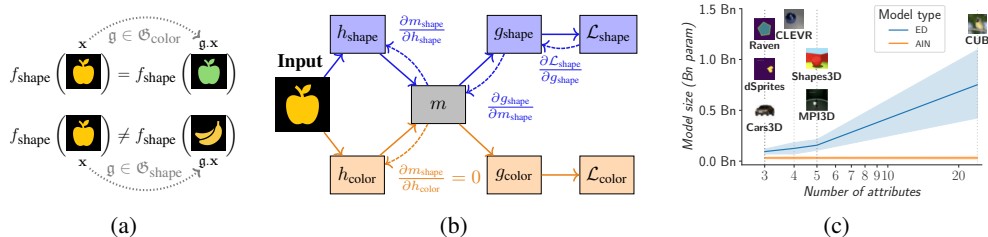

Figure 4: (a) The function $f_{\text{shape}}$ is *attribute invariant* as the prediction is only affected by shape transformations $\mathfrak{g} \in \mathfrak{G}_{\text{shape}}$, while it remains unaffected under any other attribute (e.g., color) transformation $\mathfrak{g} \in \mathfrak{G}_j$ such that $j \neq$ "shape". (b) In Attribute Invariant Networks, only the shape encoder can receive a nonzero gradient for the shape attribute; all other attribute encoders receive a zero gradient. (c) Attribute Invariant Networks' size is nearly constant for different numbers of attributes, while ED models do not scale. Model size provided as mean and $95\%$ confidence interval over different residual networks (i.e., `ResNet{18,34,50,101,152}`).

**Attribute invariance.** Ideally, to favor compositional generalization, the prediction of an attribute should be invariant to group actions associated with other attributes [12]. In other words, the output of an attribute should be invariant under transformations that affect any other attribute. For instance, the prediction of the attribute shape should remain unaffected when the color of the object is altered. We refer to this property as *attribute invariance* (see Figure 4a).

**Definition 4.1** (Attribute invariance). Let $\mathbf{x}$ be a sample, and let $f_i(\mathbf{x})$ be the logit corresponding to attribute $i$. For each attribute $j$, let $\mathfrak{G}_j$ be a group of transformations acting on the input space. Then, $f_i$ is said to be *attribute invariant* if for every group action $\mathfrak{g} \in \mathfrak{G}_j$ with $j \neq i$, $f_i(\mathfrak{g}.\mathbf{x}) = f_i(\mathbf{x})$.

**Blueprint of Attribute Invariant Networks.** In order to guarantee attribute invariance, we propose *Attribute-Invariant Nets (AINs)*, a class of neural architectures designed to support efficient compositional generalization by construction (see Figure 4b). In general, for each attribute $i \in \{1, \ldots, P\}$ an AIN is a composition of three functions $f_i = g_i \circ m \circ h_i$, where

- $h_i(\theta_i) : \mathbb{R}^D \to \mathbb{R}^M$ is an *encoder* that extracts an attribute-specific representation $\mathbf{q}_i$.
- $m(\psi) : \mathbb{R}^M \to \mathbb{R}^S$ is a shared *meta-model* independently transforming any attribute-specific representation into a compressed space, thus generating a set of attribute-specific embeddings $\{\mathbf{z}_i\}_{i=1}^P = \{m(h_i(\mathbf{x}))\}_{i=1}^P$.
- $g_i(\phi_i) : \mathbb{R}^S \to \mathbb{R}$ is a *classification head* mapping compressed representations to the corresponding attribute-specific logit $\hat{y}_i = g_i(\mathbf{z}_i)$.

The structure of AINs allows the optimization of each encoder $h_i$ to be sensitive to group actions related to attribute $i$, while being invariant to group actions of any other attribute, as illustrated in the following theorem (proof in Appendix G).

**Theorem 4.2** (Attribute invariances in gradient updates). *Let $(\mathbf{x}, \mathbf{y})$ be a sample, and let $f_j(\mathbf{x})$ be an AIN's logit corresponding to attribute $j$. Then, for every group action $\mathfrak{g} \in \mathfrak{G}_j$, if $j \neq i$, then $\nabla_{h_i}\mathcal{L}(y_j, f_j(\mathbf{x})) = \nabla_{h_i}\mathcal{L}(y_j, f_j(\mathfrak{g}.\mathbf{x})) = 0$. On the other hand, if $j = i$, then $\nabla_{h_i}\mathcal{L}(y_i, f_i(\mathbf{x})) \neq \nabla_{h_i}\mathcal{L}(y_i, f_i(\mathfrak{g}.\mathbf{x}))$.*

This is not the case for existing architectures where the encoder parameters are sensitive to gradients from all attributes, i.e., $\nabla_h\mathcal{L}(y_j, f_j(\mathbf{x})) \neq 0, \ \forall j \in \{1, \ldots, P\}$. The only exceptions are ED models, which can be seen as a special case of AINs where the meta-model is also attribute-specific. However, this significantly increases the number of parameters of the model. Indeed, notice that 1-layer encoders and classification heads are sufficient for attribute invariances in gradient updates. This means that in practice $|\theta_i|, |\phi| \ll |\psi|$, thus making their contributions negligible w.r.t. the size of the model $m$. As a result, ED parameters complexity can be reduced from linear in the number of attributes $\Theta(P \times \psi)$ to almost constant $\Theta(\psi)$ (see Figure 4c).

## 4.2 Results and discussion

In this section, we aim to address the following research question: **(R3)** How do AINs balance scalability and generalization compared to existing models? To answer it, we compare three different architectural patterns (monolithic models, AIN, and ED) leveraging the same experimental setup introduced in Section 3.2. We select a ResNet-18 [20] as the backbone for all the models. We only disentangle the first convolutional layer in AINs, as in preliminary experiments, we observe that this is the minimum amount of additional weights sufficient to improve attribute invariance.

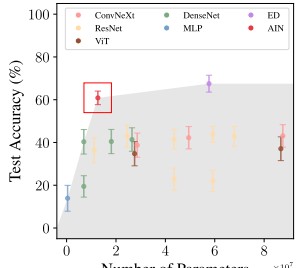

Figure 5: Pareto optimality.

**AINs establish a new Pareto optimal in the compositional generalization task (R3).** Table 1 shows the results of the comparison between the accuracies achieved on the compositional generalization split by the evaluated models (monolithic, AIN, and ED). Similar to ED architectures, AINs consistently and significantly outperform their monolithic counterparts on compositional generalization tasks ($+23.43\%$ average test accuracy). This empirically supports the efficacy of enforcing attribute invariance in gradient updates, highlighting the crucial role that it can play in enhancing compositional generalization. Contrary to ED

| Model | Overhead | Test accuracy (%) | | | | | |
|-------|----------|---------|-------|---------|---------|--------|-------|
| | | Shapes3D | MPI3D | dSprites | I-RAVEN | Cars3D | CLEVR |
| RN-18 | 0% | $85.47^{\pm 0.71}$ | $41.59^{\pm 2.16}$ | $21.43^{\pm 0.63}$ | $11.30^{\pm 2.80}$ | $33.90^{\pm 1.84}$ | $24.68^{\pm 1.86}$ |
| AIN | 6.4%-16% | $85.26^{\pm 0.63}$ | $54.02^{\pm 1.04}$ | $61.43^{\pm 0.33}$ | $66.23^{\pm 2.26}$ | $43.90^{\pm 0.79}$ | $54.30^{\pm 2.15}$ |
| ED | 300%-600% | $96.09^{\pm 2.65}$ | $70.76^{\pm 0.24}$ | $61.31^{\pm 0.61}$ | $76.12^{\pm 2.70}$ | $46.70^{\pm 0.98}$ | $53.98^{\pm 1.46}$ |

Table 1: Compositional generalization accuracy (%) of three architectures (monolithic, AIN, and ED) across 5 datasets. ED and AIN are based on a standard ResNet-18. We report the SEM over three seeds. For each family, we include the parameter overhead compared to the base monolithic model.

architectures, on the other hand, AINs incur a substantially lower parameter overhead. While both approaches exhibit parameter growth that scales linearly with the number of task-relevant generative factors (ranging from 3 to 6 in our experiments), the scaling coefficient for AINs is markedly smaller. Consequently, AINs introduce an overhead between 6.4% and 16% relative to the base model size, significantly lower than the 600% overhead associated with ED models. Overall, AINs achieve a new Pareto-optimal operational point in the scalability-generalization tradeoff, as shown in Figure 5.

## 5 Related work

**Compositional Generalization.** Endowing neural networks with compositional generalization capabilities is a long-standing problem in the literature [29–32]. A wide range of works was produced in the past decades on the topic [5–10, 33–44]. Montero et al. [7, 8, 34] showed that a higher degree of disentanglement does not necessarily translate to better compositional generalization, but limited their investigation to unsupervised and slot-based architectures. Schott et al. [5] focused on supervised learning, showing that many of the inductive biases usually integrated into neural networks are not sufficient to generalize compositionally. More recent works [35, 36] showed that compositional representations are insufficient for generalization due to memorization and shortcut phenomena. On a parallel line of research, others explored compositional generalization in object-centric models [37–41], focusing, however, on *objects* rather than *attributes* as we do. Other works [45, 46] explored attributes disentanglement in object-centric representation learning with monolithic-style models, but did not specifically investigate compositional generalization of these models. While alternative training strategies, such as iterated learning [6, 10], were also proposed as a potential enabler of compositional generalization, this work rather focused on architectural approaches, e.g., fully-disentangled models [9].

**Generating compositional splits.** Different strategies to create compositional evaluation splits were proposed. Montero et al. [7, 8] evaluated a rigorous definition of compositional generalization (*Recombination-to-Range*). Despite hinting that a larger subset of attributes could also be used, they only investigate the exclusion of two attributes at a time (for a few selected pairs of attributes) as in the (inefficient) pair-wise evaluation strategy. On the other hand, other works [5–7, 34] considered a different setting where only a corner of the generative factor hyperspace was excluded. This allowed to decouple the number of training runs from the number of attributes of the dataset but, at the same time, resulted in a more shallow definition of compositionality, where test samples could potentially share up to $I - 1$ task-relevant factors with training samples. While Okawa et al. [11] introduced a concept distance metric akin to our compositional similarity index, our works significantly develop the idea further, proposing a scalable evaluation framework, broader empirical analysis, and a formalization that unifies various generalization regimes under a compositional lens.

**Disentangled representation learning.** Following the initial proposal of Bengio et al. [12], many works explored supervised [9, 16, 47, 48], semi-supervised [49], and unsupervised [50–54] architectures that aimed at separating distinct, independent, and informative generative factors of variation in the data [55]. In practical terms, this ensures linear independence among factors within the model's latent space. In this work, however, we are primarily interested in compositionality, which is concerned with *combinations* of generative factors. The relationship between disentanglement and compositionality remains an open topic of debate to this day. While some works showed that the two are not necessarily correlated [7, 33], others observed that disentangled representations can improve compositionality and generalization in specific settings [9, 56, 57].

# 6 Conclusions, limitations, and future work

**Limitations**   The methods explored in this work – orthotopic evaluation and AINs – rely on the assumption that (at least a subset of) the generative factors of variation are accessible. This is analogous to how evaluating classification accuracy or training supervised models requires access to ground-truth labels. Additionally, while AINs, like ED, scale linearly with the number of attributes, they do so with a significantly lower constant factor, making them substantially more efficient in practice. Finally, we emphasize that neither the attribute-invariant gradient updates used by AINs nor the explicit disentanglement enforced by ED guarantee perfect disentanglement or complete independence among attributes. Rather, they serve as effective mechanisms to promote these properties.

**Conclusions and future works**   In this paper, we investigated compositional generalization in computer vision architectures. In particular, we focused on the problem of supervised learning of compositional and disentangled representations for (known) generative factors underlying visual scenes. Firstly, we proposed a dataset-agnostic and efficient evaluation framework to benchmark compositionality and disentanglement in a principled fashion, unifying previously proposed approaches and increasing the evaluation efficiency. We extensively validated this proposed approach against alternative methodologies and previous works. As part of this validation, we trained more than 5000 SOTA vision models and observed that they still substantially fail to solve the compositional generalization problem. While it is well known that unsupervised learning of disentangled representations is fundamentally impossible without inductive biases [54], we additionally showed that learning disentangled representations is empirically hard even in fully supervised settings (experiments on $c = 1$), even when strong inductive biases on the models' architecture (e.g., ED) are provided. The results also highlight a critical limitation: common vision architectures lack compositionality and disentanglement, which could in turn compromise sample efficiency and increase the vulnerability to failures in real-world environments, where data distributions often exhibit heavy-tailed behavior. Finally, we identify attribute invariance as a necessary precondition for compositional generalization. Inspired by this observation, we propose a new class of neural architectures, Attribute Invariant Networks (AIN), that endows attribute invariance in the gradient computation. Empirically, we show that AINs achieve a new Pareto optimality in the scalability-generalization tradeoff, significantly improving compositional generalization of monolithic models with only a minimal parameter overhead. Possible future works could explore the extension of this study to real-world datasets (e.g., COCO or ImageNet), where the generative factors are noisy and possibly unknown.

## Acknowledgments

We thank Gian Hess for his contributions to the experimental framework. We are grateful to Aleksandar Terzic and Nicolas Menet for the helpful comments and discussions during the course of the project. We thank the reviewers for their valuable feedback, which helped us refine our manuscript and strengthen our contributions. We thank Abu Sebastian for providing managerial support. This work is supported by the Swiss National Science Foundation (SNF), grants 10002666 and 224226.

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

# Supplementary Material

## Table of Contents

# A  Extended orthotopic evaluation details

## A.1  Additional degrees of freedom in the creation of compositional test splits

In the main text, we presented a simplified setting of the framework to pinpoint the specific impact of the $c$ parameter on compositional generalization. However, the generality of the proposed setting can be further increased by relaxing some of the constraints on the evaluation orthotopes. In particular, we identify three additional degrees of freedom that can be specified on top of the minimal setup, namely:

- size of the orthotopes,
- number of orthotopes,
- position of the orthotopes.

The rest of this supplementary note will provide an intuition about what each of these additional parameters influences in the setup in practice and how they can be integrated in the original splitting procedure presented in Algorithm 1.

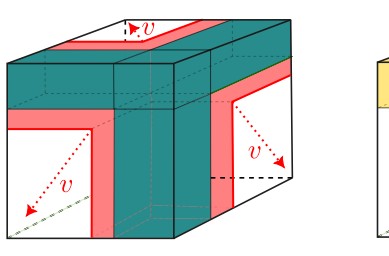 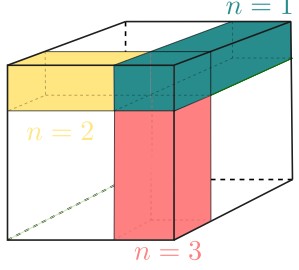 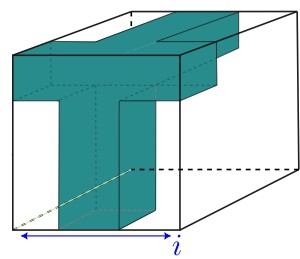

(a) Variable orthotopes' size.   (b) Variable orthotopes' number.   (c) Variable orthotopes' position.

Figure 6: **Additional degrees of freedom in the split generation procedure**. In this figure, we give a pictorial intuition (in the simplified setting of a dataset composed of only 3 generative factors) on how the procedure presented in Algorithm 1 could be extended and generalized further. In **(a)** we parametrize the size of the excluded cylinder with $v$ (indicating the percentage of additional volume excluded with respect to the available, not excluded volume). In **(b)**, we parametrize the number of excluded orthotopes $n$. The colors in this figure show the incrementally increasing volume removed from the training data (e.g., for $n = 2$ the yellow volume is added to the original green volume of $n = 1$). Finally, in **(c)** we parametrize the position of the orthotopes in the hypercube, allowing them to occupy regions which are not exclusively limited to the corners of the cube itself.

### A.1.1  Size of the hypercubes.

In the main set of results, we studied the behavior of $c$ excluding a fixed, minimal number of attribute values for each generative factors. This is, of course, an arbitrary choice, motivated by the observation that *the majority of models were already failing in this simple setting*. In principle, it is possible to make the task arbitrarily harder, excluding, during training, more and more elements from the set of possible values that each attribute can assume. In other words, we can increase the difficulty of the task by enlarging the support of each test orthotope in every $c$-dimensional subspace. An intuition for this generalization is given in Figure 6a. The fact that increasing the volume of the excluded orthotopes decreases compositional generalization is expected for different reasons. Firstly, since the size of the dataset is constant, the number of observations available during training decreases, requiring the models to increase their sample efficiency to retain the same performance. Secondly, increasing the percentage of excluded observations directly reduces data variety, as fewer examples will be available for each specific combination of task relevant generative factors.

In practice, there exist different strategies to implement this, the simplest being adding a parameter $v \in [0, 1]$ that controls the percentage of values excluded for each attribute. For $v = 0$ we recover the main paper setting, while for $v = 1$ all attribute values except one are excluded, since we still need to guarantee the observability of every attribute value during training. All the intermediate steps

can be studied for $0 < v < 1$. This parameter is then used in the `exclusion` subroutine in order to generate exclusion slices of variable size. The pseudo-code for the extended procedure is reported in Algorithm 2.

---

**Algorithm 2** Extended compositional split generation

---

1: **procedure** SPLITDATASET($X$, $G$, $c$, $v$)
2: $\quad \mathcal{S} = \binom{G}{c+1} = \{S \subseteq G : |S| = c+1\}$      ▷ Get all combinations of $c$ generative factors $G$.
3: $\quad$ **for each** $s \in \mathcal{S}$ **do**      ▷ Iterative orthotope pruning.
4: $\quad\quad X_{proj} = \mathbf{proj}_s X$      ▷ Project $X$ onto the $c$-dimensional subspace $s$.
5: $\quad\quad E_s = \text{exclusion}(s, v)$      ▷ Infer excluded slice of the subspace $s$.
6: $\quad\quad X = X \setminus (X_{proj} \cap E_s)$      ▷ Pruning operation in the subspace.

---

### A.1.2   Number of hypercubes

Orthotopic evaluation works by excluding all the orthotopes in $\mathcal{S} = \binom{G}{c+1}$ simultaneously. This guarantees a strict compositional evaluation on *all the attributes at once*. However, this procedure could end up excluding a significant part of the available data (especially when $c$ is small and the number of task-relevant generative factors is large), due to the large number of excluded orthotopes.

On the other side of the spectrum, we find what we referred to as *pair-wise evaluation*, which consists of training a separate model for each pair of generative factors. The advantage of this technique lies in the relatively small amount of percentage of data excluded during training, since we limit $|\mathbf{G}| = 2$ and we (ideally) train models invariant to every other generative factor $\mathbf{O}$. In other words, in pair-wise evaluation, there only exists a single exclusion orthotope in the space. The downsides of this approach, however, are many (corresponding exactly to the advantages of the orthotopic evaluation framework). Firstly, it requires training a different model for each combination of two different generative factors, which results in $\Theta(|\mathbf{F}|^2)$ evaluation complexity, which scales quadratically in the number of generative factors. Secondly, we only train models to predict pairs of attributes whose usefulness is rather limited in practical scenarios.

Once again, however, orthotopic and pair-wise evaluations do not exist in isolation, but are rather the extremes of a discrete spectrum that can be obtained by selecting only a subset of cardinality $n$ of the orthotopes $\hat{\mathcal{S}} \subset \mathcal{S}$. An intuition for this generalization is given in Figure 6b. Naturally, in this case, compositionality can be strictly evaluated only on the tuples of attributes corresponding to the included orthotopes. Nonetheless, these intermediate scenarios might still be interesting to investigate in at least two scenarios: 1) when we only care about evaluating compositional generalization on a subset of attributes, or 2) in situations characterized by an extreme data scarcity or a limited number of values for some attributes, that would be easier to detach from the main dataset and evaluate separately (in order not to influence excessively the results of the entire evaluation). The pseudo-code for this extended procedure is reported in Algorithm 3.

---

**Algorithm 3** Extended compositional split generation

---

1: **procedure** SPLITDATASET($X$, $G$, $c$, $n$)
2: $\quad \mathcal{S} = \binom{G}{c+1} = \{S \subseteq G : |S| = c+1, |\mathcal{S}| = n\}$      ▷ Get all combinations of factors.
3: $\quad$ **for each** $s \in \mathcal{S}$ **do**      ▷ Iterative orthotope pruning.
4: $\quad\quad X_{proj} = \mathbf{proj}_s X$      ▷ Project $X$ onto the $c$-dimensional subspace $s$.
5: $\quad\quad E_s = \text{exclusion}(s)$      ▷ Infer excluded slice of the subspace $s$.
6: $\quad\quad X = X \setminus (X_{proj} \cap E_s)$      ▷ Pruning operation in the subspace.

---

### A.1.3   Position of the hypercubes.

In the vanilla setting, we always assumed the excluded regions to be in some "corner" of the hyperspace. While this simplifies the implementation, it is not necessary, and the excluded orthotopes could be interchangeably placed in any region of the hyperspace.

The relaxation of the excluded slice position could have a measurable impact in settings where a natural *order* for the values of the attributes exists (e.g., size or position). For instance, it would

simplify the task for models that can, to some extent, perform interpolation but not extrapolation (which is another possible way to generalize other than learning compositional representations in the current framework). On the other hand, it does not make a difference for attributes that do not incorporate any notion of ordering in their values (the space would be invariant to shuffling operations). The pseudo-code for this extended procedure is reported in Algorithm 4.

---

**Algorithm 4** Extended compositional split generation

---

1: **procedure** SPLITDATASET($X$, $G$, $c$, $i$)
2:     $\mathcal{S} = \binom{G}{c+1} = \{S \subseteq G : |S| = c + 1\}$         ▷ Get all combinations of factors.
3:     **for each** $s \in \mathcal{S}$ **do**         ▷ Iterative orthotope pruning.
4:         $X_{proj} = \mathbf{proj}_s X$         ▷ Project $X$ onto the $c$-dimensional subspace $s$.
5:         $E_s = \text{exclusion}(s, i)$         ▷ Infer excluded slice of the subspace $s$.
6:         $X = X \setminus (X_{proj} \cap E_s)$         ▷ Pruning operation in the subspace.

---

### A.2 Extension to the new ladder of compositional evaluation difficulty

In the main text, we aimed to provide an intuitive introduction to the new concept of a ladder of compositional evaluation difficulty. In this supplementary material, we further develop some of the aspects introduced in the main text and provide more background to some of the plots reported in this section.

**Broader discussion.** We expand here on the brief introduction to the ladder of compositional generalization difficulty presented in the main text. The ladder is sketched to provide different steps of difficulty related to different ranges of values of the compositional similarity index $c$. The steps are, from the hardest to the simplest:

- **Extrapolation generalization** ($c = 0$); in this setting, part of the examples of the evaluation split share exactly 0 values between training and testing. In other words, they represent unseen values of the generative factors for the model. In the context of the attributes' hypercube, this translates to having full slices (over some dimensions) of the hypercube excluded during training.

- **Compositional generalization** ($1 \leq c \leq I - 1$); in this setting, all the generative factors' values are observed, but a subset of their combination is excluded from the training data. In the context of the attributes' hypercube, this corresponds to excluding partial slices of the hypercube excluded during training.

- **In-distribution generalization** ($c = I$); in this setting, all the generative factors' values, as well as all their possible combinations, are observed. In the hypercube's language, the entire hypercube is effectively observed.

We create a further distinction within the compositional generalization case, between disentangled composition generalization ($c = 1$) and entangled compositional generalization ($1 < c \leq I$). We include an example that should provide an intuition regarding the difference between the entangled ($1 < c \leq I - 1$) and disentangled ($c = 1$) compositional generalization settings. Consider Example 2.2. A model that learns holistic representations for the factors (shape, size) can smoothly generalize to the test example (small, cube, blue obj.) when evaluated with $c = 2$. This is possible because the pair (small, cube) is observable during training, because training and evaluation samples can share up to 2 attributes (including the pair shape-size). However, the same model fails when evaluated with $c = 1$ since the two concepts can only be observed independently in the training data.

In Figure 7, we show an additional experiment to help build up intuition on the different levels of the proposed ladder of compositional evaluation difficulty. In this experiment, we measure the mean cosine similarity between 100 randomly sampled training and testing embeddings for different values of $c$. We consider $c = 6$ and several total attributes $P = 6$, where each attribute can assume 8 different values. Firstly, we randomly sample two vectors $v_{train}$ and $v_{test}$ of size 6, where each element is in the range $[0, 7]$. Then, for each $c \in [0, 6]$, we set the first $c$ elements of the testing vector to be equal to the training vector. This allows us to evaluate the similarity between training and testing observations for the most difficult samples of the testing split, that is, when they only agree on the minimum number of attributes shared between training and testing ($c$). At this point, we encode the attribute values into

distributed vector representations, extracted from a pre-defined codebook of representations of size $(6^8, 1024)$ (one for each possible combination of attributes) for holistic representations and $(8, 2048)$ for compositional representations. We use the same encodings for the values of different attributes; however, this does not influence the resulting cosine similarity, since it is computed position-wise (hence no interferences can happen between attributes). For compositional representation, the final encoding consists of a simple concatenation of each attribute's value encoding (concatenation compositionality). We also experimented with a superposition of the individual attributes' value encodings (additive compositionality) and obtained similar results, thanks to the quasi-orthogonality guaranteed by the high dimensionality of the representations. In $n$-holistic representations, on the other hand, we get from the codebook a representation for the combination of the first $n$ common attributes, with $n \in [2, I - 1]$, and then concatenate it with compositional representations for the remaining attributes. In practice, this simulates the representations that would be extracted by a model that entangles the first $n$ attributes, while learning compositional representations of the remaining $c - n$ attributes. In this scenario, it is expected that properly disentangled representations, where each attribute is encoded separately, have a non-zero ($1/I$) similarity between training and testing similarity also when $c = 1$. At the same time, it is also natural that $n$-holistic representations have 0 similarity with the corresponding training observation despite having $c - 1$ attributes in common, since the representation for those $c - 1$ attributes is not disentangled and depends also on the $c$-th attribute.

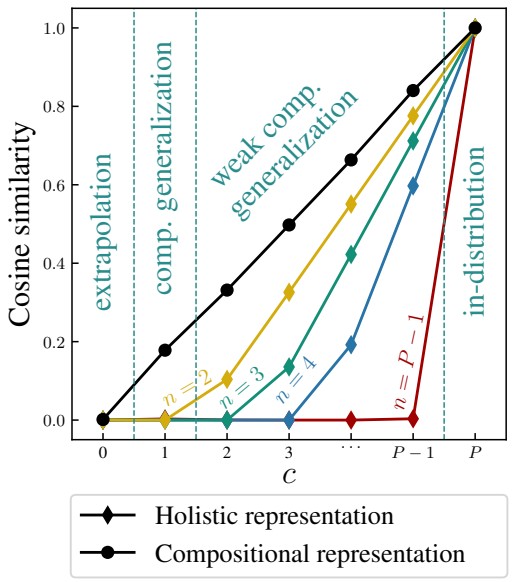

Figure 7: Ladder of compositional evaluation.

## A.3 Orthotopic evaluation framework - an example

Consider the simple dataset with three generative factors, $(g_1, g_2, g_3)$, defined as

$$X = \{(0, 3, 2), (2, 4, 0), (3, 4, 2), (1, 1, 1), (1, 3, 2)\}.$$

Assume that the projection subspace is $(g_1, g_2)$, and the thresholds in this space are $t_1 = 2, t_2 = 1$. In this case, exclude would return all possible evaluation samples (given by the cartesian product of the values $g_i \geq t_i$). Then, $X_{proj} \cap \text{exclusion}(s) = \{(2, 4, 0), (3, 4, 2)\}$, which would be removed from the training data (and automatically become part of its complement, the evaluation data).

# B  Additional experimental setup details

## B.1  Datasets

We consider a range of vision datasets widely used in representation learning literature. Firstly, dSprites [14] is a synthetic dataset of low-resolution binary images of elementary shapes. Similarly, I-RAVEN [15] is a visual analogical reasoning dataset containing synthetic images of B/W simple sprites. Shapes3D [16] is a well-known dataset in the representation learning domain containing synthetic low-resolution images of 3-dimensional solids in a simple environment. On a similar note, CLEVR [17] is another dataset initially proposed in the context of elementary visual reasoning containing higher-quality renderings of solid shapes with more sophisticated generative factors compared to Shapes3D (e.g., material). Cars3D [18] is a dataset containing low-resolution synthetic renderings of simple cars' CAD models (increased subject complexity). We also include a real-world dataset, MPI3D [19], containing pictures of physical solids attached to a robotic finger.

**I-RAVEN** I-RAVEN [15] is a modified version of the RAVEN [58] dataset, which removes biases in the answer sets. RAVEN is derived from the Raven Progressive Matrices (RPM) [59], a widely accepted test of human abstract visual reasoning capabilities. The RAVEN dataset serves to assess the visual reasoning capabilities of machines. The perception of the objects present in a panel and their attributes is the first step in solving an abstract visual reasoning task. For a system to generalize in visual abstract reasoning, it must also demonstrate strong generalization in perception. While I-RAVEN consists of multiple panels per sample, we treat each individual panel as an independent sample. Examples from the dataset are illustrated in Figure 8. The dataset features the following generative factors:

- shape (5 classes)
- size (6 classes)
- color (10 classes)
- angle (8 classes)

While the shape attribute usually has no natural order, shapes in I-RAVEN can be ordered by their number of vertices. Out of these generative factors, we exclude angle because of its ambiguity, hence considering $\mathbf{G} = \{\text{shape}, \text{size}, \text{color}\}$ and $\mathbf{O} = \{\text{angle}\}$.

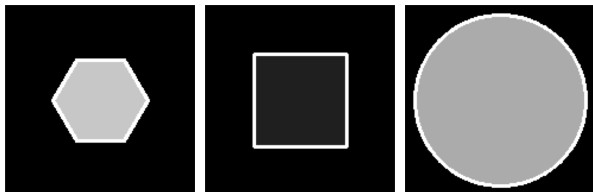

Figure 8: Three images from the I-RAVEN dataset.

**dSprites** dSprites [14] is a procedurally generated dataset of 2D shapes. The dataset was created to evaluate the disentanglement properties of unsupervised learning methods and is widely used as a benchmark. It features very simple objects with four independent factors of variation. Samples from the dataset are shown in Figure 9. Objects in the dataset have the following generative factors:

- shape (3 classes)
- scale (6 classes)
- orientation (40 classes)
- $x$-position (32 classes)
- $y$-position (32 classes)

Out of these generative factors, we exclude orientation because of its ambiguity, and effectively consider $\mathbf{G} = \{\text{shape}, \text{scale}, x, y\}$ and $\mathbf{O} = \{\text{orientation}\}$.

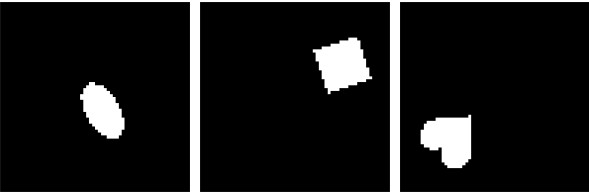

Figure 9: Three images from the dSprites dataset.

**Shapes3D**   3D Shapes [16] is a procedurally generated dataset of 3D shapes. Like dSprites, it was created to assess the ability of models to reveal latent generative factors. Unlike dSprites, the objects are three-dimensional, and the objects are in an environment that also has attributes, such as the color of the wall behind an object. Samples from the dataset are shown in Figure 10. Images in 3D Shapes have the following generative factors:

- floor color (10 classes)
- wall color (10 classes)
- object color (10 classes)
- scale (8 classes)
- shape (4 classes)
- orientation (15 classes)

Among these generative factors, we select all of them except for orientation to be task-relevant, yielding $\mathbf{G} = \{\text{shape}, \text{scale}, \text{floor hue}, \text{object hue}, \text{wall hue}\}$ and $\mathbf{O} = \{\text{orientation}\}$.

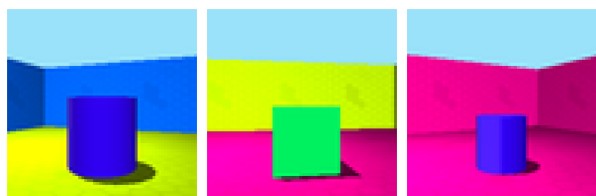

Figure 10: Three images from the 3D Shapes dataset.

**MPI3D**   MPI3D datasets are designed to benchmark representation learning algorithms in both simulated and real-world environments. We use the *Real world simple shapes* dataset, which is a dataset of photographs of a robot arm holding up various simple objects [19]. In contrast to the other datasets used in this work, this is effectively a real-world dataset. Samples from the dataset are shown in Figure 11. The generative factors for this dataset are:

- color (6 classes)
- shape (6 classes)
- size (2 classes)
- height (3 classes)
- background color (3 classes)
- x-axis (40 classes)
- y-axis (40 classes)

To avoid excluding a disproportionate amount of observations, we do not consider the size attribute (that can only assume two values) among the task-relevant generative factors. Hence, for this dataset, $\mathbf{G} = \{\text{color}, \text{shape}, \text{height}, \text{bgcolor}, x, y\}$ and $\mathbf{O} = \{\text{orientation}\}$.

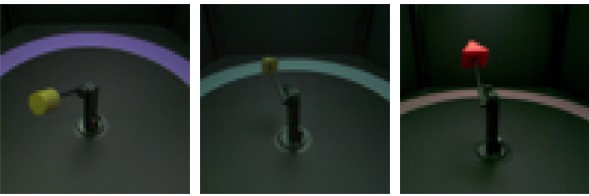

Figure 11: Three images from the MPI3D dataset.

**Cars3D** Cars3D was introduced by [18] to study the problem of visual analogy-making using deep neural networks. The dataset contains images of a wide variety of cars. Among the datasets used in this work, its type attribute has the largest number of classes. Samples from the dataset are shown in Figure 12. The generative factors for this dataset are:

- orientation (24 classes)
- elevation (4 classes)
- type (183 classes)
- height (3 classes)

Because of its ambiguity, we exclude the height parameter from the task-relevant generative factors. Effectively, we consider $\mathbf{G} = \{\text{elevation}, \text{type}, \text{orientation}\}$ and $\mathbf{O} = \{\text{height}\}$.

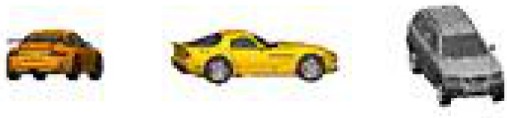

Figure 12: Three images from the 3D Cars dataset.

**CLEVR** CLEVR [17] is a synthetic vision dataset initially introduced in the domain of elementary visual reasoning. This dataset is composed of high-quality images of solids rendered with Blender. Compared to other synthetically generated datasets, the objects in this dataset include a wider range of object-level generative factors (e.g., the object texture). Samples from the dataset are shown in Figure 13. The generative factors for this dataset are:

- shape (3 classes)
- size (3 classes)
- material (2 classes)
- color (8 classes)

In this case, $\mathbf{G} = \mathbf{F} = \{\text{shape}, \text{size}, \text{material}, \text{color}\}$, while $\mathbf{O} = \emptyset$.

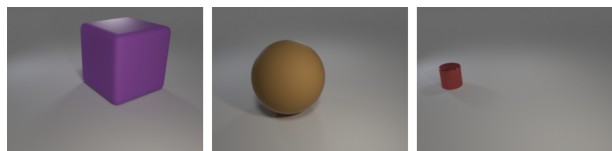

Figure 13: Three images from the CLEVR dataset.

As mentioned in the main text, we define attribute-wise exclusion thresholds for the orthotopic evaluation experiments, such that roughly 60% of the observations of each dataset are used for training and the remaining 40% for testing. We report in Table 2 the thresholds for each dataset. The reported exclusion thresholds correspond to the attributes reported in the same table and correspond to the lowest index excluded from the interval of attributes. Indices are considered in the interval $[0, n-1]$, where $n$ is the number of values that the attribute can assume.

| Dataset | Attributes | $c$ | Exclusion thresholds |
|---|---|---|---|
| dSprites | shape, scale, $x$, $y$ | 0 | $2, 4, 30, 30$ |
| | | 1 | $2, 3, 14, 14$ |
| | | 2 | $1, 3, 16, 16$ |
| | | 3 | $1, 1, 4, 4$ |
| I-RAVEN | shape, scale, $x$, $y$ | 0 | $9, 5, 4$ |
| | | 1 | $6, 3, 3$ |
| | | 2 | $3, 2, 1$ |
| Cars3D | orientation, elevation, type | 0 | $26, 3, 160$ |
| | | 1 | $15, 2, 113$ |
| | | 2 | $6, 1, 43$ |
| CLEVR | shape, size, material, color | 0 | $2, 2, 1, 7$ |
| | | 1 | $2, 2, 1, 7$ |
| | | 2 | $2, 1, 1, 3$ |
| | | 3 | $1, 1, 1, 1$ |
| Shapes3D | floor, wall, obj, scale, shape | 0 | $9, 9, 9, 7, 3$ |
| | | 1 | $7, 7, 7, 6, 3$ |
| | | 2 | $6, 5, 6, 5, 2$ |
| | | 3 | $4, 4, 4, 3, 1$ |
| | | 4 | $2, 2, 1, 1, 1$ |
| MPI3D | color, shape, size, height, bgcolor, $x$, $y$ | 0 | $5, 5, 2, 2, 38, 38$ |
| | | 1 | $5, 4, 2, 2, 34, 34$ |
| | | 2 | $4, 3, 2, 2, 27, 27$ |
| | | 3 | $3, 4, 1, 2, 22, 22$ |
| | | 4 | $2, 2, 1, 1, 10, 10$ |
| | | 5 | $1, 1, 1, 1, 1, 1$ |

Table 2: Attribute-wise exclusion thresholds used for each dataset.

## B.2   Models

In this section, we provide additional details on the models used in our experiments. Most of the models' implementation used in the experiments are taken from the `torchvision.models` zoo. The models are either directly used or extended in the case of custom implementations (e.g., in the case of the ablation of alternative activation functions presented in Appendix C.4).

The MLP model is taken from Schott et al. [5], and consists of the following layers: `[Linear(64*64*number-channels, 90), ReLU(), Linear(90, 90), ReLU(), Linear(90, 90), ReLU(), Linear(90, 90), ReLU(), Linear(90, number-factors)]`.

The checkpoints used for the pre-trained models are, respectively,

- DenseNet-121: `https://download.pytorch.org/models/densenet121-a639ec97.pth`;
- ResNet-101: `https://download.pytorch.org/models/resnet101-cd907fc2.pth`;
- DenseNet-152: `https://download.pytorch.org/models/resnet152-f82ba261.pth`;

The proposed Explicitly Disentangled (ED) model instantiates a different net (RN-18) for every task-relevant generative factor. Compared to Separate Architecture (SA) [9], ED presents different improvements. First, for single-channel inputs, ED adjusts the ResNet to be single channel instead of repeating the input on three channels as SA does. Second, ED performs *maxpool* before *BatchNorm*, while SA does it after. Then, ED uses a *tanh* activation before the final linear layer and no output activation, while the SA uses *ReLU* as output activation. Finally, a major difference lies in the type of readout used by the model, as discussed in the following Appendix B.3.

The same pre-processing is applied to the input of every model. The pre-processing is dataset-dependent and corresponds to the following augmentations:

- Cars3D $\rightarrow$ `[resize(64)]`
- CLEVR $\rightarrow$ `[resize(128)]`
- dSprites $\rightarrow$ `[resize(64), gaussBlur(kernel=(23,23), sigma=(0.1,0.3)]`
- I-RAVEN $\rightarrow$ `[resize(64), pad(5), randCrop(64), randRotation(360), gaussBlur(kernel=(41,41), sigma=(0.1,0.3)]`
- MPI3D $\rightarrow$ `[resize(64), gaussBlur(kernel=(23,23), sigma=0.2]`
- Shapes3D $\rightarrow$ `[resize(64), gaussBlur(kernel=(23,23), sigma=0.2]`

Figure 14 includes a conceptual overview of the differences between the different model blueprints studied in this work. For monolithic models, the same network is used to extract a unique representation of the input image, which is then fed into the different classification heads that predict logits for the individual attributes. On the other hand, ED models disentangle the full feature extraction and feed a representation which, ideally, should only contain information related to a specific attribute. Attribute invariant networks represent a compromise between the two: an initial part of the feature extraction is disentangled, but at a certain point, all the representations are concatenated over a new dimension and processed using the same weights. The advantage of AINs lies in their parameter efficiency: the shared weights in the second part of the feature extraction do not need to be replicated for each attribute, as the model learns a single set of weights shared across different attributes. ED can be seen as a special case of AINs, where the entire model is disentangled and there are no shared parameters.

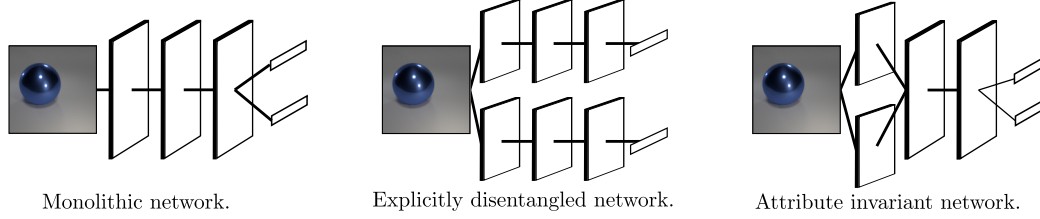

Monolithic network.                Explicitly disentangled network.        Attribute invariant network.

Figure 14: Comparison between different architectural blueprints studied in this work.

## B.3 Linear and FPE-based readouts for ED

In the Shared Architecture from Madan et al. [9], the prediction of the attributes' logits is performed through simple linear readouts on top of the extracted attribute representations. Building on this, our ED architecture aims to further improve the performance on compositional generalization by encoding prior knowledge regarding relationships between attributes. Specifically, it encourages the model to respect the structural similarity of neighboring classes. This enhancement is particularly valuable for attributes that endow a natural definition of ordering, where the similarity between adjacent attribute values (e.g., color shades or size increments) should be reflected in the learned representations. The ED model leverages fractional power encodings (FPE) [60, 61] to encode these attributes in a structured way, preserving task-relevant patterns in the embeddings and endowing similarity between representations.

In FPE, an integer attribute $i$ is encoded by binding a random base vector $\mathbf{z} \sim p(\mathbf{z})$ (where $p(\mathbf{z})$ depends on the specific initialization scheme, in our case $\mathcal{N}(0, 1)$), known as a phasor, to itself $i$ times.

$$\mathbf{z}(i) := \mathbf{z}^{(\circ i)} = \mathbf{z} \circ ... \circ \mathbf{z} \tag{1}$$

We use circular convolution as the binding operation, which can be efficiently computed via the Fast Fourier Transform (FFT), where binding corresponds to element-wise multiplication in the frequency domain. This also allows for adjusting the relative distance between encoded values to non-integer values by multiplying by a parameter $\alpha \in \mathbb{R}$:

$$\mathbf{z}(i, \alpha) = F^{-1}(F\mathbf{z})^{\alpha \cdot i} \tag{2}$$

where $F$ and $F^{-1}$ are the Vandermonde matrices representing the discrete Fourier transform and its inverse.

The distribution of phases in the FPE vectors determines the shape of the kernel in the embedding space [61]. We leverage a normal distribution, which leads to the kernel induced by the vector product of the embeddings approximating a Gaussian kernel (shown in Figure 15). Increasing the variance of the phase distribution leads to a decrease in kernel width. We adjust the variance of the phase distribution based on each attribute's specific characteristics, using a larger variance for ungraded attributes like type and a smaller variance for graded attributes such as size. This allows the model to incorporate prior knowledge regarding the relationships between classes, ensuring that the similarity between attributes' values is maintained in their corresponding representations and can be eventually leveraged by the model.

How is this form of initialization implemented in the model? During initialization, a base phasor is sampled. Based on it, a representation vector for each possible value of each attribute is produced (following the previously reported methodology). These vectors are then aggregated in a codebook $\mathbf{C}$ and become an invariant component of the model. At inference time, each attribute's representation extracted by the model is compared with the corresponding $\mathbf{C}$ using cosine similarity, yielding $n$ different similarity values (one for each attribute's value). The final prediction is eventually produced by taking the argmax of these similarity indices, i.e., computing the index of the most similar representation in $\mathbf{C}$. The model is trained using the cross-entropy loss between the ground-truth one-hot vector and the vector of cosine similarities.

In practice, FPE readouts perform slightly better than linear readouts on average. The results of an ablation between the two types of readout are included in Figure 16. We can observe that, for most of the datasets, the difference between the two is marginal. However, in the case of dSprites FPE

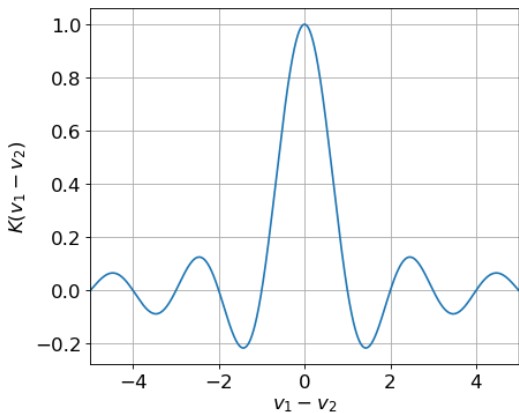

Figure 15: Kernel induced by the FPE initialization on the target representations. Computing the similarity between two attribute values $v_1$ and $v_2$ in the representation space yields the shown similarity kernel $K(v_1 - v_2)$.

readouts significantly improve the results. On the other hand, FPE readouts perform slightly worse than linear ones on MPI3D.

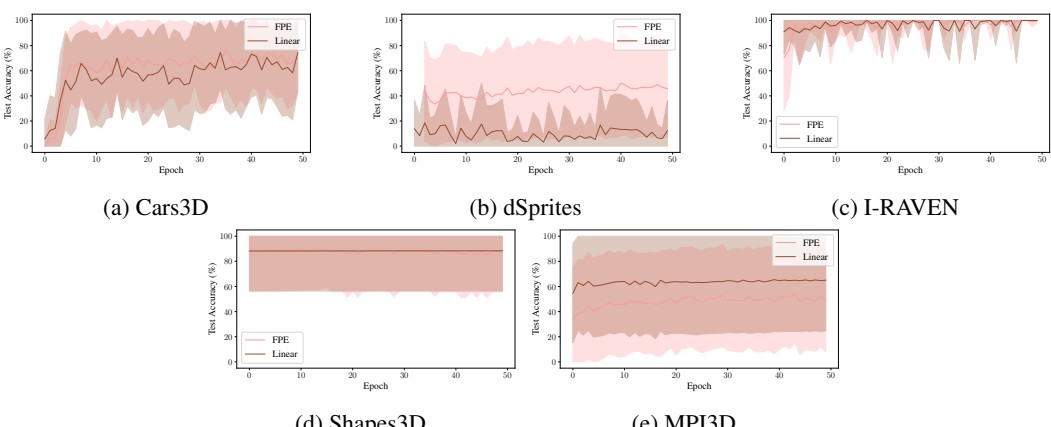

Figure 16: Ablation on the difference between linear and kernel-based readouts for the ED model. The standard deviation is reported across five different random seeds.

## B.4 Model selection

We include here a small ablation on the metric used for model selection in our experiments. Identifying a robust and consistent selection metric is, indeed, of vital importance to select models that balance in-distribution (ID) and out-of-distribution (OOD) performances. Ideally, we would like to select a model that performs well on both ID and OOD settings. However, in the majority of the situations, there is a trade-off between the two: models that perform well in-distribution generalize poorly in OOD scenarios, whereas models that perform well OOD are usually underfit ID.

Based on these observations, we investigate three different selection metrics:

- **In-distribution validation accuracy** (ID); this metric is computed on a held-out split of 10% of the training data (containing only the training combinations of attributes) and validates in-distribution generalization.

- **Out-of-distribution validation accuracy** (OOD); this metric is computed on samples from a subset of combinations of factors (in our experiments, one combination) held out from the training data, validating compositional generalization.

- **Weighted in-out-distribution validation accuracy** (WIO), representing a weighted composition of ID and OOD selection metrics. In particular, we define $\text{val}_{\text{WIO}} = \text{val}_{\text{ID}} + {}^1/\lambda \cdot (\text{val}_{\text{OOD}} - 100)$, where $\lambda$ is an hyperparameter that allows to balance the relative importance of the two. In our experiments, we set $\lambda = 10$ (OOD validation data only marginally influences the WIO accuracy) to avoid the selection of "under-fit" models that perform well out-of-distribution but under-perform in-distribution.

We compare the test accuracy achieved by the models selected based on each metric against the best test accuracy achieved at any point in time during the training by the model (oracle test accuracy). The comparison is performed using all the training traces from the pair-wise experiments presented in Section C, including every dataset, model, seed, and attribute combination. The results of this ablation are included in Figure 17. From these results, we can observe how the in-distribution validation accuracy is usually a better metric for model selection with an average accuracy delta with the oracle selection equal to 9.26%. ID is closely followed by WIO, which shows an increase of 0.71% in the test accuracy delta. Finally, OOD proved to be the worst selection metric among the investigated ones, with an average $\Delta$ from the oracle selection of 9.97% test accuracy. Hence, in this work, we opt to use the ID validation accuracy itself to perform model selection. However, we highlight that the gap between the selected model and the best-performing model on test data is still large ($\approx 10\%$), which means that significant gains in future works could still be achieved by discovering more robust and balanced selection metrics.

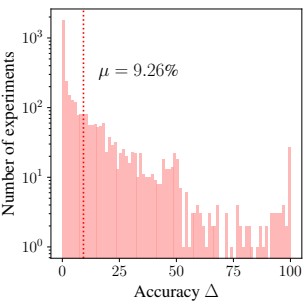
(a) $\Delta$ between oracle and ID-selected test accuracy.

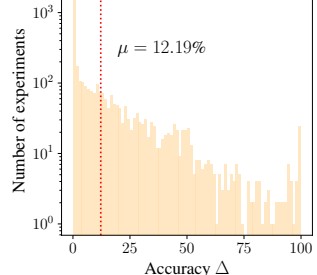
(b) $\Delta$ between oracle and OOD-selected test accuracy.

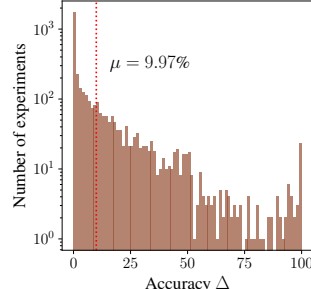
(c) $\Delta$ between oracle and WIO-selected test accuracy.

Figure 17: Ablation of different model selection metrics.

## C    Evaluating pair-wise compositional generalization

In this supplementary section, we include additional results on compositional generalization evaluated on combinations of only two attributes (pair-wise evaluation). This corresponds to the special case of the general orthotopic evaluation formulation presented in Section A.1, where only one orthotope is excluded from the training data and used for evaluation purposes. Generating the splits corresponds to projecting the entire dataset onto two dimensions, one for each generative factor, and excluding the observations corresponding to specific combinations of the factors. The process is shown in Figure 18 (a) for a fictitious example where the generative factors are shape and color. In practice, the models are trained to predict only two generative factors $g_1 \in \mathbf{G}, g_2 \in \mathbf{G}$ while being invariant to all other factors $\mathbf{G} \setminus \{g_1, g_2\}$. A general blueprint of the inference pipeline in this setting is shown in Figure 18 (b). By definition, pair-wise evaluation is expensive in terms of the number of training and evaluation runs (scaling quadratically in the number of attribute combinations) since a different model has to be trained for each combination. However, it maximizes the size of the training split and evaluates compositionality with a fine granularity on the attributes.

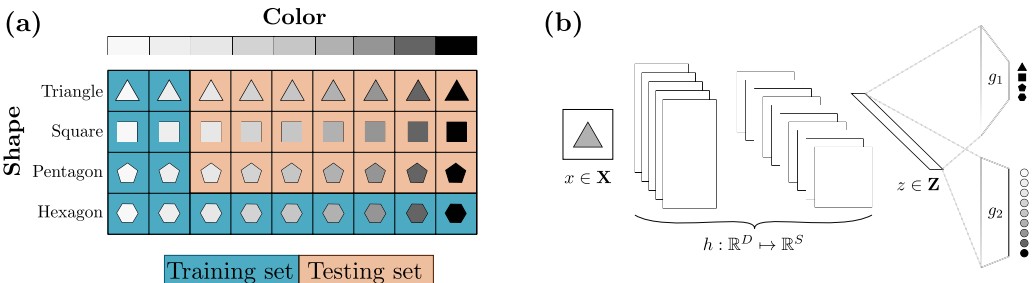

Figure 18: Pairwise compositional evaluation.

### C.1    Experimental setup

**Datasets and models.**   We investigate pair-wise compositional generalization on a subset of the datasets (dSprites, I-RAVEN, Cars3D, Shapes3D, and MPI3D) and the whole range of models studied in the main paper (monolithic models, such as ResNets, DenseNets, ViT, MLP, ConvNeXt, and a supervised disentangled model, ED). Contrary to the experiments on orthotopic evaluation, in this experiments we only considered a subset of the generative factors of each dataset. In particular, we include:

- *Cars3D* → elevation, type, orientation;
- *dSprites* → scale (size), shape;
- *I-RAVEN* → color, type (shape), size;
- *Shapes3D* → floor hue, wall hue, obj. hue, scale (size), shape;
- *MPI3D* → color, background, height, shape, size.

To average out the effect of stochasticity in the training process, we repeat every training run with 5 different seeds and average the results on them. Also in this case, motivated by the ablation in Section B.4, we use in-distribution validation accuracy (held-out validation set of 10% of the training data) to perform model selection.

**Split difficulty.**   The pair-wise evaluation setting is equivalent to orthotopic evaluation with fixed $c = 1$, considering only a subset of 2 attributes at a time. We fix the percentage of the combinations excluded during training at $10\%$ and find adaptive attribute-wise thresholds for every dataset and attribute combination. The adaptive discovery of the attribute-wise threshold is performed using a constraint satisfaction solver (CSP), which adjusts the exclusion of combinations to match the desired difficulty fraction closely. Since directly optimizing the number of excluded combinations for each attribute can lead to imbalanced difficulties, where some attributes are disproportionately affected by the exclusion process, we add a regularization to the objective function. This regularization term penalizes large disparities between the relative exclusion sizes of different attributes, thus encouraging a more evenly distributed difficulty across different attributes.

## C.2 Experimental results

In this section, we report the experimental results of the large-scale evaluation on pair-wise compositional generalization, obtained training a total of more than 3600 different models. We report the evaluation results for each dataset, including training accuracy (in-distribution, training data), ID validation accuracy (in-distribution, held-out data), OOD validation accuracy (composition generalization, held-out from the training data), and test accuracy (OOD testing split). The mean accuracy and the Standard Error of the Mean (SEM, w.r.t. different attribute combinations and seeds) are reported for each model.

**Cars3D** The results for the Cars3D dataset are included in Table 3. The explicit disentanglement (ED) model only slightly outperforms all the other baselines in-distribution but significantly overperforms in both OOD validation (84.24%, 4.2% more than the closest monolithic model) and, most importantly, test accuracy (73.38%, 13.73% more than the closest monolithic model). The majority of the models completely memorized the training data (100% accuracy) and converged in-distribution (95% validation accuracy) but failed to learn a solution that could generalize well to unseen combinations of the generative factors.

| Model | Pretrained | Train | | ID Val | | OOD Val | | Test | |
|---|---|---|---|---|---|---|---|---|---|
| | | AVG | SEM | AVG | SEM | AVG | SEM | AVG | SEM |
| MLP | ✗ | 47.50 | 5.72 | 47.14 | 5.78 | 12.51 | 5.66 | 6.11 | 1.32 |
| ResNet-18 | ✗ | 99.10 | 0.17 | 95.85 | 0.31 | 68.45 | 10.18 | 52.31 | 6.56 |
| ResNet-34 | ✗ | 99.08 | 0.16 | 95.91 | 0.32 | 65.02 | 9.87 | 52.73 | 6.23 |
| ResNet-50 | ✗ | 99.02 | 0.18 | 95.68 | 0.39 | 78.25 | 8.96 | 57.21 | 4.64 |
| ResNet-101 | ✗ | 99.00 | 0.18 | 95.67 | 0.42 | 77.24 | 9.07 | 59.65 | 4.26 |
| ResNet-101 | ✓ | 98.94 | 0.14 | 82.48 | 1.62 | 45.77 | 10.03 | 26.34 | 2.17 |
| ResNet-152 | ✗ | 99.09 | 0.20 | 95.15 | 0.52 | 73.20 | 9.72 | 55.93 | 4.46 |
| ResNet-152 | ✓ | 99.03 | 0.16 | 83.47 | 1.41 | 49.32 | 9.35 | 28.85 | 2.37 |
| DenseNet-121 | ✗ | 99.12 | 0.16 | 96.06 | 0.29 | 70.14 | 9.72 | 56.21 | 5.92 |
| DenseNet-121 | ✓ | 48.96 | 2.71 | 29.11 | 2.30 | 19.07 | 6.81 | 6.56 | 1.21 |
| DenseNet-161 | ✗ | 99.06 | 0.19 | 96.06 | 0.27 | 67.46 | 12.14 | 61.07 | 6.64 |
| DenseNet-201 | ✗ | 99.03 | 0.18 | 95.74 | 0.29 | 69.14 | 12.16 | 59.54 | 6.28 |
| ConvNeXt-small | ✗ | 98.87 | 0.22 | 94.03 | 0.43 | 68.15 | 10.01 | 51.41 | 6.12 |
| ConvNeXt-base | ✗ | 98.88 | 0.22 | 93.98 | 0.41 | 66.38 | 9.86 | 51.90 | 5.78 |
| WideResNet | ✗ | 99.05 | 0.18 | 95.79 | 0.43 | 80.04 | 8.79 | 59.44 | 4.62 |
| ViT | ✗ | 98.86 | 0.18 | 87.68 | 1.17 | 45.53 | 10.21 | 31.11 | 5.01 |
| Swin-tiny | ✗ | 98.85 | 0.16 | 93.10 | 0.38 | 64.21 | 8.48 | 43.96 | 6.62 |
| Swin-base | ✗ | 98.80 | 0.16 | 93.93 | 0.33 | 60.76 | 8.52 | 51.32 | 5.73 |
| ED | ✗ | 99.06 | 0.18 | **96.37** | 0.33 | **84.24** | 8.95 | **73.38** | 4.62 |

Table 3: Pairwise evaluation on the Cars3D benchmark.

**dSprites** A similar behavior was also observed in the dSprites dataset, whose evaluation results are reported in Table 4. Compared to the Cars3D, here the in-distribution results for almost every model reach exactly 100%. This is expected to some extent, since the dataset is much more simplistic than Cars3D (containing only binary low-resolution images of simple geometric shapes). However, the failure on compositional OOD data is symmetrically more pronounced: every monolithic model (with very few exceptions, such as the ResNet-152 and ResNet-50) score exactly 0% on test data. ED, on the other hand, is then only model whose performances suffer from only a minor drop between in- and out-of-distribution, achieving an average test accuracy of 84.25%. This impressive drop of almost every model might be a result of different factors. Firstly, the models could possibly be too overparametrized for task and have enough capacity for simply memorizing entangled information about the combinations perfectly, virtually reducing the pressure to learn compositional representation to zero. Furthermore, in this dataset the only combination studied is the combination of shape and size attributes, which is notoriously difficult (e.g., also Montero et al. [8] observed it) but in other datasets is averaged by many other "easier" combinations of attributes.

| Model | Pretrained | Train | | ID Val | | OOD Val | | Test | |
|---|---|---|---|---|---|---|---|---|---|
| | | AVG | SEM | AVG | SEM | AVG | SEM | AVG | SEM |
| MLP | ✗ | 98.46 | 0.35 | 98.68 | 0.35 | 0.01 | 0.01 | 0.00 | 0.00 |
| ResNet-18 | ✗ | 100.00 | 0.00 | 100.00 | 0.00 | 0.00 | 0.00 | 0.00 | 0.00 |
| ResNet-34 | ✗ | 100.00 | 0.00 | 100.00 | 0.00 | 0.00 | 0.00 | 0.00 | 0.00 |
| ResNet-50 | ✗ | 100.00 | 0.00 | 100.00 | 0.00 | 0.01 | 0.01 | 10.79 | 10.79 |
| ResNet-101 | ✗ | 100.00 | 0.00 | 100.00 | 0.00 | 0.01 | 0.01 | 0.00 | 0.00 |
| ResNet-101 | ✓ | 100.00 | 0.00 | 100.00 | 0.00 | 0.00 | 0.00 | 0.00 | 0.00 |
| ResNet-152 | ✗ | 100.00 | 0.00 | 100.00 | 0.00 | 2.73 | 2.73 | 17.77 | 11.58 |
| ResNet-152 | ✓ | 99.99 | 0.00 | 100.00 | 0.00 | 0.00 | 0.00 | 0.00 | 0.00 |
| DenseNet-121 | ✗ | 100.00 | 0.00 | 100.00 | 0.00 | 0.00 | 0.00 | 0.00 | 0.00 |
| DenseNet-121 | ✓ | 59.31 | 0.14 | 59.27 | 0.17 | 0.63 | 0.27 | 0.51 | 0.16 |
| DenseNet-161 | ✗ | 100.00 | 0.00 | 100.00 | 0.00 | 0.00 | 0.00 | 0.00 | 0.00 |
| DenseNet-201 | ✗ | 100.00 | 0.00 | 100.00 | 0.00 | 0.00 | 0.00 | 0.00 | 0.00 |
| ConvNeXt-small | ✗ | 99.97 | 0.01 | 100.00 | 0.00 | 0.00 | 0.00 | 0.00 | 0.00 |
| ConvNeXt-base | ✗ | 99.98 | 0.01 | 100.00 | 0.00 | 2.22 | 2.22 | 0.00 | 0.00 |
| WideResNet | ✗ | 100.00 | 0.00 | 100.00 | 0.00 | 6.76 | 4.71 | 0.09 | 0.07 |
| ViT | ✗ | 91.42 | 3.93 | 95.12 | 2.53 | 0.00 | 0.00 | 0.00 | 0.00 |
| Swin-tiny | ✗ | 98.95 | 0.27 | 99.74 | 0.08 | 0.00 | 0.00 | 0.00 | 0.00 |
| Swin-base | ✗ | 17.94 | 4.52 | 22.40 | 6.11 | 0.00 | 0.00 | 0.00 | 0.00 |
| ED | ✗ | 100.00 | 0.00 | 100.00 | 0.00 | **79.87** | 19.97 | **84.25** | 15.69 |

Table 4: Pairwise evaluation on the dSprites benchmark.

**I-RAVEN** The results for the I-RAVEN dataset are reported in Table 5. Also in this case the realization of an explicit disentanglement of the generative factors for the forward and backward passes in the architecture itself proves to be an effective strategy, with ED achieving 100% on both in-distribution and out-of-distribution samples. Monolithic models, on the other hand, achieve in most of the cases perfect accuracy in-distribution but fail to score more than 70% on the compositional generalization split. Contrary to the previous dataset, for this dataset more recent architectures such as ConvNeXts and Swin Transformers achieve noticeably better results compared to older backbones, with best accuracies of 68.78% and 62.05% respectively. As in the other datasets, also for I-RAVEN the pre-training on Imagenet-1k never brings substantial benefit and, in some cases, even prevents the models from converging on the training data itself. Compared to ResNets, WideResNets and DenseNets also perform better, achieving at least 7.08% and 12.32% higher accuracy on the test split.

| Model | Pretrained | Train | | ID Val | | OOD Val | | Test | |
|---|---|---|---|---|---|---|---|---|---|
| | | AVG | SEM | AVG | SEM | AVG | SEM | AVG | SEM |
| MLP | ✗ | 95.41 | 1.05 | 99.33 | 0.26 | 43.87 | 12.18 | 6.85 | 2.28 |
| ResNet-18 | ✗ | 99.97 | 0.01 | 100.00 | 0.00 | 79.21 | 10.11 | 37.67 | 6.44 |
| ResNet-34 | ✗ | 99.97 | 0.01 | 100.00 | 0.00 | 75.10 | 11.14 | 40.06 | 8.49 |
| ResNet-50 | ✗ | 99.98 | 0.01 | 100.00 | 0.00 | 74.71 | 10.80 | 34.96 | 7.23 |
| ResNet-101 | ✗ | 99.96 | 0.01 | 100.00 | 0.00 | 86.71 | 7.69 | 47.41 | 9.08 |
| ResNet-101 | ✓ | 99.97 | 0.01 | 100.00 | 0.00 | 61.59 | 12.55 | 16.36 | 4.90 |
| ResNet-152 | ✗ | 99.87 | 0.05 | 100.00 | 0.00 | 84.77 | 8.08 | 43.98 | 10.11 |
| ResNet-152 | ✓ | 99.97 | 0.01 | 100.00 | 0.00 | 55.42 | 12.14 | 9.20 | 3.31 |
| DenseNet-121 | ✗ | 99.98 | 0.01 | 100.00 | 0.00 | 80.03 | 9.79 | 59.73 | 12.11 |
| DenseNet-121 | ✓ | 76.95 | 1.48 | 72.38 | 1.65 | 16.47 | 2.69 | 14.44 | 2.56 |
| DenseNet-161 | ✗ | 99.99 | 0.00 | 100.00 | 0.00 | 77.11 | 10.57 | 49.01 | 11.77 |
| DenseNet-201 | ✗ | 99.98 | 0.01 | 100.00 | 0.00 | 80.90 | 9.01 | 25.29 | 10.08 |
| ConvNeXt-small | ✗ | 99.79 | 0.07 | 100.00 | 0.00 | 68.65 | 10.69 | 68.78 | 11.57 |
| ConvNeXt-base | ✗ | 99.41 | 0.21 | 100.00 | 0.00 | 85.52 | 6.25 | 68.75 | 11.70 |
| WideResNet | ✗ | 99.83 | 0.07 | 100.00 | 0.00 | 79.83 | 10.67 | 54.49 | 9.83 |
| ViT | ✗ | 99.56 | 0.12 | 99.99 | 0.00 | 53.30 | 13.03 | 57.93 | 11.39 |
| Swin-tiny | ✗ | 99.64 | 0.06 | 100.00 | 0.00 | 56.60 | 12.30 | 58.79 | 10.20 |
| Swin-base | ✗ | 93.10 | 6.34 | 93.58 | 6.40 | 50.93 | 10.94 | 62.09 | 10.74 |
| ED | ✗ | 100.00 | 0.00 | 100.00 | 0.00 | **100.00** | 0.00 | **100.00** | 0.00 |

Table 5: Pairwise evaluation on I-RAVEN.

**Shapes3D** The results for the Shapes3D dataset are included in Table 6. Yet another time, ED achieves the best accuracy on the compositional test split, scoring almost perfect accuracy on generalization data. Monolithic models are also competitive in this benchmark. ConvNeXts, as in I-RAVEN, achieve the best performance with 93.82% OOD accuracy, closely followed by ResNets (90.03%) and Swin Transformers (89.02%). In-domain, all the models (except for the pre-trained DenseNet-121) achieve perfect accuracy (both on training and validation data). Overall, good performances on this specific benchmark were also observed in previous studies [5].

| Model | Pretrained | Train | | ID Val | | OOD Val | | Test | |
|---|---|---|---|---|---|---|---|---|---|
| | | AVG | SEM | AVG | SEM | AVG | SEM | AVG | SEM |
| MLP | ✗ | 100.00 | 0.00 | 100.00 | 0.00 | 55.98 | 6.32 | 21.32 | 2.93 |
| ResNet-18 | ✗ | 100.00 | 0.00 | 100.00 | 0.00 | 89.47 | 4.74 | 90.03 | 2.74 |
| ResNet-34 | ✗ | 100.00 | 0.00 | 100.00 | 0.00 | 85.58 | 4.88 | 80.33 | 4.16 |
| ResNet-50 | ✗ | 100.00 | 0.00 | 100.00 | 0.00 | 83.76 | 5.52 | 76.47 | 4.79 |
| ResNet-101 | ✗ | 99.48 | 0.52 | 100.00 | 0.00 | 79.69 | 5.82 | 70.87 | 5.45 |
| ResNet-101 | ✓ | 100.00 | 0.00 | 100.00 | 0.00 | 72.75 | 5.92 | 68.51 | 4.13 |
| ResNet-152 | ✗ | 100.00 | 0.00 | 100.00 | 0.00 | 69.75 | 6.38 | 68.99 | 5.02 |
| ResNet-152 | ✓ | 100.00 | 0.00 | 100.00 | 0.00 | 77.18 | 5.13 | 69.13 | 4.49 |
| DenseNet-121 | ✗ | 99.98 | 0.02 | 100.00 | 0.00 | 89.22 | 4.06 | 88.56 | 2.94 |
| DenseNet-121 | ✓ | 74.48 | 2.83 | 74.21 | 2.84 | 35.48 | 4.45 | 40.46 | 2.64 |
| DenseNet-161 | ✗ | 100.00 | 0.00 | 100.00 | 0.00 | 86.50 | 4.89 | 87.19 | 3.57 |
| DenseNet-201 | ✗ | 100.00 | 0.00 | 100.00 | 0.00 | 86.57 | 4.89 | 85.42 | 4.21 |
| ConvNeXt-small | ✗ | 99.84 | 0.16 | 100.00 | 0.00 | 92.61 | 3.59 | 91.17 | 4.17 |
| ConvNeXt-base | ✗ | 99.99 | 0.00 | 100.00 | 0.00 | 92.71 | 3.37 | 93.82 | 3.08 |
| WideResNet | ✗ | 100.00 | 0.00 | 100.00 | 0.00 | 84.54 | 4.43 | 81.10 | 3.72 |
| ViT | ✗ | 99.99 | 0.00 | 100.00 | 0.00 | 87.69 | 2.79 | 74.69 | 3.66 |
| Swin-tiny | ✗ | 100.00 | 0.00 | 100.00 | 0.00 | 91.17 | 4.25 | 89.00 | 4.68 |
| Swin-base | ✗ | 99.99 | 0.00 | 100.00 | 0.00 | 91.90 | 3.92 | 89.02 | 4.76 |
| ED | ✗ | 100.00 | 0.00 | 100.00 | 0.00 | **97.95** | 2.05 | **96.92** | 1.77 |

Table 6: Pairwise evaluation on Shapes3D.

**MPI3D** Overall, the results on the MPI3D shown in Table 7 align broadly with those obtained on the other datasets. The explicit disentanglement model (ED) achieves the best test accuracy, with a margin of 14.4% from the closest monolithic model. The performance among monolithic models is heterogeneous. The DenseNet-161 is the best-performing monolithic model (50.28%), closely followed by the DenseNet-121 (46.58%) and the ResNet-18 (45.53%). ConvNeXts are less competitive, achieving a best accuracy of only 41.74%, while visual transformers show decisively inferior performances on this task (both Swin Transformers and ViT). All models, except for the pre-trained DenseNet-121, converge and score perfect accuracy in-distribution.

| Model | Pretrained | Train | | ID Val | | OOD Val | | Test | |
|---|---|---|---|---|---|---|---|---|---|
| | | AVG | SEM | AVG | SEM | AVG | SEM | AVG | SEM |
| MLP | ✗ | 95.69 | 0.65 | 96.27 | 0.60 | 0.50 | 0.17 | 3.27 | 1.66 |
| ResNet-18 | ✗ | 99.99 | 0.00 | 99.99 | 0.00 | 30.65 | 5.66 | 45.53 | 5.76 |
| ResNet-34 | ✗ | 99.98 | 0.00 | 99.99 | 0.00 | 24.62 | 4.87 | 36.55 | 5.48 |
| ResNet-50 | ✗ | 99.99 | 0.00 | 99.99 | 0.00 | 24.40 | 5.06 | 41.40 | 5.66 |
| ResNet-101 | ✗ | 99.98 | 0.00 | 99.97 | 0.01 | 19.35 | 4.47 | 35.03 | 5.39 |
| ResNet-101 | ✓ | 99.95 | 0.01 | 99.93 | 0.01 | 18.71 | 4.57 | 16.35 | 3.02 |
| ResNet-152 | ✗ | 99.96 | 0.01 | 99.95 | 0.02 | 24.61 | 5.22 | 34.90 | 5.59 |
| ResNet-152 | ✓ | 99.94 | 0.01 | 99.92 | 0.02 | 17.71 | 4.44 | 18.31 | 3.80 |
| DenseNet-121 | ✗ | 99.99 | 0.00 | 99.99 | 0.00 | 27.76 | 6.06 | 46.58 | 6.33 |
| DenseNet-121 | ✓ | 78.79 | 2.86 | 78.83 | 2.86 | 27.35 | 4.48 | 25.92 | 2.66 |
| DenseNet-161 | ✗ | 99.99 | 0.00 | 100.00 | 0.00 | 30.95 | 6.45 | 50.28 | 6.71 |
| DenseNet-201 | ✗ | 99.99 | 0.00 | 99.99 | 0.00 | 29.67 | 6.18 | 45.44 | 6.44 |
| ConvNeXt-small | ✗ | 99.92 | 0.01 | 99.96 | 0.01 | 30.65 | 6.14 | 41.74 | 6.16 |
| ConvNeXt-base | ✗ | 98.89 | 0.72 | 99.57 | 0.27 | 30.67 | 6.16 | 39.66 | 6.31 |
| WideResNet | ✗ | 99.98 | 0.00 | 99.99 | 0.00 | 26.52 | 5.38 | 41.24 | 5.92 |
| ViT | ✗ | 98.39 | 0.23 | 98.76 | 0.20 | 20.01 | 4.77 | 24.25 | 4.98 |
| Swin-tiny | ✗ | 99.87 | 0.03 | 99.95 | 0.01 | 24.11 | 5.97 | 30.97 | 5.29 |
| Swin-base | ✗ | 99.76 | 0.04 | 99.92 | 0.01 | 24.37 | 5.87 | 26.32 | 4.97 |
| ED | ✗ | 99.98 | 0.00 | 99.99 | 0.00 | **50.66** | 4.27 | **64.68** | 4.24 |

Table 7: Pairwise evaluation on MPI3D.

## C.3 Attribute-level experimental results

In this section, we re-elaborate the results reported in Section C using a different granularity. Rather than providing an overview of the results at the dataset level, we break down the data to the attributes combination level to provide additional insights on the generalization performances of the single attributes. To reduce the clutter and increase interpretability, we group the results by model's architecture, plotting the mean test accuracy over the different random seeds, model sizes, and pre-training. Figure 19 include these new attribute-level visualizations for each one of the studied datasets. Different insights can be derived looking at the data from this perspective.

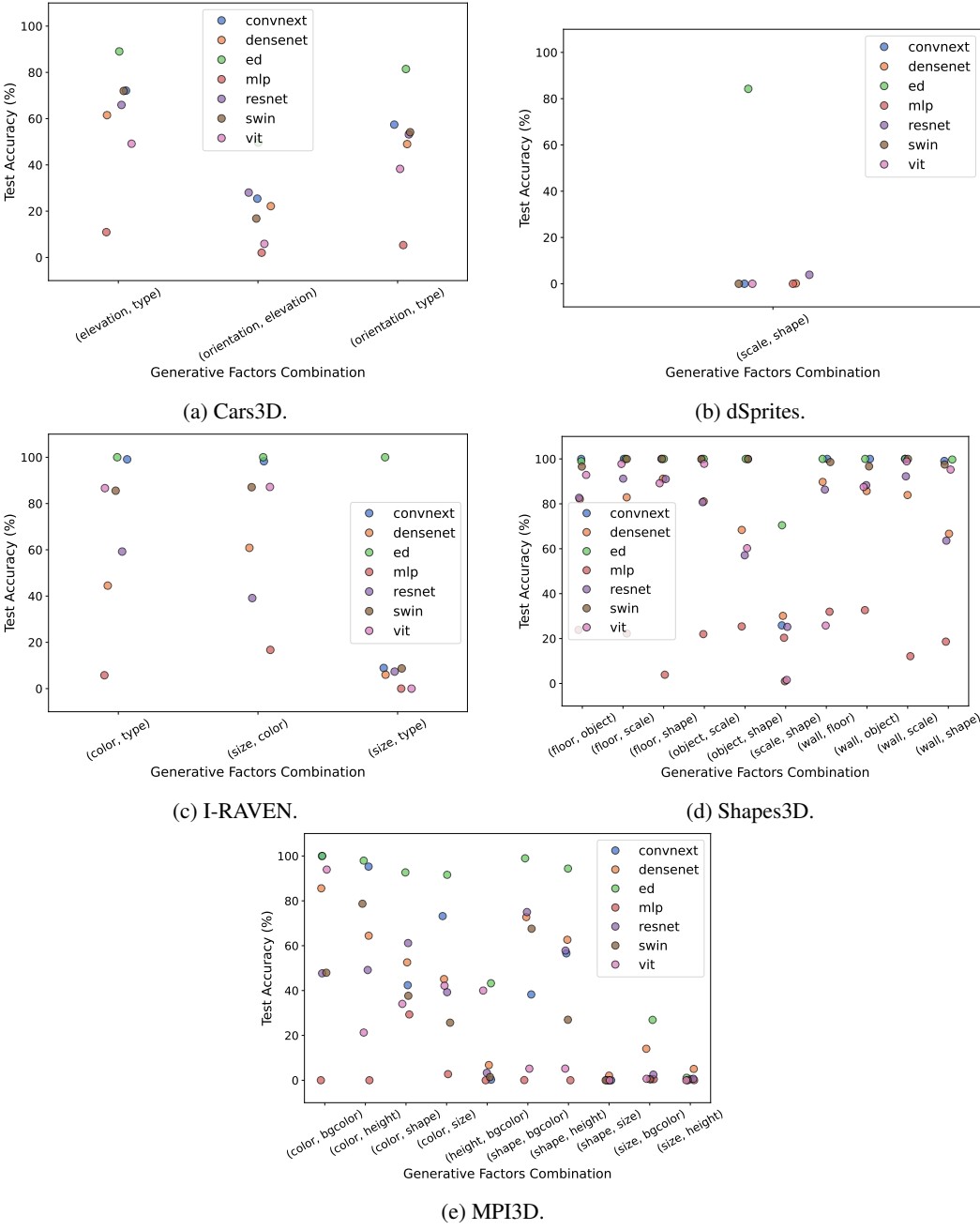

Figure 19: Pair-wise compositional evaluation with attribute-combination granularity.

Firstly, we can observe that *some combinations of generative factors* (e.g., size-height in MPI3D or size-type in I-RAVEN) *are significantly more challenging than others* (e.g., object-scale and wall-scale in Shapes3D). In particular, the failure observed empirically is consistent across the majority of the models investigated, indicating that the difficulty in generalizing compositionally on them could be independent from the specific architectures and training techniques used in this work.

Secondly, it stands out that *some combinations of generative factors are consistently challenging across different datasets*. This is the case, for example, for the shape-size combination, for which almost no model architecture can properly generalize across all the datasets that have it (dSprites, MPI3D, I-RAVEN, and dSprites). Given its consistency across multiple models and datasets, we speculate that this specific combination of generative factors could be intrinsically harder to classify, possibly to the causal interactions between the factors as proposed by Montero et al. [8].

Finally, we can observed that the "strong baseline" considered in this work, the Explicit Disentanglement model, is consistently better than the other models almost on every combination of attributes. The conclusion that can be drawn from this observation is that disentangling the forward and backward passes in the generative factor prediction uniformly and consistently increases the performances on all the investigated generative factors, rather than only improving them on a small subset of them.

## C.4   Impact of learnable activation functions on compositional generalization

As part of the investigation, we also ablate the impact of programmable activation functions on the model compositional generalization. This ablation stems as an attempt to translate into practice some of the receipts hinted by Ren et al. [10], in particular the necessity to enforce the composition of the generation function $\mathcal{G}_x : (\mathbf{F}, \epsilon_x) \mapsto \mathbb{R}^D$ and the feature extraction function $h : \mathbb{R}^D \mapsto \mathbb{R}^S$ to maintain an isomorphism between the generative factors space $\mathbf{G}$ and the learned representation space $\mathbf{Z}$. One possible idea to implement this is to make the feature extraction function as close as possible to a linear map, hence enforcing quasi-isomorphism on the part of the composite function on which we have control ($h$). In practice, we try to replace the activation functions in the building blocks of different convolutional networks, namely ResNets, DenseNets, and ConvNeXt, with the programmable activation function PReLU,

$$\text{PReLU}(x) = \max(0, x) + a \min(0, x),$$

where $a$ is a learnable vector (one parameter for every input dimension). We hypothesize that the use of this activation function could represent an additional degree of freedom for the model, which could choose during training between 1) more expressiveness, given by an hard non-linearity such as ReLU or 2) a "softer" non-linearity that matches more closely the behavior of a linear activation function.

We empirically validate this hypothesis on a subset of the dataset: Cars3D, I-RAVEN, Shapes3D, and MPI3D. Figure 20 includes a high-level overview of the results. The plot shows the test accuracy of different model families, where the standard error of the mean (SEM) is reported considering as population the different model sizes and random seeds. In the legend, "Standard" and "PReLU" refer to the vanilla and the custom implementations of the architectures, respectively. Overall, we can observe that programmable activation functions are consistently better than the original activation function (ReLU) for ResNets. On the other hand, the behavior is not consistent for DenseNets (that also use ReLU in their stacked basic blocks) and seem to be consistently worse in the case of ConvNeXt (where, on the other hand, the default activation function is GeLU).

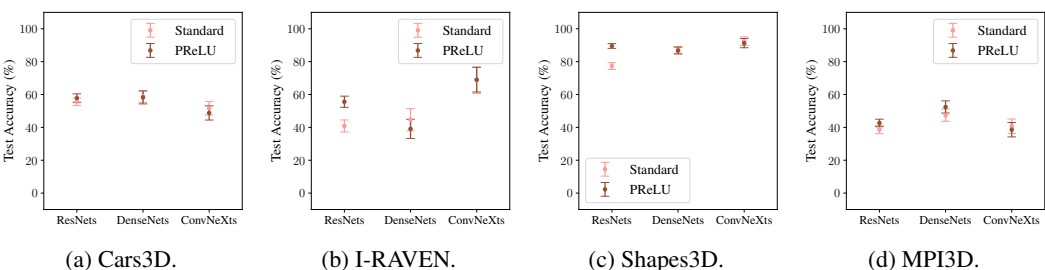

(a) Cars3D.          (b) I-RAVEN.          (c) Shapes3D.          (d) MPI3D.

Figure 20: Ablation on activation functions.

To gain a more complete understanding of the effects of using programmable activation functions, we break down the results for different models in Table 8, showing the test accuracy difference between their PReLU and vanilla versions. We can make different observations based on these results. Firstly, we can see that the main trend observed on the high-level visualization translates consistently to the individual models. The performance of both ConvNeXt models drops when GeLU functions are replaced with PReLUs, while for DenseNets the trend is not consistent. On the other hand, ResNets greatly benefit from these alternative activation functions. We also observe that some models (e.g., ResNet-18 and ResNet-50) greatly improve their test scores when PReLU activation functions are used, showing impressive two-digit gains on the compositional generalization split (+20.86% and +31.52%, respectively). In Figure 21, we also report the distribution of the $a$ parameters for different trained models, which seems to remain close to the initialization value (0.5) in the majority of the models.

In conclusion, while programmable activation functions yield notable improvements in compositional generalization for certain models and tasks—particularly in the case of ResNets—the current evidence does not support the conclusion that they offer a comprehensive panacea to the limitations of convolutional neural networks in compositional generalization. Moreover, their performance does not consistently surpass that of other widely used activation functions, such as GeLU in contemporary vision architectures.

| Model | I-RAVEN | Cars3D | Shapes3D | MPI3D | $\mu$ |
|---|---|---|---|---|---|
| ResNet-18 | 20.86 | 2.93 | 5.02 | 8.64 | 9.36 |
| ResNet-34 | 9.63 | 4.41 | 11.45 | 11.04 | 9.13 |
| ResNet-50 | 31.52 | 1.77 | 13.61 | 5.01 | 12.98 |
| ResNet-101 | 9.15 | -3.01 | 15.64 | -1.93 | 4.96 |
| ResNet-152 | 2.69 | 5.02 | 15.28 | -6.49 | 4.13 |
| DenseNet-121 | -21.04 | -0.72 | -4.01 | 6.47 | -4.82 |
| DenseNet-161 | -6.29 | 1.46 | 3.18 | 1.26 | -0.10 |
| DenseNet-201 | 10.27 | -2.87 | -0.12 | 7.21 | 3.62 |
| ConvNeXt-small | -3.34 | -0.41 | 0.54 | -4.30 | -1.88 |
| ConvNeXt-base | 3.85 | -5.78 | -2.99 | 0.06 | -1.22 |

Table 8: Test accuracy difference between PReLu and standard models on different datasets.

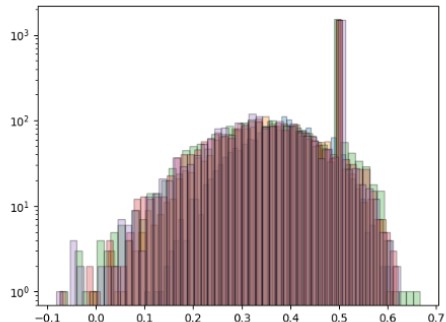

Figure 21: $a$ values distribution.

## C.5 Grokking in compositional generalization for visual representation learning

*Grokking* is a phenomenon initially observed in the context of neural networks trained on algorithmic datasets, for which a generalizing solution was learned only with a considerable delay (in terms of training iterations) compared to a solution that could overfit the training data [27]. While this phenomenon is not common in computer vision tasks, Liu et al. [28] observed that it is possible to emulate grokking behaviors using specific initialization for neural networks' weights, e.g., initializing them with large magnitudes. In the same work, the authors also speculate that the emergence of grokking might be more easily observable when the generalization heavily relies on learning good representations from scratch. Compositional generalization is, by nature, a task that heavily relies on this category of tasks. Hence, we design a control experiment to verify whether training with a sensibly larger number of update steps (up to ×240) can lead the model to learn better, more compositional representations. We restrict the scope of the empirical evaluation to three datasets (dSprites, I-RAVEN, and Cars3D). Since the aim is limited to investigating whether a grokking behavior can emerge in any setting, we resort to the pairwise evaluation scheme, selecting only a few of the most significant generative factor pairs for each dataset. We experiment with a single

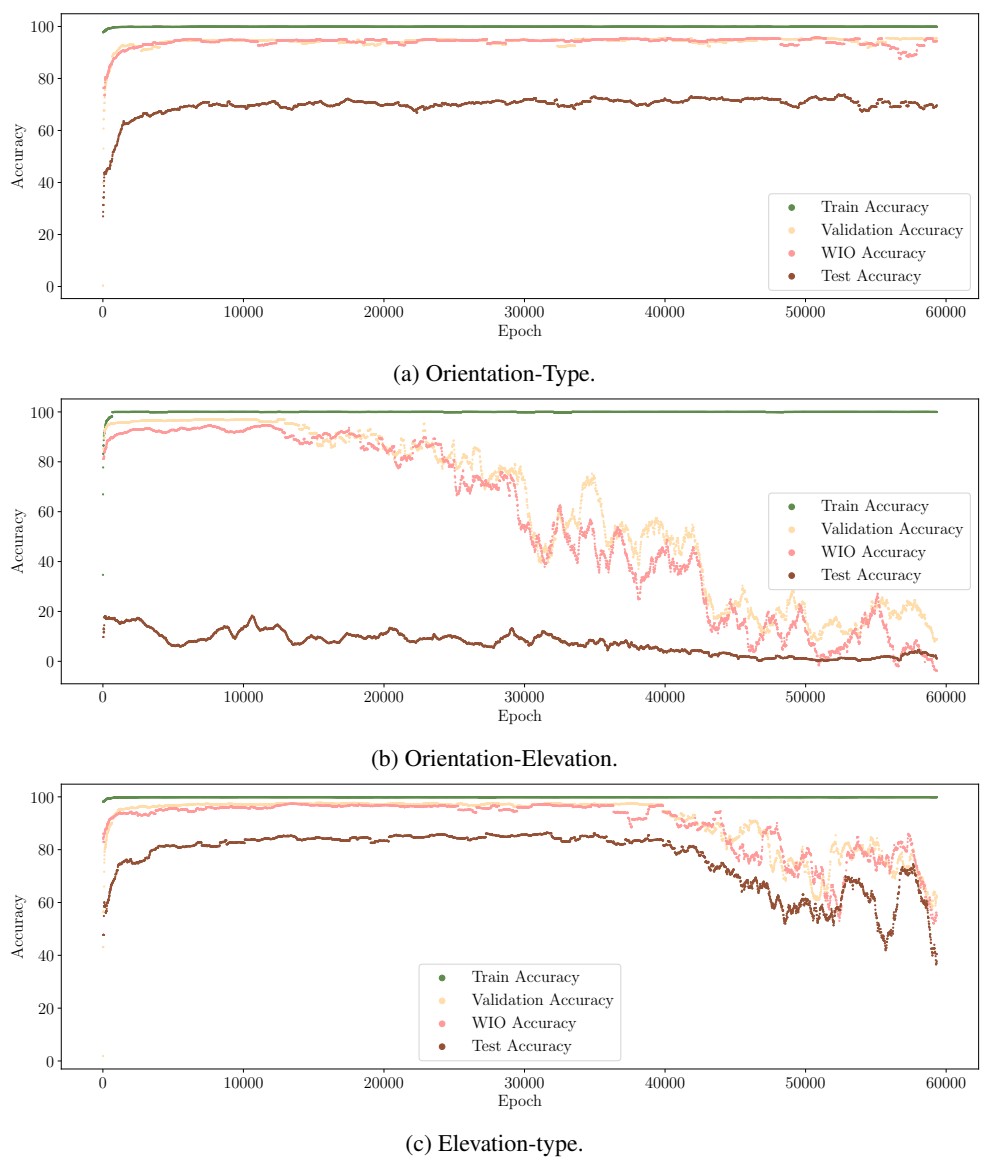

(a) Orientation-Type.

(b) Orientation-Elevation.

(c) Elevation-type.

Figure 22: Grokking experiments on Cars3D.

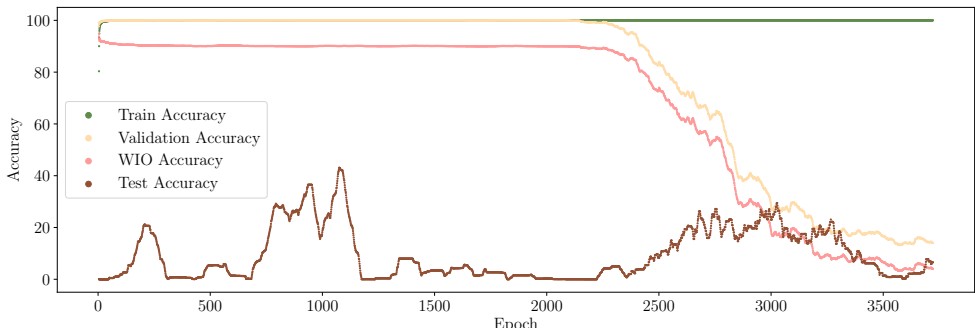

Figure 23: Grokking experiments on dSprites.

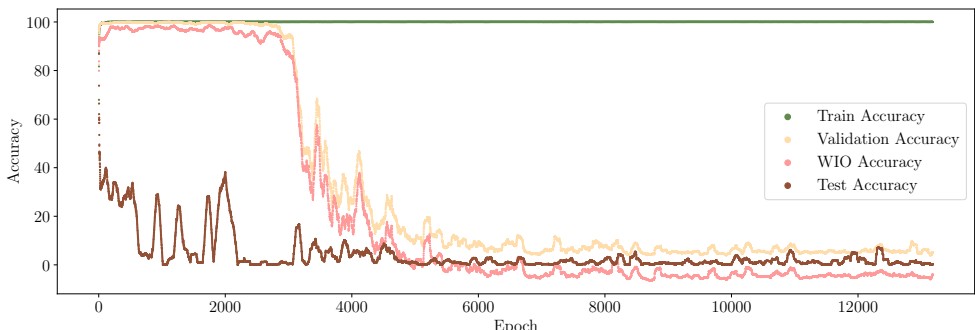

Figure 24: Grokking experiments on I-RAVEN.

architecture, a ResNet-101 trained from scratch, chosen because of its good performance on the pair-wise evaluation experiments and its overparametrized regime for the given datasets ($43'548'619$ parameters). We also adjust some of the hyperparameters of the network, using a higher weight decay (0.1) and a lower learning rate (0.0001) during training, and use two different random seeds (reporting for each step the best accuracy achieved among the two seeds).

The results for the different attribute combinations considered in the evaluation for the Cars3D dataset (orientation-type, orientation-elevation, and elevation-type) are shown in Figure 22. In this dataset, the number of epochs was increased from 250 to 60k. The training curves clearly show that using a larger number of update steps, the networks converge to stable, imperfect representations in the best scenario (e.g., the models trained on the orientation-type combination) and irremediably diverge even in-distribution in the worst cases (e.g., for the orientation-elevation and elevation-type combinations). Naturally, the training accuracy always converged to $100\%$, signaling that the model converged to a solution that was purely memorizing all the training samples. The same behavior was also confirmed for the dSprites and I-RAVEN datasets, in Figures 23 and 24 respectively, for which the extension of the number of training epochs only resulted in a clear divergence of the models.

Overall, these experiments allow us to exclude that the lack of compositionality in the learned representations is due to an insufficient training of the models. Furthermore, these results also hint that grokking phenomena are unlikely to be observed in the context of compositional generalization, despite it being a task where learning well-structured representations is of utmost importance.

# D    Extended orthotopic evaluation results

## D.1    In-domain comparison between pair-wise and orthotopic

Figure 25 reports the model-wise difference between validation accuracy obtained in the orthotopic and pair-wise evaluation frameworks. On average, we observe that the two are quite comparable (1.09% average difference).

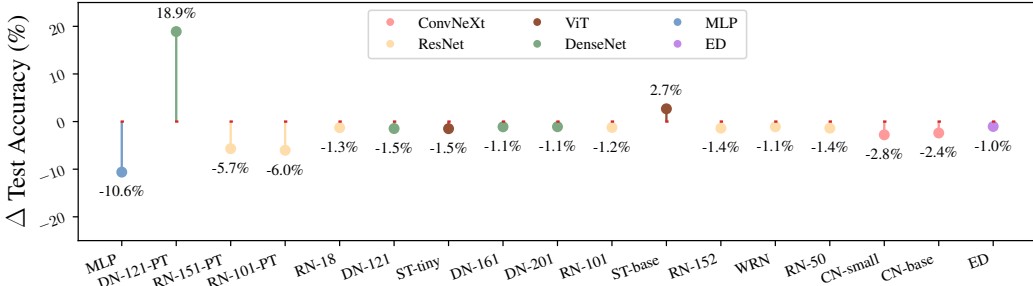

Figure 25: In-distribution (validation) delta accuracy between orthotopic and pair-wise evaluation.

# E    Additional experiments on Noisy-MPI3d-Real

We design and run an additional experimental validation on the proposed orthotopic evaluation and ladder of compositional generalization to test their robustness under more realistic conditions (noise on the labels, unlabeled generative factors). We modify one of the experiments (on MPI3d-real) by perturbing 2 attribute labels at random in 10% of the training samples to simulate noisy annotations (i.e., re-introducing $\epsilon_y$ in the experimental setup). Additionally, we also removed the annotations of two task-relevant factors (color, x-position) from the training data, effectively morphing them from task-relevant ($G$) to task-irrelevant ($O$) generative factors. With these modifications, we aim to achieve a more realistic, yet fully controllable, setting with unknown latent generative factors and random label noise, as would be the case in a real-world dataset.

The results of these experiments are reported in Table 9.

| Model | c=0 | c=1 | c=2 | c=3 |
|---|---|---|---|---|
| Convnext-small | 0.0 | 32.9 | 40.1 | 46.9 |
| ResNet50 | 0.0 | 38.1 | 57.8 | 75.2 |
| Swin-tiny | 0.0 | 29.6 | 45.6 | 49.0 |
| AIN | 0.0 | 46.8 | 67.6 | 83.7 |
| ED | 0.0 | 71.1 | 67.6 | 83.2 |

Table 9: Performance of models across different values of $c$

We can observe that, even in this noisy setting closer to real-world scenarios our main results hold: the relation between compositional similarity and evaluation difficulty, the superiority of architectures that explicitly incorporate some inductive bias toward compositionality in their architecture (e.g., ED and AIN), and the strict increase in generalization accuracy achieved by ED and AINs compared to monolithic architectures.

## E.1 Extended experimental results

### E.1.1 dSprites

| Model | Pretrained | Train AVG | Train SEM | ID Val AVG | ID Val SEM | OOD Val AVG | OOD Val SEM | Test AVG | Test SEM |
|---|---|---|---|---|---|---|---|---|---|
| MLP | ✗ | 88.15 | 8.68 | 90.76 | 6.98 | 66.67 | 33.33 | 0.00 | 0.00 |
| ResNet-18 | ✗ | 100.00 | 0.00 | 100.00 | 0.00 | 100.00 | 0.00 | 0.00 | 0.00 |
| ResNet-50 | ✗ | 99.83 | 0.10 | 99.99 | 0.00 | 100.00 | 0.00 | 0.00 | 0.00 |
| ResNet-101 | ✗ | 99.84 | 0.04 | 100.00 | 0.00 | 100.00 | 0.00 | 0.00 | 0.00 |
| ResNet-101 | ✓ | 99.98 | 0.00 | 100.00 | 0.00 | 100.00 | 0.00 | 0.00 | 0.00 |
| ResNet-152 | ✗ | 99.78 | 0.17 | 100.00 | 0.00 | 100.00 | 0.00 | 0.00 | 0.00 |
| ResNet-152 | ✓ | 99.97 | 0.02 | 100.00 | 0.00 | 100.00 | 0.00 | 0.00 | 0.00 |
| DenseNet-121 | ✗ | 100.00 | 0.00 | 100.00 | 0.00 | 100.00 | 0.00 | 0.00 | 0.00 |
| DenseNet-121 | ✓ | 100.00 | 0.00 | 100.00 | 0.00 | 100.00 | 0.00 | 0.00 | 0.00 |
| DenseNet-161 | ✗ | 100.00 | 0.00 | 100.00 | 0.00 | 100.00 | 0.00 | 0.00 | 0.00 |
| DenseNet-201 | ✗ | 100.00 | 0.00 | 100.00 | 0.00 | 100.00 | 0.00 | 0.00 | 0.00 |
| ConvNeXt-tiny | ✗ | 99.88 | 0.08 | 100.00 | 0.00 | 100.00 | 0.00 | 0.00 | 0.00 |
| ConvNeXt-small | ✗ | 99.96 | 0.02 | 100.00 | 0.00 | 100.00 | 0.00 | 0.00 | 0.00 |
| ConvNeXt-base | ✗ | 99.98 | 0.01 | 100.00 | 0.00 | 100.00 | 0.00 | 0.00 | 0.00 |
| WideResNet | ✗ | 99.81 | 0.10 | 100.00 | 0.00 | 100.00 | 0.00 | 0.00 | 0.00 |
| Swin-tiny | ✗ | 99.89 | 0.01 | 100.00 | 0.00 | 100.00 | 0.00 | 0.00 | 0.00 |
| Swin-base | ✗ | 99.90 | 0.03 | 100.00 | 0.00 | 100.00 | 0.00 | 0.00 | 0.00 |
| ED | ✗ | 100.00 | 0.00 | 100.00 | 0.00 | 100.00 | 0.00 | 0.00 | 0.00 |

Table 10: $c = 0$

| Model | Pretrained | Train AVG | Train SEM | ID Val AVG | ID Val SEM | OOD Val AVG | OOD Val SEM | Test AVG | Test SEM |
|---|---|---|---|---|---|---|---|---|---|
| MLP | ✗ | 86.88 | 1.16 | 87.23 | 0.41 | 98.33 | 1.67 | 2.03 | 0.77 |
| ResNet-18 | ✗ | 99.98 | 0.02 | 100.00 | 0.00 | 100.00 | 0.00 | 21.43 | 0.63 |
| ResNet-50 | ✗ | 99.75 | 0.10 | 99.95 | 0.03 | 100.00 | 0.00 | 24.69 | 2.35 |
| ResNet-101 | ✗ | 99.03 | 0.89 | 99.99 | 0.01 | 100.00 | 0.00 | 21.30 | 3.61 |
| ResNet-101 | ✓ | 99.93 | 0.01 | 99.96 | 0.02 | 100.00 | 0.00 | 8.59 | 0.77 |
| ResNet-152 | ✗ | 99.57 | 0.29 | 99.99 | 0.01 | 100.00 | 0.00 | 28.35 | 0.92 |
| ResNet-152 | ✓ | 99.87 | 0.06 | 99.93 | 0.01 | 100.00 | 0.00 | 9.40 | 1.52 |
| DenseNet-121 | ✗ | 100.00 | 0.00 | 100.00 | 0.00 | 100.00 | 0.00 | 56.13 | 13.45 |
| DenseNet-121 | ✓ | 100.00 | 0.00 | 99.98 | 0.00 | 100.00 | 0.00 | 17.87 | 0.63 |
| DenseNet-161 | ✗ | 99.99 | 0.00 | 100.00 | 0.00 | 100.00 | 0.00 | 36.13 | 1.92 |
| DenseNet-201 | ✗ | 100.00 | 0.00 | 100.00 | 0.00 | 100.00 | 0.00 | 36.60 | 1.21 |
| ConvNeXt-tiny | ✗ | 99.92 | 0.01 | 100.00 | 0.00 | 100.00 | 0.00 | 20.32 | 3.22 |
| ConvNeXt-small | ✗ | 99.95 | 0.02 | 100.00 | 0.00 | 100.00 | 0.00 | 33.79 | 1.16 |
| ConvNeXt-base | ✗ | 99.93 | 0.02 | 100.00 | 0.00 | 100.00 | 0.00 | 35.20 | 1.31 |
| WideResNet | ✗ | 99.76 | 0.11 | 99.98 | 0.01 | 100.00 | 0.00 | 22.90 | 4.42 |
| Swin-tiny | ✗ | 99.44 | 0.21 | 99.95 | 0.01 | 100.00 | 0.00 | 18.99 | 1.60 |
| Swin-base | ✗ | 99.53 | 0.08 | 99.97 | 0.02 | 100.00 | 0.00 | 37.73 | 1.91 |
| ED | ✗ | 100.00 | 0.00 | 100.00 | 0.00 | 100.00 | 0.00 | 61.31 | 0.61 |

Table 11: $c = 1$

| Model | Pretrained | Train | | ID Val | | OOD Val | | Test | |
|---|---|---|---|---|---|---|---|---|---|
| | | AVG | SEM | AVG | SEM | AVG | SEM | AVG | SEM |
| ConvNeXt-Base | ✗ | 99.95 | 0.04 | 100.00 | 0.00 | 100.00 | 0.00 | 98.83 | 0.89 |
| ConvNeXt-Small | ✗ | 99.87 | 0.09 | 100.00 | 0.00 | 100.00 | 0.00 | 98.71 | 1.18 |
| ConvNeXt-Tiny | ✗ | 99.91 | 0.07 | 100.00 | 0.00 | 100.00 | 0.00 | 94.31 | 5.39 |
| DenseNet-121 | ✗ | 100.00 | 0.00 | 100.00 | 0.00 | 100.00 | 0.00 | 99.85 | 0.13 |
| DenseNet-121 | ✓ | 100.00 | 0.00 | 100.00 | 0.00 | 100.00 | 0.00 | 95.72 | 1.33 |
| DenseNet-161 | ✗ | 100.00 | 0.00 | 100.00 | 0.00 | 100.00 | 0.00 | 99.25 | 0.29 |
| DenseNet-201 | ✗ | 100.00 | 0.00 | 100.00 | 0.00 | 100.00 | 0.00 | 99.34 | 0.45 |
| ED | ✗ | 100.00 | 0.00 | 100.00 | 0.00 | 100.00 | 0.00 | 99.57 | 0.19 |
| MLP | ✗ | 69.74 | 1.50 | 69.32 | 1.69 | 97.50 | 2.50 | 5.45 | 0.77 |
| ResNet-101 | ✗ | 99.79 | 0.08 | 99.99 | 0.00 | 100.00 | 0.00 | 96.99 | 0.94 |
| ResNet-101 | ✓ | 99.90 | 0.01 | 99.97 | 0.01 | 100.00 | 0.00 | 89.04 | 1.65 |
| ResNet-152 | ✗ | 99.82 | 0.08 | 99.99 | 0.01 | 99.17 | 0.83 | 95.24 | 0.21 |
| ResNet-152 | ✓ | 99.82 | 0.05 | 99.98 | 0.00 | 100.00 | 0.00 | 80.98 | 5.57 |
| ResNet-18 | ✗ | 100.00 | 0.00 | 100.00 | 0.00 | 100.00 | 0.00 | 86.68 | 2.75 |
| ResNet-50 | ✗ | 99.92 | 0.01 | 99.98 | 0.01 | 100.00 | 0.00 | 88.27 | 0.78 |
| Swin-Base | ✗ | 99.63 | 0.06 | 99.98 | 0.01 | 100.00 | 0.00 | 94.90 | 1.19 |
| Swin-Tiny | ✗ | 99.56 | 0.18 | 99.98 | 0.00 | 100.00 | 0.00 | 79.13 | 1.13 |
| WideResNet | ✗ | 99.87 | 0.04 | 99.99 | 0.00 | 100.00 | 0.00 | 94.13 | 3.00 |

Table 12: $c = 2$

| Model | Pretrained | Train | | ID Val | | OOD Val | | Test | |
|---|---|---|---|---|---|---|---|---|---|
| | | AVG | SEM | AVG | SEM | AVG | SEM | AVG | SEM |
| ConvNeXt-Base | ✗ | 99.97 | 0.01 | 100.00 | 0.00 | 100.00 | 0.00 | 99.89 | 0.07 |
| ConvNeXt-Small | ✗ | 99.93 | 0.01 | 100.00 | 0.00 | 100.00 | 0.00 | 99.85 | 0.08 |
| ConvNeXt-Tiny | ✗ | 99.96 | 0.03 | 100.00 | 0.00 | 100.00 | 0.00 | 99.43 | 0.39 |
| DenseNet-121 | ✗ | 100.00 | 0.00 | 100.00 | 0.00 | 100.00 | 0.00 | 100.00 | 0.00 |
| DenseNet-121 | ✓ | 100.00 | 0.00 | 100.00 | 0.00 | 100.00 | 0.00 | 96.88 | 1.12 |
| DenseNet-161 | ✗ | 100.00 | 0.00 | 100.00 | 0.00 | 100.00 | 0.00 | 100.00 | 0.00 |
| DenseNet-201 | ✗ | 100.00 | 0.00 | 100.00 | 0.00 | 100.00 | 0.00 | 99.99 | 0.01 |
| ED | ✗ | 99.99 | 0.00 | 100.00 | 0.00 | 100.00 | 0.00 | 97.17 | 1.64 |
| MLP | ✗ | 80.07 | 0.78 | 78.07 | 0.36 | 97.50 | 2.50 | 0.74 | 0.07 |
| ResNet-101 | ✗ | 99.91 | 0.03 | 99.99 | 0.01 | 100.00 | 0.00 | 95.31 | 1.47 |
| ResNet-101 | ✓ | 99.70 | 0.04 | 99.97 | 0.00 | 100.00 | 0.00 | 63.09 | 7.89 |
| ResNet-152 | ✗ | 99.84 | 0.02 | 99.97 | 0.00 | 100.00 | 0.00 | 89.69 | 3.37 |
| ResNet-152 | ✓ | 99.72 | 0.02 | 99.96 | 0.01 | 100.00 | 0.00 | 68.94 | 6.08 |
| ResNet-18 | ✗ | 99.98 | 0.02 | 99.99 | 0.01 | 100.00 | 0.00 | 97.87 | 0.84 |
| ResNet-50 | ✗ | 99.92 | 0.04 | 99.99 | 0.00 | 100.00 | 0.00 | 84.84 | 0.40 |
| Swin-Base | ✗ | 99.69 | 0.04 | 99.97 | 0.01 | 100.00 | 0.00 | 60.37 | 2.11 |
| Swin-Tiny | ✗ | 99.77 | 0.02 | 99.97 | 0.00 | 100.00 | 0.00 | 50.13 | 1.49 |
| WideResNet | ✗ | 99.61 | 0.08 | 99.99 | 0.00 | 100.00 | 0.00 | 82.72 | 6.63 |

Table 13: $c = 3$

### E.1.2  I-RAVEN

| Model | Pretrained | Train | | ID Val | | OOD Val | | Test | |
|---|---|---|---|---|---|---|---|---|---|
| | | AVG | SEM | AVG | SEM | AVG | SEM | AVG | SEM |
| ConvNeXt-Base | ✗ | 99.68 | 0.24 | 100.00 | 0.00 | 100.00 | 0.00 | 0.00 | 0.00 |
| ConvNeXt-Small | ✗ | 99.39 | 0.20 | 100.00 | 0.00 | 100.00 | 0.00 | 0.00 | 0.00 |
| ConvNeXt-Tiny | ✗ | 99.84 | 0.06 | 100.00 | 0.00 | 83.45 | 16.55 | 0.00 | 0.00 |
| DenseNet-121 | ✗ | 100.00 | 0.00 | 100.00 | 0.00 | 100.00 | 0.00 | 0.00 | 0.00 |
| DenseNet-121 | ✓ | 94.08 | 5.90 | 95.36 | 4.64 | 92.17 | 7.83 | 0.00 | 0.00 |
| DenseNet-161 | ✗ | 100.00 | 0.00 | 100.00 | 0.00 | 100.00 | 0.00 | 0.00 | 0.00 |
| DenseNet-201 | ✗ | 99.99 | 0.00 | 100.00 | 0.00 | 100.00 | 0.00 | 0.00 | 0.00 |
| ED | ✗ | 99.99 | 0.01 | 100.00 | 0.00 | 100.00 | 0.00 | 0.00 | 0.00 |
| MLP | ✗ | 91.72 | 1.82 | 97.74 | 0.62 | 55.36 | 29.36 | 0.00 | 0.00 |
| ResNet-101 | ✗ | 99.96 | 0.03 | 100.00 | 0.00 | 100.00 | 0.00 | 0.00 | 0.00 |
| ResNet-101 | ✓ | 98.62 | 1.13 | 95.88 | 4.08 | 79.12 | 14.87 | 0.00 | 0.00 |
| ResNet-152 | ✗ | 99.98 | 0.01 | 100.00 | 0.00 | 100.00 | 0.00 | 0.00 | 0.00 |
| ResNet-152 | ✓ | 97.78 | 2.20 | 93.39 | 6.61 | 75.39 | 14.24 | 0.00 | 0.00 |
| ResNet-18 | ✗ | 99.98 | 0.00 | 100.00 | 0.00 | 100.00 | 0.00 | 0.00 | 0.00 |
| ResNet-50 | ✗ | 99.83 | 0.08 | 100.00 | 0.00 | 100.00 | 0.00 | 0.00 | 0.00 |
| Swin-Base | ✗ | 99.05 | 0.33 | 99.97 | 0.01 | 84.52 | 15.48 | 0.00 | 0.00 |
| Swin-Tiny | ✗ | 99.50 | 0.08 | 99.94 | 0.02 | 100.00 | 0.00 | 0.00 | 0.00 |
| WideResNet | ✗ | 99.95 | 0.02 | 100.00 | 0.00 | 100.00 | 0.00 | 0.00 | 0.00 |

Table 14: $c = 0$

| Model | Pretrained | Train | | ID Val | | OOD Val | | Test | |
|---|---|---|---|---|---|---|---|---|---|
| | | AVG | SEM | AVG | SEM | AVG | SEM | AVG | SEM |
| ConvNeXt-Base | ✗ | 99.39 | 0.31 | 100.00 | 0.00 | 100.00 | 0.00 | 42.40 | 4.81 |
| ConvNeXt-Small | ✗ | 99.60 | 0.17 | 100.00 | 0.00 | 100.00 | 0.00 | 44.86 | 2.22 |
| ConvNeXt-Tiny | ✗ | 99.15 | 0.53 | 100.00 | 0.00 | 100.00 | 0.00 | 37.36 | 2.89 |
| DenseNet-121 | ✗ | 100.00 | 0.00 | 100.00 | 0.00 | 100.00 | 0.00 | 3.76 | 1.69 |
| DenseNet-121 | ✓ | 99.94 | 0.01 | 100.00 | 0.00 | 100.00 | 0.00 | 2.30 | 0.84 |
| DenseNet-161 | ✗ | 99.99 | 0.00 | 100.00 | 0.00 | 100.00 | 0.00 | 6.79 | 0.85 |
| DenseNet-201 | ✗ | 100.00 | 0.00 | 100.00 | 0.00 | 100.00 | 0.00 | 5.94 | 0.48 |
| ED | ✗ | 100.00 | 0.00 | 100.00 | 0.00 | 100.00 | 0.00 | 76.12 | 2.70 |
| MLP | ✗ | 95.11 | 0.64 | 98.80 | 0.42 | 100.00 | 0.00 | 6.75 | 1.62 |
| ResNet-101 | ✗ | 99.87 | 0.08 | 100.00 | 0.00 | 100.00 | 0.00 | 34.10 | 2.22 |
| ResNet-101 | ✓ | 99.97 | 0.00 | 100.00 | 0.00 | 100.00 | 0.00 | 7.65 | 1.39 |
| ResNet-152 | ✗ | 99.73 | 0.20 | 100.00 | 0.00 | 100.00 | 0.00 | 40.14 | 2.93 |
| ResNet-152 | ✓ | 99.89 | 0.04 | 99.99 | 0.01 | 100.00 | 0.00 | 5.36 | 1.17 |
| ResNet-18 | ✗ | 99.99 | 0.00 | 100.00 | 0.00 | 100.00 | 0.00 | 11.30 | 2.80 |
| ResNet-50 | ✗ | 99.82 | 0.09 | 100.00 | 0.00 | 100.00 | 0.00 | 28.14 | 4.84 |
| Swin-Base | ✗ | 97.79 | 0.38 | 99.85 | 0.02 | 66.67 | 33.33 | 19.98 | 0.73 |
| Swin-Tiny | ✗ | 99.01 | 0.47 | 99.90 | 0.04 | 70.47 | 29.53 | 28.03 | 2.45 |
| WideResNet | ✗ | 99.89 | 0.08 | 100.00 | 0.00 | 100.00 | 0.00 | 34.04 | 5.25 |

Table 15: $c = 1$

| Model | Pretrained | Train | | ID Val | | OOD Val | | Test | |
|---|---|---|---|---|---|---|---|---|---|
| | | AVG | SEM | AVG | SEM | AVG | SEM | AVG | SEM |
| ConvNeXt-Base | ✗ | 0.00 | 0.00 | 0.00 | 0.00 | 0.00 | 0.00 | 0.00 | 0.00 |
| ConvNeXt-Small | ✗ | 33.09 | 33.09 | 33.33 | 33.33 | 33.33 | 33.33 | 2.74 | 2.74 |
| ConvNeXt-Tiny | ✗ | 32.45 | 32.45 | 33.33 | 33.33 | 33.33 | 33.33 | 1.41 | 1.41 |
| DenseNet-121 | ✗ | 99.99 | 0.00 | 100.00 | 0.00 | 100.00 | 0.00 | 12.04 | 0.96 |
| DenseNet-121 | ✓ | 99.96 | 0.01 | 100.00 | 0.00 | 95.77 | 4.23 | 4.14 | 0.71 |
| DenseNet-161 | ✗ | 99.93 | 0.05 | 100.00 | 0.00 | 100.00 | 0.00 | 14.38 | 1.70 |
| DenseNet-201 | ✗ | 99.99 | 0.01 | 100.00 | 0.00 | 100.00 | 0.00 | 9.60 | 1.66 |
| ED | ✗ | 100.00 | 0.00 | 100.00 | 0.00 | 100.00 | 0.00 | 96.24 | 3.58 |
| MLP | ✗ | 29.98 | 29.98 | 32.84 | 32.84 | 33.33 | 33.33 | 2.36 | 2.36 |
| ResNet-101 | ✗ | 99.89 | 0.06 | 100.00 | 0.00 | 100.00 | 0.00 | 59.41 | 5.58 |
| ResNet-101 | ✓ | 99.96 | 0.01 | 99.99 | 0.01 | 100.00 | 0.00 | 6.95 | 0.61 |
| ResNet-152 | ✗ | 99.83 | 0.08 | 100.00 | 0.00 | 100.00 | 0.00 | 63.83 | 16.22 |
| ResNet-152 | ✓ | 99.96 | 0.01 | 100.00 | 0.00 | 91.12 | 8.88 | 5.70 | 1.04 |
| ResNet-18 | ✗ | 99.99 | 0.01 | 100.00 | 0.00 | 100.00 | 0.00 | 17.54 | 1.20 |
| ResNet-50 | ✗ | 99.81 | 0.17 | 100.00 | 0.00 | 100.00 | 0.00 | 41.45 | 5.12 |
| Swin-Base | ✗ | 65.75 | 32.88 | 66.54 | 33.27 | 66.67 | 33.33 | 4.01 | 3.96 |
| Swin-Tiny | ✗ | 98.80 | 0.27 | 99.85 | 0.10 | 100.00 | 0.00 | 1.93 | 0.82 |
| WideResNet | ✗ | 33.31 | 33.31 | 33.33 | 33.33 | 33.33 | 33.33 | 15.73 | 15.73 |

Table 16: $c = 2$

### E.1.3 Cars3D

| Model | Pretrained | Train | | ID Val | | OOD Val | | Test | |
|---|---|---|---|---|---|---|---|---|---|
| | | AVG | SEM | AVG | SEM | AVG | SEM | AVG | SEM |
| ConvNeXt-Base | ✗ | 98.24 | 0.07 | 89.10 | 0.06 | 100.00 | 0.00 | 0.00 | 0.00 |
| ConvNeXt-Small | ✗ | 98.22 | 0.10 | 87.67 | 1.20 | 100.00 | 0.00 | 0.00 | 0.00 |
| ConvNeXt-Tiny | ✗ | 98.23 | 0.08 | 87.90 | 1.39 | 66.67 | 33.33 | 0.01 | 0.01 |
| DenseNet-121 | ✗ | 99.51 | 0.33 | 94.46 | 0.41 | 100.00 | 0.00 | 0.00 | 0.00 |
| DenseNet-121 | ✓ | 99.82 | 0.02 | 60.88 | 0.65 | 66.67 | 33.33 | 0.00 | 0.00 |
| DenseNet-161 | ✗ | 99.52 | 0.24 | 95.33 | 0.44 | 100.00 | 0.00 | 0.00 | 0.00 |
| DenseNet-201 | ✗ | 99.17 | 0.28 | 94.60 | 0.41 | 100.00 | 0.00 | 0.00 | 0.00 |
| ED | ✗ | 99.85 | 0.02 | 94.89 | 0.41 | 100.00 | 0.00 | 0.00 | 0.00 |
| MLP | ✗ | 30.51 | 2.68 | 26.43 | 1.90 | 66.67 | 33.33 | 0.00 | 0.00 |
| ResNet-101 | ✗ | 99.48 | 0.25 | 93.50 | 0.23 | 66.67 | 33.33 | 0.00 | 0.00 |
| ResNet-101 | ✓ | 99.56 | 0.11 | 70.90 | 0.65 | 66.67 | 33.33 | 0.00 | 0.00 |
| ResNet-152 | ✗ | 99.74 | 0.05 | 94.39 | 0.52 | 100.00 | 0.00 | 0.00 | 0.00 |
| ResNet-152 | ✓ | 99.49 | 0.02 | 75.78 | 0.43 | 66.67 | 33.33 | 0.00 | 0.00 |
| ResNet-18 | ✗ | 99.85 | 0.02 | 94.37 | 0.60 | 100.00 | 0.00 | 0.00 | 0.00 |
| ResNet-50 | ✗ | 99.64 | 0.12 | 93.86 | 0.34 | 100.00 | 0.00 | 0.00 | 0.00 |
| Swin-Base | ✗ | 98.27 | 0.06 | 91.87 | 0.94 | 100.00 | 0.00 | 0.00 | 0.00 |
| Swin-Tiny | ✗ | 98.27 | 0.07 | 91.01 | 0.17 | 100.00 | 0.00 | 0.00 | 0.00 |
| WideResNet | ✗ | 99.42 | 0.33 | 94.01 | 0.48 | 100.00 | 0.00 | 0.00 | 0.00 |

Table 17: $c = 0$

| Model | Pretrained | Train | | ID Val | | OOD Val | | Test | |
|---|---|---|---|---|---|---|---|---|---|
| | | AVG | SEM | AVG | SEM | AVG | SEM | AVG | SEM |
| ConvNeXt-Base | ✗ | 98.81 | 0.07 | 87.21 | 0.50 | 33.33 | 33.33 | 30.08 | 1.35 |
| ConvNeXt-Small | ✗ | 98.73 | 0.05 | 86.28 | 0.18 | 33.33 | 33.33 | 27.41 | 0.54 |
| ConvNeXt-Tiny | ✗ | 98.68 | 0.04 | 86.32 | 0.07 | 33.33 | 33.33 | 25.90 | 1.17 |
| DenseNet-121 | ✗ | 99.64 | 0.27 | 92.24 | 0.85 | 100.00 | 0.00 | 38.13 | 0.83 |
| DenseNet-121 | ✓ | 99.85 | 0.03 | 54.95 | 1.38 | 33.33 | 33.33 | 6.38 | 0.59 |
| DenseNet-161 | ✗ | 99.89 | 0.02 | 93.91 | 0.65 | 100.00 | 0.00 | 40.37 | 1.15 |
| DenseNet-201 | ✗ | 99.58 | 0.23 | 93.83 | 0.57 | 100.00 | 0.00 | 38.77 | 0.61 |
| ED | ✗ | 99.91 | 0.01 | 94.62 | 0.59 | 100.00 | 0.00 | 46.70 | 0.98 |
| MLP | ✗ | 47.58 | 2.19 | 41.82 | 2.58 | 33.33 | 33.33 | 4.53 | 1.87 |
| ResNet-101 | ✗ | 99.48 | 0.32 | 92.90 | 0.94 | 66.67 | 33.33 | 35.01 | 1.94 |
| ResNet-101 | ✓ | 99.51 | 0.15 | 67.39 | 0.47 | 66.67 | 33.33 | 10.15 | 1.05 |
| ResNet-152 | ✗ | 99.33 | 0.28 | 92.04 | 0.81 | 66.67 | 33.33 | 34.30 | 0.97 |
| ResNet-152 | ✓ | 99.60 | 0.06 | 69.24 | 1.13 | 100.00 | 0.00 | 11.29 | 0.59 |
| ResNet-18 | ✗ | 99.89 | 0.03 | 92.88 | 0.51 | 100.00 | 0.00 | 33.90 | 1.84 |
| ResNet-50 | ✗ | 99.43 | 0.27 | 92.42 | 1.29 | 100.00 | 0.00 | 39.20 | 2.36 |
| Swin-Base | ✗ | 98.81 | 0.00 | 90.79 | 0.57 | 100.00 | 0.00 | 33.30 | 0.92 |
| Swin-Tiny | ✗ | 98.84 | 0.03 | 90.47 | 0.28 | 100.00 | 0.00 | 31.13 | 0.59 |
| WideResNet | ✗ | 99.69 | 0.12 | 93.77 | 0.87 | 100.00 | 0.00 | 39.89 | 1.89 |

Table 18: $c = 1$

| Model | Pretrained | Train | | ID Val | | OOD Val | | Test | |
|---|---|---|---|---|---|---|---|---|---|
| | | AVG | SEM | AVG | SEM | AVG | SEM | AVG | SEM |
| ConvNeXt-Base | ✗ | 98.75 | 0.05 | 80.87 | 0.03 | 66.67 | 33.33 | 38.65 | 0.63 |
| ConvNeXt-Small | ✗ | 98.82 | 0.03 | 79.17 | 0.22 | 66.67 | 33.33 | 34.87 | 1.40 |
| ConvNeXt-Tiny | ✗ | 98.75 | 0.11 | 80.87 | 1.13 | 100.00 | 0.00 | 35.06 | 0.53 |
| DenseNet-121 | ✗ | 99.87 | 0.01 | 91.97 | 0.38 | 66.67 | 33.33 | 57.88 | 0.83 |
| DenseNet-121 | ✓ | 99.88 | 0.02 | 44.80 | 1.60 | 33.33 | 33.33 | 7.74 | 0.44 |
| DenseNet-161 | ✗ | 99.87 | 0.01 | 93.03 | 0.13 | 66.67 | 33.33 | 61.33 | 1.78 |
| DenseNet-201 | ✗ | 99.39 | 0.25 | 91.70 | 0.21 | 66.67 | 33.33 | 58.06 | 0.81 |
| ED | ✗ | 99.89 | 0.02 | 92.53 | 0.19 | 66.67 | 33.33 | 71.25 | 1.34 |
| MLP | ✗ | 35.50 | 3.17 | 28.13 | 2.03 | 0.00 | 0.00 | 3.16 | 0.74 |
| ResNet-101 | ✗ | 99.84 | 0.06 | 90.90 | 0.30 | 66.67 | 33.33 | 47.98 | 1.71 |
| ResNet-101 | ✓ | 99.62 | 0.06 | 57.00 | 2.05 | 33.33 | 33.33 | 14.54 | 1.21 |
| ResNet-152 | ✗ | 99.84 | 0.06 | 90.87 | 0.43 | 66.67 | 33.33 | 47.98 | 0.82 |
| ResNet-152 | ✓ | 99.39 | 0.13 | 58.87 | 0.95 | 100.00 | 0.00 | 14.80 | 0.16 |
| ResNet-18 | ✗ | 99.66 | 0.23 | 90.77 | 0.33 | 100.00 | 0.00 | 51.00 | 0.19 |
| ResNet-50 | ✗ | 99.85 | 0.07 | 91.17 | 0.49 | 100.00 | 0.00 | 51.49 | 1.09 |
| Swin-Base | ✗ | 98.85 | 0.02 | 86.60 | 0.76 | 66.67 | 33.33 | 46.09 | 1.50 |
| Swin-Tiny | ✗ | 98.81 | 0.08 | 84.30 | 0.81 | 66.67 | 33.33 | 39.90 | 0.95 |
| WideResNet | ✗ | 99.62 | 0.16 | 90.73 | 0.30 | 66.67 | 33.33 | 51.11 | 0.88 |

Table 19: $c = 2$

### E.1.4 Shapes3D

| Model | Pretrained | Train | | ID Val | | OOD Val | | Test | |
|---|---|---|---|---|---|---|---|---|---|
| | | AVG | SEM | AVG | SEM | AVG | SEM | AVG | SEM |
| ConvNeXt-Base | ✗ | 99.99 | 0.01 | 100.00 | 0.00 | 100.00 | 0.00 | 0.00 | 0.00 |
| ConvNeXt-Small | ✗ | 100.00 | 0.00 | 100.00 | 0.00 | 100.00 | 0.00 | 0.00 | 0.00 |
| ConvNeXt-Tiny | ✗ | 100.00 | 0.00 | 100.00 | 0.00 | 100.00 | 0.00 | 0.00 | 0.00 |
| DenseNet-121 | ✗ | 100.00 | 0.00 | 100.00 | 0.00 | 100.00 | 0.00 | 0.00 | 0.00 |
| DenseNet-121 | ✓ | 100.00 | 0.00 | 100.00 | 0.00 | 100.00 | 0.00 | 0.00 | 0.00 |
| DenseNet-161 | ✗ | 100.00 | 0.00 | 100.00 | 0.00 | 100.00 | 0.00 | 0.00 | 0.00 |
| DenseNet-201 | ✗ | 100.00 | 0.00 | 100.00 | 0.00 | 100.00 | 0.00 | 0.00 | 0.00 |
| ED | ✗ | 100.00 | 0.00 | 100.00 | 0.00 | 100.00 | 0.00 | 0.00 | 0.00 |
| MLP | ✗ | 99.61 | 0.37 | 99.99 | 0.01 | 100.00 | 0.00 | 0.00 | 0.00 |
| ResNet-101 | ✗ | 99.97 | 0.03 | 100.00 | 0.00 | 100.00 | 0.00 | 0.00 | 0.00 |
| ResNet-101 | ✓ | 100.00 | 0.00 | 100.00 | 0.00 | 100.00 | 0.00 | 0.00 | 0.00 |
| ResNet-152 | ✗ | 100.00 | 0.00 | 100.00 | 0.00 | 100.00 | 0.00 | 0.00 | 0.00 |
| ResNet-152 | ✓ | 100.00 | 0.00 | 100.00 | 0.00 | 100.00 | 0.00 | 0.00 | 0.00 |
| ResNet-18 | ✗ | 100.00 | 0.00 | 100.00 | 0.00 | 100.00 | 0.00 | 0.00 | 0.00 |
| ResNet-50 | ✗ | 100.00 | 0.00 | 100.00 | 0.00 | 100.00 | 0.00 | 0.00 | 0.00 |
| Swin-Base | ✗ | 99.98 | 0.01 | 100.00 | 0.00 | 100.00 | 0.00 | 0.00 | 0.00 |
| Swin-Tiny | ✗ | 99.99 | 0.00 | 100.00 | 0.00 | 100.00 | 0.00 | 0.00 | 0.00 |
| WideResNet | ✗ | 100.00 | 0.00 | 100.00 | 0.00 | 100.00 | 0.00 | 0.00 | 0.00 |

Table 20: $c = 0$

| Model | Pretrained | Train | | ID Val | | OOD Val | | Test | |
|---|---|---|---|---|---|---|---|---|---|
| | | AVG | SEM | AVG | SEM | AVG | SEM | AVG | SEM |
| ConvNeXt-Base | ✗ | 99.99 | 0.01 | 100.00 | 0.00 | 100.00 | 0.00 | 88.47 | 1.21 |
| ConvNeXt-Small | ✗ | 99.79 | 0.21 | 100.00 | 0.00 | 100.00 | 0.00 | 87.21 | 0.42 |
| ConvNeXt-Tiny | ✗ | 100.00 | 0.00 | 100.00 | 0.00 | 100.00 | 0.00 | 88.47 | 0.38 |
| DenseNet-121 | ✗ | 100.00 | 0.00 | 100.00 | 0.00 | 100.00 | 0.00 | 79.90 | 0.48 |
| DenseNet-121 | ✓ | 100.00 | 0.00 | 100.00 | 0.00 | 100.00 | 0.00 | 62.27 | 1.76 |
| DenseNet-161 | ✗ | 100.00 | 0.00 | 100.00 | 0.00 | 100.00 | 0.00 | 78.74 | 0.76 |
| DenseNet-201 | ✗ | 100.00 | 0.00 | 100.00 | 0.00 | 100.00 | 0.00 | 81.14 | 0.59 |
| ED | ✗ | 100.00 | 0.00 | 100.00 | 0.00 | 100.00 | 0.00 | 96.09 | 2.65 |
| MLP | ✗ | 99.86 | 0.14 | 99.97 | 0.03 | 100.00 | 0.00 | 68.75 | 6.20 |
| ResNet-101 | ✗ | 99.59 | 0.41 | 99.99 | 0.01 | 100.00 | 0.00 | 80.08 | 4.93 |
| ResNet-101 | ✓ | 100.00 | 0.00 | 100.00 | 0.00 | 100.00 | 0.00 | 67.56 | 3.77 |
| ResNet-152 | ✗ | 100.00 | 0.00 | 100.00 | 0.00 | 100.00 | 0.00 | 74.47 | 5.86 |
| ResNet-152 | ✓ | 100.00 | 0.00 | 100.00 | 0.00 | 100.00 | 0.00 | 65.30 | 4.98 |
| ResNet-18 | ✗ | 100.00 | 0.00 | 100.00 | 0.00 | 100.00 | 0.00 | 85.47 | 0.71 |
| ResNet-50 | ✗ | 100.00 | 0.00 | 100.00 | 0.00 | 100.00 | 0.00 | 86.63 | 1.81 |
| Swin-Base | ✗ | 99.96 | 0.02 | 100.00 | 0.00 | 100.00 | 0.00 | 83.37 | 1.46 |
| Swin-Tiny | ✗ | 99.98 | 0.00 | 100.00 | 0.00 | 100.00 | 0.00 | 83.72 | 0.48 |
| WideResNet | ✗ | 100.00 | 0.00 | 100.00 | 0.00 | 100.00 | 0.00 | 81.45 | 3.95 |

Table 21: $c = 1$

| Model | Pretrained | Train | | ID Val | | OOD Val | | Test | |
|---|---|---|---|---|---|---|---|---|---|
| | | AVG | SEM | AVG | SEM | AVG | SEM | AVG | SEM |
| ConvNeXt-Base | ✗ | 100.00 | 0.00 | 100.00 | 0.00 | 100.00 | 0.00 | 100.00 | 0.00 |
| ConvNeXt-Small | ✗ | 100.00 | 0.00 | 100.00 | 0.00 | 100.00 | 0.00 | 100.00 | 0.00 |
| ConvNeXt-Tiny | ✗ | 100.00 | 0.00 | 100.00 | 0.00 | 100.00 | 0.00 | 99.99 | 0.00 |
| DenseNet-121 | ✗ | 100.00 | 0.00 | 100.00 | 0.00 | 100.00 | 0.00 | 99.91 | 0.02 |
| DenseNet-121 | ✓ | 100.00 | 0.00 | 100.00 | 0.00 | 100.00 | 0.00 | 85.44 | 2.66 |
| DenseNet-161 | ✗ | 100.00 | 0.00 | 100.00 | 0.00 | 100.00 | 0.00 | 99.55 | 0.19 |
| DenseNet-201 | ✗ | 100.00 | 0.00 | 100.00 | 0.00 | 100.00 | 0.00 | 99.93 | 0.01 |
| ED | ✗ | 100.00 | 0.00 | 100.00 | 0.00 | 100.00 | 0.00 | 100.00 | 0.00 |
| MLP | ✗ | 99.95 | 0.04 | 99.99 | 0.01 | 100.00 | 0.00 | 79.78 | 6.39 |
| ResNet-101 | ✗ | 100.00 | 0.00 | 100.00 | 0.00 | 100.00 | 0.00 | 99.14 | 0.60 |
| ResNet-101 | ✓ | 100.00 | 0.00 | 100.00 | 0.00 | 100.00 | 0.00 | 91.60 | 0.38 |
| ResNet-152 | ✗ | 100.00 | 0.00 | 100.00 | 0.00 | 100.00 | 0.00 | 98.82 | 0.80 |
| ResNet-152 | ✓ | 100.00 | 0.00 | 100.00 | 0.00 | 100.00 | 0.00 | 86.64 | 2.08 |
| ResNet-18 | ✗ | 100.00 | 0.00 | 100.00 | 0.00 | 100.00 | 0.00 | 99.98 | 0.01 |
| ResNet-50 | ✗ | 100.00 | 0.00 | 100.00 | 0.00 | 100.00 | 0.00 | 99.68 | 0.13 |
| Swin-Base | ✗ | 99.97 | 0.01 | 100.00 | 0.00 | 100.00 | 0.00 | 99.86 | 0.12 |
| Swin-Tiny | ✗ | 100.00 | 0.00 | 100.00 | 0.00 | 100.00 | 0.00 | 99.99 | 0.01 |
| WideResNet | ✗ | 100.00 | 0.00 | 100.00 | 0.00 | 100.00 | 0.00 | 99.97 | 0.01 |

Table 22: $c = 2$

| Model | Pretrained | Train | | ID Val | | OOD Val | | Test | |
|---|---|---|---|---|---|---|---|---|---|
| | | AVG | SEM | AVG | SEM | AVG | SEM | AVG | SEM |
| ConvNeXt-Base | ✗ | 100.00 | 0.00 | 100.00 | 0.00 | 100.00 | 0.00 | 100.00 | 0.00 |
| ConvNeXt-Small | ✗ | 99.98 | 0.01 | 100.00 | 0.00 | 100.00 | 0.00 | 100.00 | 0.00 |
| ConvNeXt-Tiny | ✗ | 100.00 | 0.00 | 100.00 | 0.00 | 100.00 | 0.00 | 100.00 | 0.00 |
| DenseNet-121 | ✗ | 100.00 | 0.00 | 100.00 | 0.00 | 100.00 | 0.00 | 99.98 | 0.01 |
| DenseNet-121 | ✓ | 100.00 | 0.00 | 100.00 | 0.00 | 100.00 | 0.00 | 86.57 | 2.14 |
| DenseNet-161 | ✗ | 100.00 | 0.00 | 100.00 | 0.00 | 100.00 | 0.00 | 99.99 | 0.00 |
| DenseNet-201 | ✗ | 100.00 | 0.00 | 100.00 | 0.00 | 100.00 | 0.00 | 99.98 | 0.01 |
| ED | ✗ | 100.00 | 0.00 | 100.00 | 0.00 | 100.00 | 0.00 | 100.00 | 0.00 |
| MLP | ✗ | 99.29 | 0.24 | 99.97 | 0.02 | 100.00 | 0.00 | 61.90 | 6.47 |
| ResNet-101 | ✗ | 100.00 | 0.00 | 100.00 | 0.00 | 100.00 | 0.00 | 99.54 | 0.29 |
| ResNet-101 | ✓ | 100.00 | 0.00 | 100.00 | 0.00 | 100.00 | 0.00 | 87.05 | 0.97 |
| ResNet-152 | ✗ | 100.00 | 0.00 | 100.00 | 0.00 | 100.00 | 0.00 | 99.74 | 0.16 |
| ResNet-152 | ✓ | 100.00 | 0.00 | 100.00 | 0.00 | 100.00 | 0.00 | 89.38 | 1.95 |
| ResNet-18 | ✗ | 100.00 | 0.00 | 100.00 | 0.00 | 100.00 | 0.00 | 99.96 | 0.01 |
| ResNet-50 | ✗ | 100.00 | 0.00 | 100.00 | 0.00 | 100.00 | 0.00 | 99.56 | 0.19 |
| Swin-Base | ✗ | 99.97 | 0.01 | 100.00 | 0.00 | 100.00 | 0.00 | 99.98 | 0.02 |
| Swin-Tiny | ✗ | 100.00 | 0.00 | 100.00 | 0.00 | 100.00 | 0.00 | 99.97 | 0.01 |
| WideResNet | ✗ | 100.00 | 0.00 | 100.00 | 0.00 | 100.00 | 0.00 | 99.89 | 0.09 |

Table 23: $c = 3$

| Model | Pretrained | Train | | ID Val | | OOD Val | | Test | |
|---|---|---|---|---|---|---|---|---|---|
| | | AVG | SEM | AVG | SEM | AVG | SEM | AVG | SEM |
| ConvNeXt-Base | ✗ | 100.00 | 0.00 | 100.00 | 0.00 | 100.00 | 0.00 | 100.00 | 0.00 |
| ConvNeXt-Small | ✗ | 100.00 | 0.00 | 100.00 | 0.00 | 100.00 | 0.00 | 100.00 | 0.00 |
| ConvNeXt-Tiny | ✗ | 100.00 | 0.00 | 100.00 | 0.00 | 100.00 | 0.00 | 100.00 | 0.00 |
| DenseNet-121 | ✗ | 100.00 | 0.00 | 100.00 | 0.00 | 100.00 | 0.00 | 100.00 | 0.00 |
| DenseNet-121 | ✓ | 100.00 | 0.00 | 100.00 | 0.00 | 100.00 | 0.00 | 94.98 | 1.31 |
| DenseNet-161 | ✗ | 100.00 | 0.00 | 100.00 | 0.00 | 100.00 | 0.00 | 100.00 | 0.00 |
| DenseNet-201 | ✗ | 100.00 | 0.00 | 100.00 | 0.00 | 100.00 | 0.00 | 100.00 | 0.00 |
| ED | ✗ | 100.00 | 0.00 | 100.00 | 0.00 | 100.00 | 0.00 | 100.00 | 0.00 |
| MLP | ✗ | 99.57 | 0.22 | 99.98 | 0.01 | 100.00 | 0.00 | 55.03 | 14.76 |
| ResNet-101 | ✗ | 100.00 | 0.00 | 100.00 | 0.00 | 100.00 | 0.00 | 100.00 | 0.00 |
| ResNet-101 | ✓ | 100.00 | 0.00 | 100.00 | 0.00 | 100.00 | 0.00 | 98.23 | 0.58 |
| ResNet-152 | ✗ | 100.00 | 0.00 | 99.99 | 0.01 | 100.00 | 0.00 | 100.00 | 0.00 |
| ResNet-152 | ✓ | 100.00 | 0.00 | 100.00 | 0.00 | 100.00 | 0.00 | 96.97 | 0.27 |
| ResNet-18 | ✗ | 100.00 | 0.00 | 100.00 | 0.00 | 100.00 | 0.00 | 100.00 | 0.00 |
| ResNet-50 | ✗ | 100.00 | 0.00 | 100.00 | 0.00 | 100.00 | 0.00 | 100.00 | 0.00 |
| Swin-Base | ✗ | 99.94 | 0.05 | 100.00 | 0.00 | 100.00 | 0.00 | 100.00 | 0.00 |
| Swin-Tiny | ✗ | 100.00 | 0.00 | 100.00 | 0.00 | 100.00 | 0.00 | 100.00 | 0.00 |
| WideResNet | ✗ | 100.00 | 0.00 | 100.00 | 0.00 | 100.00 | 0.00 | 100.00 | 0.00 |

Table 24: $c = 4$

### E.1.5 MPI3D

| Model | Pretrained | Train | | ID Val | | OOD Val | | Test | |
|---|---|---|---|---|---|---|---|---|---|
| | | AVG | SEM | AVG | SEM | AVG | SEM | AVG | SEM |
| ConvNeXt-Base | ✗ | 99.53 | 0.05 | 99.51 | 0.06 | 100.00 | 0.00 | 0.00 | 0.00 |
| ConvNeXt-Small | ✗ | 99.46 | 0.16 | 99.67 | 0.02 | 100.00 | 0.00 | 0.00 | 0.00 |
| ConvNeXt-Tiny | ✗ | 99.33 | 0.21 | 99.54 | 0.09 | 100.00 | 0.00 | 0.00 | 0.00 |
| DenseNet-121 | ✗ | 99.13 | 0.40 | 98.68 | 0.21 | 100.00 | 0.00 | 0.00 | 0.00 |
| DenseNet-121 | ✓ | 99.66 | 0.07 | 97.32 | 0.37 | 100.00 | 0.00 | 0.00 | 0.00 |
| DenseNet-161 | ✗ | 99.61 | 0.07 | 98.80 | 0.20 | 100.00 | 0.00 | 0.00 | 0.00 |
| DenseNet-201 | ✗ | 99.29 | 0.19 | 98.53 | 0.46 | 100.00 | 0.00 | 0.00 | 0.00 |
| ED | ✗ | 99.57 | 0.12 | 99.33 | 0.23 | 100.00 | 0.00 | 0.00 | 0.00 |
| MLP | ✗ | 11.25 | 2.68 | 14.93 | 2.68 | 0.00 | 0.00 | 0.00 | 0.00 |
| ResNet-101 | ✗ | 98.90 | 0.32 | 99.18 | 0.25 | 100.00 | 0.00 | 0.00 | 0.00 |
| ResNet-101 | ✓ | 98.83 | 0.47 | 96.91 | 0.46 | 100.00 | 0.00 | 0.00 | 0.00 |
| ResNet-152 | ✗ | 98.87 | 0.26 | 98.51 | 0.28 | 100.00 | 0.00 | 0.00 | 0.00 |
| ResNet-152 | ✓ | 98.79 | 0.32 | 96.87 | 0.32 | 100.00 | 0.00 | 0.00 | 0.00 |
| ResNet-18 | ✗ | 99.28 | 0.24 | 98.55 | 0.06 | 100.00 | 0.00 | 0.00 | 0.00 |
| ResNet-50 | ✗ | 98.71 | 0.35 | 98.84 | 0.11 | 100.00 | 0.00 | 0.00 | 0.00 |
| Swin-Base | ✗ | 98.60 | 0.08 | 99.59 | 0.02 | 100.00 | 0.00 | 0.00 | 0.00 |
| Swin-Tiny | ✗ | 98.65 | 0.34 | 99.52 | 0.05 | 100.00 | 0.00 | 0.00 | 0.00 |
| WideResNet | ✗ | 99.13 | 0.08 | 99.53 | 0.09 | 100.00 | 0.00 | 0.00 | 0.00 |

Table 25: $c = 0$

| Model | Pretrained | Train | | ID Val | | OOD Val | | Test | |
|---|---|---|---|---|---|---|---|---|---|
| | | AVG | SEM | AVG | SEM | AVG | SEM | AVG | SEM |
| ConvNeXt-Base | ✗ | 99.32 | 0.14 | 99.70 | 0.04 | 100.00 | 0.00 | 39.32 | 1.30 |
| ConvNeXt-Small | ✗ | 99.65 | 0.02 | 99.66 | 0.01 | 100.00 | 0.00 | 37.99 | 0.95 |
| ConvNeXt-Tiny | ✗ | 99.50 | 0.03 | 99.63 | 0.08 | 100.00 | 0.00 | 36.88 | 0.17 |
| DenseNet-121 | ✗ | 99.73 | 0.03 | 99.72 | 0.03 | 100.00 | 0.00 | 54.92 | 1.02 |
| DenseNet-121 | ✓ | 99.59 | 0.04 | 97.95 | 0.63 | 100.00 | 0.00 | 23.45 | 0.63 |
| DenseNet-161 | ✗ | 99.77 | 0.04 | 99.81 | 0.03 | 100.00 | 0.00 | 57.90 | 1.01 |
| DenseNet-201 | ✗ | 99.66 | 0.02 | 99.54 | 0.07 | 100.00 | 0.00 | 53.85 | 2.56 |
| ED | ✗ | 99.67 | 0.00 | 99.66 | 0.20 | 100.00 | 0.00 | 70.76 | 0.24 |
| MLP | ✗ | 16.06 | 3.35 | 18.60 | 2.93 | 16.67 | 16.67 | 1.39 | 0.36 |
| ResNet-101 | ✗ | 99.30 | 0.17 | 99.36 | 0.08 | 100.00 | 0.00 | 42.49 | 1.91 |
| ResNet-101 | ✓ | 99.43 | 0.08 | 98.32 | 0.24 | 100.00 | 0.00 | 27.66 | 1.23 |
| ResNet-152 | ✗ | 99.59 | 0.03 | 99.28 | 0.14 | 100.00 | 0.00 | 45.89 | 0.46 |
| ResNet-152 | ✓ | 98.96 | 0.19 | 98.29 | 0.10 | 100.00 | 0.00 | 24.59 | 0.68 |
| ResNet-18 | ✗ | 99.43 | 0.21 | 99.29 | 0.30 | 100.00 | 0.00 | 41.59 | 2.16 |
| ResNet-50 | ✗ | 99.38 | 0.02 | 99.54 | 0.16 | 100.00 | 0.00 | 45.47 | 1.69 |
| Swin-Base | ✗ | 99.29 | 0.06 | 99.70 | 0.02 | 100.00 | 0.00 | 37.02 | 0.39 |
| Swin-Tiny | ✗ | 98.98 | 0.29 | 99.69 | 0.06 | 100.00 | 0.00 | 34.79 | 1.64 |
| WideResNet | ✗ | 99.31 | 0.09 | 99.49 | 0.22 | 100.00 | 0.00 | 46.96 | 1.17 |

Table 26: $c = 1$

| Model | Pretrained | Train | | ID Val | | OOD Val | | Test | |
|---|---|---|---|---|---|---|---|---|---|
| | | AVG | SEM | AVG | SEM | AVG | SEM | AVG | SEM |
| ConvNeXt-Base | ✗ | 99.55 | 0.12 | 99.77 | 0.04 | 100.00 | 0.00 | 56.93 | 0.79 |
| ConvNeXt-Small | ✗ | 99.42 | 0.13 | 99.76 | 0.06 | 100.00 | 0.00 | 54.68 | 1.70 |
| ConvNeXt-Tiny | ✗ | 0.00 | 0.00 | 0.00 | 0.00 | 0.00 | 0.00 | 0.00 | 0.00 |
| DenseNet-121 | ✗ | 99.75 | 0.05 | 99.63 | 0.04 | 100.00 | 0.00 | 60.32 | 0.49 |
| DenseNet-121 | ✓ | 99.51 | 0.01 | 98.96 | 0.04 | 83.33 | 16.67 | 39.66 | 1.49 |
| DenseNet-161 | ✗ | 99.81 | 0.02 | 99.83 | 0.04 | 100.00 | 0.00 | 58.53 | 1.72 |
| DenseNet-201 | ✗ | 99.61 | 0.14 | 99.51 | 0.08 | 100.00 | 0.00 | 54.18 | 0.16 |
| ED | ✗ | 99.62 | 0.02 | 99.85 | 0.02 | 100.00 | 0.00 | 66.83 | 0.72 |
| MLP | ✗ | 26.23 | 2.50 | 28.81 | 2.30 | 0.00 | 0.00 | 6.95 | 1.98 |
| ResNet-101 | ✗ | 98.82 | 0.74 | 99.40 | 0.33 | 100.00 | 0.00 | 50.14 | 1.49 |
| ResNet-101 | ✓ | 95.47 | 0.68 | 93.21 | 1.24 | 100.00 | 0.00 | 27.43 | 1.33 |
| ResNet-152 | ✗ | 99.60 | 0.07 | 99.56 | 0.10 | 100.00 | 0.00 | 48.13 | 3.54 |
| ResNet-152 | ✓ | 99.26 | 0.04 | 99.22 | 0.21 | 100.00 | 0.00 | 45.74 | 0.82 |
| ResNet-18 | ✗ | 99.66 | 0.04 | 99.78 | 0.06 | 100.00 | 0.00 | 55.56 | 2.18 |
| ResNet-50 | ✗ | 99.54 | 0.06 | 99.51 | 0.06 | 100.00 | 0.00 | 51.50 | 2.14 |
| Swin-Base | ✗ | 99.17 | 0.18 | 99.74 | 0.04 | 100.00 | 0.00 | 51.48 | 0.27 |
| Swin-Tiny | ✗ | 99.31 | 0.06 | 99.79 | 0.01 | 100.00 | 0.00 | 50.50 | 2.10 |
| WideResNet | ✗ | 99.69 | 0.08 | 99.67 | 0.01 | 100.00 | 0.00 | 55.77 | 1.52 |

Table 27: $c = 2$

| Model | Pretrained | Train | | ID Val | | OOD Val | | Test | |
|---|---|---|---|---|---|---|---|---|---|
| | | AVG | SEM | AVG | SEM | AVG | SEM | AVG | SEM |
| ConvNeXt-Base | ✗ | 99.56 | 0.07 | 99.84 | 0.00 | 100.00 | 0.00 | 87.22 | 1.16 |
| ConvNeXt-Small | ✗ | 99.56 | 0.08 | 99.77 | 0.03 | 100.00 | 0.00 | 86.17 | 0.64 |
| ConvNeXt-Tiny | ✗ | 0.00 | 0.00 | 0.00 | 0.00 | 0.00 | 0.00 | 0.00 | 0.00 |
| DenseNet-121 | ✗ | 99.65 | 0.07 | 99.85 | 0.04 | 100.00 | 0.00 | 96.91 | 0.60 |
| DenseNet-121 | ✓ | 99.63 | 0.13 | 98.94 | 0.20 | 100.00 | 0.00 | 73.12 | 2.14 |
| DenseNet-161 | ✗ | 99.79 | 0.07 | 99.91 | 0.01 | 100.00 | 0.00 | 96.68 | 0.14 |
| DenseNet-201 | ✗ | 99.57 | 0.14 | 99.78 | 0.05 | 100.00 | 0.00 | 95.83 | 0.63 |
| ED | ✗ | 99.67 | 0.03 | 99.78 | 0.01 | 100.00 | 0.00 | 97.85 | 0.02 |
| MLP | ✗ | 27.78 | 2.13 | 29.49 | 2.25 | 0.00 | 0.00 | 13.67 | 1.26 |
| ResNet-101 | ✗ | 98.86 | 0.18 | 99.03 | 0.14 | 100.00 | 0.00 | 89.53 | 0.07 |
| ResNet-101 | ✓ | 63.79 | 31.97 | 60.99 | 30.84 | 50.00 | 28.87 | 42.85 | 23.64 |
| ResNet-152 | ✗ | 99.30 | 0.23 | 99.49 | 0.12 | 100.00 | 0.00 | 90.01 | 2.60 |
| ResNet-152 | ✓ | 99.22 | 0.18 | 99.17 | 0.21 | 100.00 | 0.00 | 82.02 | 0.45 |
| ResNet-18 | ✗ | 99.61 | 0.07 | 99.71 | 0.10 | 100.00 | 0.00 | 94.95 | 0.12 |
| ResNet-50 | ✗ | 99.61 | 0.09 | 99.54 | 0.10 | 100.00 | 0.00 | 94.90 | 0.77 |
| Swin-Base | ✗ | 99.20 | 0.29 | 99.82 | 0.01 | 100.00 | 0.00 | 86.60 | 0.25 |
| Swin-Tiny | ✗ | 99.42 | 0.04 | 99.84 | 0.01 | 100.00 | 0.00 | 84.35 | 1.02 |
| WideResNet | ✗ | 99.65 | 0.07 | 99.60 | 0.12 | 100.00 | 0.00 | 94.49 | 0.29 |

Table 28: $c = 3$

| Model | Pretrained | Train | | ID Val | | OOD Val | | Test | |
|-------|-----------|-------|-----|--------|-----|---------|-----|------|-----|
| | | AVG | SEM | AVG | SEM | AVG | SEM | AVG | SEM |
| ConvNeXt-Base | ✗ | 99.67 | 0.02 | 99.76 | 0.02 | 100.00 | 0.00 | 61.03 | 0.93 |
| ConvNeXt-Small | ✗ | 99.52 | 0.03 | 99.75 | 0.03 | 100.00 | 0.00 | 60.54 | 1.14 |
| ConvNeXt-Tiny | ✗ | 99.57 | 0.09 | 99.73 | 0.03 | 100.00 | 0.00 | 60.01 | 1.44 |
| DenseNet-121 | ✗ | 99.53 | 0.05 | 99.63 | 0.06 | 100.00 | 0.00 | 68.48 | 2.83 |
| DenseNet-121 | ✓ | 99.67 | 0.09 | 99.09 | 0.17 | 100.00 | 0.00 | 55.73 | 0.29 |
| DenseNet-161 | ✗ | 99.70 | 0.05 | 99.82 | 0.04 | 100.00 | 0.00 | 71.66 | 0.79 |
| DenseNet-201 | ✗ | 99.26 | 0.12 | 99.38 | 0.07 | 100.00 | 0.00 | 68.16 | 1.51 |
| ED | ✗ | 99.57 | 0.16 | 99.69 | 0.11 | 100.00 | 0.00 | 75.08 | 1.89 |
| MLP | ✗ | 20.95 | 0.95 | 26.22 | 1.43 | 16.67 | 16.67 | 4.47 | 0.80 |
| ResNet-101 | ✗ | 97.24 | 0.80 | 98.21 | 0.11 | 100.00 | 0.00 | 59.55 | 2.96 |
| ResNet-101 | ✓ | 99.19 | 0.12 | 99.21 | 0.12 | 100.00 | 0.00 | 56.79 | 0.15 |
| ResNet-152 | ✗ | 99.42 | 0.18 | 99.69 | 0.05 | 100.00 | 0.00 | 63.05 | 2.45 |
| ResNet-152 | ✓ | 99.07 | 0.16 | 98.45 | 0.44 | 100.00 | 0.00 | 53.67 | 2.22 |
| ResNet-18 | ✗ | 99.71 | 0.06 | 99.64 | 0.01 | 100.00 | 0.00 | 65.52 | 1.22 |
| ResNet-50 | ✗ | 99.55 | 0.19 | 99.70 | 0.03 | 100.00 | 0.00 | 56.49 | 3.67 |
| Swin-Base | ✗ | 99.26 | 0.14 | 99.75 | 0.03 | 100.00 | 0.00 | 58.23 | 2.25 |
| Swin-Tiny | ✗ | 99.51 | 0.01 | 99.78 | 0.02 | 100.00 | 0.00 | 56.19 | 2.28 |
| WideResNet | ✗ | 99.50 | 0.04 | 99.75 | 0.03 | 100.00 | 0.00 | 61.60 | 0.65 |

Table 29: $c = 4$

| Model | Pretrained | Train | | ID Val | | OOD Val | | Test | |
|-------|-----------|-------|-----|--------|-----|---------|-----|------|-----|
| | | AVG | SEM | AVG | SEM | AVG | SEM | AVG | SEM |
| ConvNeXt-Base | ✗ | 99.59 | 0.03 | 99.83 | 0.03 | 100.00 | 0.00 | 93.94 | 0.93 |
| ConvNeXt-Small | ✗ | 99.56 | 0.06 | 99.75 | 0.02 | 100.00 | 0.00 | 91.69 | 0.99 |
| ConvNeXt-Tiny | ✗ | 99.59 | 0.09 | 99.79 | 0.02 | 100.00 | 0.00 | 91.60 | 0.24 |
| DenseNet-121 | ✗ | 99.71 | 0.08 | 99.74 | 0.02 | 100.00 | 0.00 | 98.39 | 0.15 |
| DenseNet-121 | ✓ | 99.63 | 0.06 | 98.67 | 0.27 | 100.00 | 0.00 | 86.63 | 0.96 |
| DenseNet-161 | ✗ | 99.73 | 0.04 | 99.65 | 0.11 | 100.00 | 0.00 | 97.87 | 0.34 |
| DenseNet-201 | ✗ | 99.64 | 0.04 | 99.66 | 0.03 | 100.00 | 0.00 | 96.85 | 0.42 |
| ED | ✗ | 99.65 | 0.07 | 99.63 | 0.03 | 83.33 | 16.67 | 97.08 | 0.58 |
| MLP | ✗ | 22.59 | 3.85 | 27.18 | 1.53 | 33.33 | 16.67 | 9.58 | 1.62 |
| ResNet-101 | ✗ | 95.64 | 1.82 | 94.86 | 1.63 | 66.67 | 16.67 | 89.20 | 3.04 |
| ResNet-101 | ✓ | 99.29 | 0.13 | 98.89 | 0.13 | 100.00 | 0.00 | 85.66 | 0.24 |
| ResNet-152 | ✗ | 99.64 | 0.06 | 99.61 | 0.10 | 100.00 | 0.00 | 97.50 | 0.50 |
| ResNet-152 | ✓ | 99.15 | 0.16 | 99.03 | 0.03 | 100.00 | 0.00 | 86.53 | 0.78 |
| ResNet-18 | ✗ | 99.70 | 0.07 | 99.60 | 0.09 | 100.00 | 0.00 | 96.03 | 0.25 |
| ResNet-50 | ✗ | 99.48 | 0.15 | 99.46 | 0.18 | 100.00 | 0.00 | 95.06 | 0.22 |
| Swin-Base | ✗ | 99.42 | 0.11 | 99.80 | 0.01 | 100.00 | 0.00 | 89.63 | 1.24 |
| Swin-Tiny | ✗ | 99.42 | 0.10 | 99.78 | 0.01 | 100.00 | 0.00 | 92.54 | 0.32 |
| WideResNet | ✗ | 99.39 | 0.11 | 99.77 | 0.05 | 100.00 | 0.00 | 96.98 | 0.40 |

Table 30: $c = 5$

### E.1.6 CLEVR

| Model | Pretrained | Train | | ID Val | | OOD Val | | Test | |
|---|---|---|---|---|---|---|---|---|---|
| | | AVG | SEM | AVG | SEM | AVG | SEM | AVG | SEM |
| ConvNeXt-Base | ✗ | 91.38 | 8.57 | 92.56 | 7.44 | 38.79 | 30.97 | 0.00 | 0.00 |
| ConvNeXt-Small | ✗ | 71.54 | 28.46 | 72.82 | 27.18 | 33.47 | 33.26 | 0.00 | 0.00 |
| ConvNeXt-Tiny | ✗ | 99.99 | 0.01 | 100.00 | 0.00 | 66.67 | 33.33 | 0.00 | 0.00 |
| DenseNet-121 | ✗ | 100.00 | 0.00 | 100.00 | 0.00 | 79.58 | 11.45 | 0.00 | 0.00 |
| DenseNet-121 | ✓ | 100.00 | 0.00 | 100.00 | 0.00 | 56.87 | 28.45 | 0.00 | 0.00 |
| DenseNet-161 | ✗ | 100.00 | 0.00 | 100.00 | 0.00 | 93.42 | 6.58 | 0.00 | 0.00 |
| DenseNet-201 | ✗ | 100.00 | 0.00 | 100.00 | 0.00 | 80.81 | 16.19 | 0.00 | 0.00 |
| ED | ✗ | 100.00 | 0.00 | 100.00 | 0.00 | 99.30 | 0.70 | 0.00 | 0.00 |
| MLP | ✗ | 0.00 | 0.00 | 0.00 | 0.00 | 0.00 | 0.00 | 0.00 | 0.00 |
| ResNet-101 | ✗ | 100.00 | 0.00 | 100.00 | 0.00 | 69.61 | 15.40 | 0.00 | 0.00 |
| ResNet-101 | ✓ | 100.00 | 0.00 | 100.00 | 0.00 | 35.82 | 28.14 | 0.00 | 0.00 |
| ResNet-152 | ✗ | 100.00 | 0.00 | 100.00 | 0.00 | 33.29 | 33.29 | 0.00 | 0.00 |
| ResNet-152 | ✓ | 100.00 | 0.00 | 100.00 | 0.00 | 24.90 | 13.28 | 0.00 | 0.00 |
| ResNet-18 | ✗ | 100.00 | 0.00 | 100.00 | 0.00 | 94.27 | 5.11 | 0.00 | 0.00 |
| ResNet-50 | ✗ | 100.00 | 0.00 | 99.98 | 0.02 | 65.46 | 32.73 | 0.00 | 0.00 |
| Swin-Base | ✗ | 99.96 | 0.01 | 100.00 | 0.00 | 70.62 | 29.03 | 0.00 | 0.00 |
| Swin-Tiny | ✗ | 99.97 | 0.01 | 100.00 | 0.00 | 71.70 | 27.18 | 0.00 | 0.00 |
| WideResNet | ✗ | 100.00 | 0.00 | 100.00 | 0.00 | 84.31 | 12.21 | 0.00 | 0.00 |

Table 31: $c = 0$

| Model | Pretrained | Train | | ID Val | | OOD Val | | Test | |
|---|---|---|---|---|---|---|---|---|---|
| | | AVG | SEM | AVG | SEM | AVG | SEM | AVG | SEM |
| ConvNeXt-Base | ✗ | 99.99 | 0.00 | 100.00 | 0.00 | 99.95 | 0.05 | 22.95 | 3.71 |
| ConvNeXt-Small | ✗ | 99.98 | 0.01 | 99.99 | 0.01 | 99.95 | 0.05 | 22.05 | 2.66 |
| ConvNeXt-Tiny | ✗ | 99.98 | 0.01 | 100.00 | 0.00 | 99.08 | 0.47 | 23.53 | 1.75 |
| DenseNet-121 | ✗ | 100.00 | 0.00 | 100.00 | 0.00 | 93.08 | 6.85 | 25.94 | 3.19 |
| DenseNet-121 | ✓ | 100.00 | 0.00 | 100.00 | 0.00 | 85.46 | 13.72 | 4.44 | 0.89 |
| DenseNet-161 | ✗ | 100.00 | 0.00 | 100.00 | 0.00 | 99.91 | 0.09 | 28.29 | 1.28 |
| DenseNet-201 | ✗ | 100.00 | 0.00 | 100.00 | 0.00 | 99.91 | 0.09 | 26.34 | 2.03 |
| ED | ✗ | 100.00 | 0.00 | 100.00 | 0.00 | 100.00 | 0.00 | 53.98 | 1.46 |
| MLP | ✗ | 0.00 | 0.00 | 0.00 | 0.00 | 0.00 | 0.00 | 0.00 | 0.00 |
| ResNet-101 | ✗ | 100.00 | 0.00 | 100.00 | 0.00 | 94.44 | 5.56 | 35.96 | 2.70 |
| ResNet-101 | ✓ | 100.00 | 0.00 | 100.00 | 0.00 | 77.50 | 22.50 | 15.18 | 2.56 |
| ResNet-152 | ✗ | 100.00 | 0.00 | 100.00 | 0.00 | 100.00 | 0.00 | 39.40 | 2.89 |
| ResNet-152 | ✓ | 100.00 | 0.00 | 99.99 | 0.01 | 88.62 | 5.76 | 16.14 | 3.56 |
| ResNet-18 | ✗ | 100.00 | 0.00 | 100.00 | 0.00 | 100.00 | 0.00 | 24.68 | 1.86 |
| ResNet-50 | ✗ | 100.00 | 0.00 | 100.00 | 0.00 | 99.95 | 0.05 | 34.08 | 9.28 |
| Swin-Base | ✗ | 99.92 | 0.05 | 99.94 | 0.01 | 99.72 | 0.16 | 11.05 | 2.80 |
| Swin-Tiny | ✗ | 99.97 | 0.01 | 99.92 | 0.02 | 98.76 | 1.24 | 11.80 | 1.78 |
| WideResNet | ✗ | 100.00 | 0.00 | 99.99 | 0.01 | 100.00 | 0.00 | 31.77 | 1.64 |

Table 32: $c = 1$

| Model | Pretrained | Train | | ID Val | | OOD Val | | Test | |
|---|---|---|---|---|---|---|---|---|---|
| | | AVG | SEM | AVG | SEM | AVG | SEM | AVG | SEM |
| ConvNeXt-Base | ✗ | 99.97 | 0.01 | 100.00 | 0.00 | 99.95 | 0.05 | 37.36 | 4.24 |
| ConvNeXt-Small | ✗ | 99.99 | 0.01 | 100.00 | 0.00 | 99.80 | 0.13 | 45.26 | 1.13 |
| ConvNeXt-Tiny | ✗ | 99.99 | 0.01 | 100.00 | 0.00 | 99.95 | 0.05 | 41.95 | 2.57 |
| DenseNet-121 | ✗ | 100.00 | 0.00 | 100.00 | 0.00 | 100.00 | 0.00 | 49.86 | 2.71 |
| DenseNet-121 | ✓ | 100.00 | 0.00 | 100.00 | 0.00 | 98.39 | 1.61 | 27.28 | 0.59 |
| DenseNet-161 | ✗ | 100.00 | 0.00 | 100.00 | 0.00 | 100.00 | 0.00 | 48.90 | 5.80 |
| DenseNet-201 | ✗ | 100.00 | 0.00 | 100.00 | 0.00 | 100.00 | 0.00 | 48.77 | 2.16 |
| ED | ✗ | 100.00 | 0.00 | 100.00 | 0.00 | 100.00 | 0.00 | 78.54 | 2.88 |
| MLP | ✗ | 0.00 | 0.00 | 0.00 | 0.00 | 0.00 | 0.00 | 0.00 | 0.00 |
| ResNet-101 | ✗ | 99.99 | 0.01 | 100.00 | 0.00 | 100.00 | 0.00 | 47.95 | 1.10 |
| ResNet-101 | ✓ | 100.00 | 0.00 | 100.00 | 0.00 | 98.19 | 1.81 | 40.73 | 1.81 |
| ResNet-152 | ✗ | 99.99 | 0.01 | 100.00 | 0.00 | 100.00 | 0.00 | 55.79 | 4.08 |
| ResNet-152 | ✓ | 99.99 | 0.01 | 99.99 | 0.00 | 94.67 | 5.33 | 36.82 | 2.98 |
| ResNet-18 | ✗ | 100.00 | 0.00 | 100.00 | 0.00 | 100.00 | 0.00 | 45.45 | 0.90 |
| ResNet-50 | ✗ | 100.00 | 0.00 | 100.00 | 0.00 | 100.00 | 0.00 | 55.98 | 8.86 |
| Swin-Base | ✗ | 99.92 | 0.04 | 99.84 | 0.01 | 99.86 | 0.08 | 29.86 | 1.06 |
| Swin-Tiny | ✗ | 99.93 | 0.02 | 99.80 | 0.05 | 99.07 | 0.93 | 29.23 | 0.52 |
| WideResNet | ✗ | 100.00 | 0.00 | 100.00 | 0.00 | 99.90 | 0.10 | 45.12 | 1.37 |

Table 33: $c = 2$

| Model | Pretrained | Train | | ID Val | | OOD Val | | Test | |
|---|---|---|---|---|---|---|---|---|---|
| | | AVG | SEM | AVG | SEM | AVG | SEM | AVG | SEM |
| ConvNeXt-Base | ✗ | 99.98 | 0.01 | 99.99 | 0.00 | 82.14 | 17.86 | 85.69 | 0.99 |
| ConvNeXt-Small | ✗ | 67.06 | 32.92 | 67.36 | 32.64 | 66.67 | 33.33 | 54.39 | 27.46 |
| ConvNeXt-Tiny | ✗ | 66.94 | 33.02 | 67.14 | 32.85 | 66.67 | 33.33 | 51.16 | 25.79 |
| DenseNet-121 | ✗ | 100.00 | 0.00 | 100.00 | 0.00 | 99.80 | 0.20 | 87.99 | 2.81 |
| DenseNet-121 | ✓ | 100.00 | 0.00 | 99.99 | 0.01 | 63.47 | 31.86 | 43.91 | 2.04 |
| DenseNet-161 | ✗ | 100.00 | 0.00 | 100.00 | 0.00 | 100.00 | 0.00 | 82.35 | 0.73 |
| DenseNet-201 | ✗ | 100.00 | 0.00 | 100.00 | 0.00 | 99.49 | 0.51 | 85.81 | 5.96 |
| ED | ✗ | 100.00 | 0.00 | 100.00 | 0.00 | 100.00 | 0.00 | 94.70 | 0.63 |
| MLP | ✗ | 0.00 | 0.00 | 0.00 | 0.00 | 0.00 | 0.00 | 0.00 | 0.00 |
| ResNet-101 | ✗ | 100.00 | 0.00 | 99.99 | 0.00 | 98.84 | 1.16 | 86.15 | 3.33 |
| ResNet-101 | ✓ | 100.00 | 0.00 | 99.98 | 0.00 | 85.94 | 14.06 | 45.01 | 10.02 |
| ResNet-152 | ✗ | 100.00 | 0.00 | 100.00 | 0.00 | 93.88 | 6.12 | 78.59 | 2.30 |
| ResNet-152 | ✓ | 100.00 | 0.00 | 100.00 | 0.00 | 70.06 | 29.94 | 56.03 | 1.01 |
| ResNet-18 | ✗ | 100.00 | 0.00 | 100.00 | 0.00 | 99.60 | 0.33 | 71.67 | 6.57 |
| ResNet-50 | ✗ | 100.00 | 0.00 | 100.00 | 0.00 | 99.54 | 0.46 | 81.29 | 2.70 |
| Swin-Base | ✗ | 99.95 | 0.01 | 99.76 | 0.02 | 81.44 | 18.56 | 71.49 | 2.51 |
| Swin-Tiny | ✗ | 99.90 | 0.03 | 99.66 | 0.06 | 67.37 | 32.63 | 66.74 | 3.14 |
| WideResNet | ✗ | 100.00 | 0.00 | 100.00 | 0.00 | 100.00 | 0.00 | 82.37 | 3.06 |

Table 34: $c = 3$

# F  Symmetries and invariances

The success of DNNs in vision tasks was initially driven by the insight that shift-invariance is a fundamental symmetry in the image domain. Based on this insight, convolutional layers were designed to satisfy shift-invariance by construction, and the incorporation of these layers significantly simplifies the optimization process in CNNs by restricting the search space to solutions that adhere to these symmetries. Indeed, for any sample $\mathbf{x} \in X$, if $f$ is shift-invariant, then $f(\mathfrak{g}.\mathbf{x}) = f(\mathbf{x})$ for any shift $\mathfrak{g} \in \mathfrak{G}$, which implies that observing either $\mathbf{x}$ or $\mathfrak{g}.\mathbf{x}$ is equivalent from the perspective of $f$. This kind of equivalence has two important implications: (i) there is no advantage in observing both, which makes $f$ inherently more sample efficient, and (ii) if $f$ correctly classifies the training sample $\mathbf{x}$, $f$ will automatically generalize to unseen test samples under the transformation $\mathfrak{g}.\mathbf{x}$, thus improving its generalization performance. Following a similar principle, Attribute Invariant Networks structurally embed **attribute invariance** to achieve compositional generalization.

# G  Proof of attribute invariances in gradient updates

The structure of AINs allows the optimization of each encoder $h_i$ to be be sensitive to group actions related to attribute $i$, while being invariant to group actions of any other attribute, as illustrated in the following theorem.

**Theorem G.1** (Attribute invariances in gradient updates). *Let $(\mathbf{x}, \mathbf{y})$ be a sample, and let $f_j(\mathbf{x})$ be an AIN's logit corresponding to attribute $j$. Then, for every group action $\mathfrak{g} \in \mathfrak{G}_j$:*

- *if $j \neq i$, then $\nabla_{h_i} \mathcal{L}(y_j, f_j(\mathbf{x})) = \nabla_{h_i} \mathcal{L}(y_j, f_j(\mathfrak{g}.\mathbf{x})) = 0$*

- *if $j = i$, then $\nabla_{h_i} \mathcal{L}(y_i, f_i(\mathbf{x})) \neq \nabla_{h_i} \mathcal{L}(y_i, f_i(\mathfrak{g}.\mathbf{x}))$*

*Proof.* (Part 1: $j \neq i$) Let us use the chain rule to compute the gradient of $f_j(\mathbf{x})$ w.r.t. the encoder $h_i$:

$$\nabla_{h_i} \mathcal{L}(y_j, f_j(\mathbf{x})) = \frac{\partial \mathcal{L}(y_j, g_j(\mathbf{z}_j))}{\partial g_j(\mathbf{z}_j)} \frac{\partial g_j(\mathbf{z}_j)}{\partial m(\mathbf{q}_j)} \frac{\partial m(\mathbf{q}_j)}{\partial h_i(\mathbf{x})}$$

Since $\mathbf{q}_j \perp h_i$, then $\frac{\partial m(\mathbf{q}_j)}{\partial h_i(\mathbf{x})} = 0$. As a result, $\nabla_{h_i} \mathcal{L}(y_j, f_j(\mathbf{x})) = 0$. Following the same procedure $\nabla_{h_i} \mathcal{L}(y_j, f_j(\mathfrak{g}.\mathbf{x})) = 0$.

(Part 2: $j = i$) For any $\mathbf{x}' = \mathfrak{g}.\mathbf{x}$, the corresponding ground truth label $y_i' \neq y_i$ by definition. As a result, $\nabla_{h_i} \mathcal{L}(y_i, f_i(\mathbf{x})) \neq \nabla_{h_i} \mathcal{L}(y_i, f_i(\mathfrak{g}.\mathbf{x}))$. $\qquad\square$

