# OpenReview forum: "Scalable Evaluation and Neural Models for Compositional Generalization"
_NeurIPS.cc/2025/Conference — NeurIPS 2025 poster_

### Official Review · Reviewer_qPCT · 2025-06-30

**Clarity:** 4
**Significance:** 4
**Originality:** 4
**Rating:** 5
**Confidence:** 3

**Summary:**

This paper unifies and extends existing compositional generalization evaluation protocols. The authors introduce a Compositional Similarity Index to measure the difficulty of generalization and use it to construct more principled dataset splits. They conduct extensive and comprehensive experiments to assess the compositional generalization capabilities of current visual backbones and further propose a new model.

**Questions:**

I have a question regarding line 97 of the paper. If the degree of freedom c=1, then test samples should share at most one attribute with the training set. However, in the example provided (Figure 2 a, top-left and bottom-left images), the two test images seem to share two factors. Doesn't this imply that c=2, rather than c=1? Could the authors clarify this?

**Ethical Concerns:**

["NO or VERY MINOR ethics concerns only"]

**Final Justification:**

I will keep my original rating. I think this paper is a good paper.

**Limitations:**

yes

**Quality:**

4

**Strengths And Weaknesses:**

Strengths:
1. The authors address key limitations in existing compositional generalization benchmarks and propose a unified evaluation methodology. This provides a more principled and practical way to assess existing approaches, which can significantly influence future research in this area.
2. The experimental evaluation is thorough and convincing, covering a wide range of model architectures and settings.

Weaknesses:
There are no major weaknesses.

---

> ### Author Rebuttal · Authors · 2025-07-31
>
> Thanks for your positive feedback regarding the potential of the proposed evaluation methodology for future works in the area of compositional generalization and the thoroughness of our experimental analysis (also highlighted by reviewers bTmd and QWmU).
>
> ---
>
> **Q1: Improved the formal definition of compositional similarity index.**
>
> Your observation regarding Line 97 is indeed correct. According to our former definition, the compositional similarity index for the example presented in Figure 2a would still be $c=2$ even when both yellow and green orthotopes are excluded from the training data.
> In fact, the $\max$ operator would automatically pick the examples adjacent to the split boundary (between train and test examples) to effectively maximize the number of shared attributes (e.g., train [orange, big, cube] and test [blue, big, cube]).
>
> However, this is not the intended behavior of $c$, which should instead measure the number of shared attributes between any training examples and the less similar example that shares _at least one attribute_.
> Note that the latter condition is fundamental, as otherwise it would be possible to find examples in the test partition that have no overlap with any fixed training example in some cases (e.g., train [orange, big, cube] and test [blue, small, cylinder]).
> For this reason, we cannot simply replace the $\max$ operator with a $\min$ in Equation 1.
>
> To solve this issue, we resort to a min/max definition of the compositional similarity index: $c(P_{train}, P_{test}) = \min_{y_2 \in P_{test}}  \max_{y_1 \in P_{train}}  \sum_{i=1}^I  1_{y_{1, i}=y_{2, i}}$.
>
> In this case, the $\min$ operation finds the maximum distance between train and test examples, while the $\max$ operation allows us to avoid the shortcut solution described above.
>
> ---
>
> Once again, thanks for spotting this issue in our definition of the $c$ index and contributing to the improvement of our work!
> We are available for any further questions or clarifications.

---

> > ### Comment · Reviewer_qPCT · 2025-08-03
> >
> > Thank you for your thoughtful feedback. After consideration, I continue to view this work as a valuable contribution and therefore maintain my rating and support its acceptance.

---

> > > ### Author Response · Authors · 2025-08-03
> > >
> > > Thanks for acknowledging our rebuttal. We are glad to hear that, after further consideration, you still consider it a valuable contribution and support its acceptance!

---

### Official Review · Reviewer_bTmd · 2025-06-30

**Clarity:** 3
**Significance:** 3
**Originality:** 3
**Rating:** 5
**Confidence:** 4

**Summary:**

The authors formalise visual compositional generalisation through a well-formulated dataset curation technique to construct a standard in visual compositional generalisation. The main idea is constructing a high-dimensional 'orthotopic' geometry composed of dataset partitions, then partitioned based on the complexity level of the measured compositional generalisation capabilities. The authors emphasise the efficiency of using orthotopic evaluation and its disentangled nature. After building this benchmark, the authors evaluate different architectures, pre-trained models, and more, training more than 5000 models. The authors conclude with constructing a new architecture, specifically endowed with inductive biases to improve visual compositional generalisation capabilities, and show its good performance and efficiency compared to its counterparts.

**Questions:**

- I am willing to raise my score if the authors can convince me regarding the novelty of their work, or highlight a different aspect that I might have missed.

**Ethical Concerns:**

["NO or VERY MINOR ethics concerns only"]

**Final Justification:**

My main concern regarding the acceptance of the paper was its originality compared to Okawa et al. (2023). Nonetheless, the authors prepared a nice explanatory rebuttal comparing their work with the aforementioned paper and succinctly highlighted the novel aspects of their paper. Therefore, I support the acceptance of the paper and I am willing to increase my score from 'borderline accept' to 'accept'.

**Limitations:**

yes

**Quality:**

3

**Strengths And Weaknesses:**

### **Strengths**

1. Nice experimental setup, ablation studies, and case studies
2. The flow of the paper is remarkable, understandable examples, showing clear attention to details
3. Training 5000 models and exploring their compositional generalisation skills with a well-crafted dataset is an important contribution to the literature
4. The newly introduced architecture is valuable in terms of its performance, efficiency, and increasing the number of works intersted in disentangled representations

### **Weaknesses**

1. The main contribution of the paper is already achieved in an uncited paper published in NeurIPS 2023 [1]. This paper already explores the idea of concept distance, which is equivalent to the compositional complexity proposed in this paper. Although the authors add a few novel details on top of the already present idea, the novelty of the paper is reduced.


#### **References**

1. Maya Okawa, Ekdeep Singh Lubana, Robert P. Dick, and Hidenori Tanaka. 2023. Compositional abilities emerge multiplicatively: exploring diffusion models on a synthetic task. In Proceedings of the 37th International Conference on Neural Information Processing Systems (NIPS '23). Curran Associates Inc., Red Hook, NY, USA, Article 2182, 50173–50195.

---

> ### Author Rebuttal · Authors · 2025-07-31
>
> Thanks for your positive feedback on our work! We are grateful for your appreciation of the empirical setup (also highlighted by reviewers qPCT and QWmU) and the flow of the paper.
> We are also glad that you considered our overall exploration of compositional generalization and the specific AIN architecture as valuable contributions to the field (as also noted in various ways by all the other reviewers).
>
> ---
>
> **Q1: Comparison with related work from Okawa et al., 2023.**
>
> Thanks for providing the reference to the work of Okawa et al. (★); we were not aware of this line of research, which is indeed extremely relevant to our submission.
> The idea of concept distance proposed in their work is, in fact, equivalent to the compositional similarity index considered in our submission, as they both quantify the (dis)similarity between training and testing concept classes (i.e., task-relevant factors) by counting the number of differing factors' values.
>
> While starting from the observation and formalization of the same phenomenon, the **two works take significantly different paths in terms of contributions, development of the idea, empirical evaluation, and conclusions.**
> For this reason, **we respectfully disagree that the main contribution of the paper is already achieved in ★**, as it does not challenge the novelty of the main contributions claimed in Lines 35-48 (efficient orthotopic compositional evaluation, wide empirical evaluation, and AINs).
>
> The major differences between the two investigations, representing in our opinion a significant advancement in terms of novelty of the paper, are:
>
> 1. **Ladder of compositional generalization, relation between compositionality and disentanglement.**
> We establish a connection between different types of generalization proposed in previous works (extrapolation, compositional generalization, interpolative generalization, in-distribution generalization), showing that they correspond to evaluating generalization with different levels of compositional similarity.
> In the case of compositional generalization, we further identify a differentiation between the evaluation performed with unitary concept distance ($c=1$) and all the other distances ($1<c<I$).
> The former tests the models' ability to perfectly disentangle unitary concepts and to recombine them arbitrarily at test time, while the latter only strictly tests the ability to compositionally recombine information (whose entanglement is linearly increasing with $c$).
> ★ does not delve into any of these topics.
>
> 2. **Computationally efficient and generalizable evaluation framework.**
> Starting from the idea of compositional similarity, we propose a procedure (dubbed orthotopic evaluation) to efficiently evaluate compositional generalization in artificial intelligence systems.
> This evaluation procedure, in particular, explicitly exposes the compositional similarity as a potential degree of freedom in the evaluation framework (among others reported in Appendix A1), allowing for testing compositional behavior for varying degrees of this parameter.
> Additionally, the proposed procedure achieves a significant speed-up compared to other strategies used in previous works in the field, reducing the evaluation complexity from combinatorial to linear in the number of generative factors.
> Concept Graph, introduced in ★, represents a minimal prototype of an experimental framework, lacking a general and scalable algorithmic procedure to efficiently extend its applicability to arbitrary supervised learning datasets at scale.
>
> 4. **Proposal of a new architectural blueprint.**
> Inspired by the tension between generalization capabilities and the number of parameters and computation observed in our empirical evaluation, we further propose Attribute Invariant Networks (AINs). This novel architectural blueprint achieves a new Pareto frontier in the compositional generalization tasks studied in the paper, significantly decreasing the number of trainable parameters compared to ED models and improving the compositional generalization compared to monolithic architectures.
>
> 5. **Different focus on the empirical evaluation results.**
> While both ★ and us observe the same behavior (i.e., the ability to generalize compositionally is inversely related to the compositional distance between training and testing), we make this observation at different scales.
> ★ is more focused on the convergence behavior _within a single training episode_, whereas we study and report compositional generalization across _multiple training episodes_.
> Effectively, we flatten the entire training procedure to a single generalization score (picked according to our selection metric, ablated in Appendix B4), picking only the "best" model out of the entire training run.
> Arguably, our design choice allows us to capture wider trends that emerge on the best models for each architecture and test more systematically their practical limits in settings that require showing compositional behaviors.
> Furthermore, we do not provide any form of conditioning to most of the models (except for ED and our proposed architectures, AIN), while the generative models studied in their experiments are explicitly conditioned to develop more compositional structure by the textual prompts used in the experiments.
>
> 6. **Scale of the experimental setting.**
> Compared to ★, our work features a significantly wider and more comprehensive set of empirical results.
> We consider an extensive range of datasets (6, compared to 2 in ★) covering a wide spectrum of data complexity, from purely synthetic (e.g., dSprites) to real-world (e.g., MPI3D-real).
> This was instrumental in allowing us to extend our analysis to wider ranges of compositional similarity (up to $c=6$ in our experiments, compared to max $c=4$ in ★), proving that the observation on relatively simple, artificial dataset (e.g., simple monochromatic shapes in dSprites) does effectively extend to more complex, real-world datasets (e.g., MPI3D).
> The wider selection of datasets and dataset-specific generative factors also implicitly increases the robustness of our observations.
> On the model side, we test more than 20 modern supervised models belonging to different families (Transformer-based, convolutional, simple MLPs, etc.), while ★ only explored a single diffusion model.
> Our evaluation also features ablations on different components of the studied models (e.g., different readouts, an increased number of optimization steps, different activation functions, and different selection metrics, which have been mostly included as appendices to the main paper).
>
> 7. **Retrospective evaluation of previous works.**
> Formalizing the concept of compositional similarity allowed us to retrospectively re-evaluate the results of previous works in the field, highlighting substantial inconsistencies in their experimental designs that hindered direct comparability between them.
> ★, on the other hand, does not engage in a similar re-evaluation of previous works that explored compositional generalization.
>
> Of course, we will include ★ in our related work section, along with a succinct summary of the differences between our work and theirs reported above.
> We are also going to remove the sentence
>
> > "To the best of our knowledge, a systematic characterization of this degree of freedom, as well as its formal definition in the first place, does not exist in the literature yet."
>
> in Lines 90-92 from our manuscript, adding a reference to ★ instead.
>
> ---
>
> Once again, thanks for providing the reference to ★ that would have been otherwise missed in our discussion, hence contributing to making our work complete!
> We are available for any further discussion or clarification.

---

> ### Comment · Reviewer_bTmd · 2025-08-03
>
> Thanks to the authors for this comprehensive and highly illuminating rebuttal. I agree with all of their points contrasting their work with Okawa et al. (2023). Thus, I increase my score and support the acceptance of the paper.

---

> > ### Author Response · Authors · 2025-08-03
> >
> > Thanks for acknowledging our rebuttal! We are glad to hear that you agree with us on the novelty of our work and that you also confidently support the acceptance of our work.

---

### Official Review · Reviewer_QWmU · 2025-07-01

**Clarity:** 3
**Significance:** 2
**Originality:** 3
**Rating:** 5
**Confidence:** 4

**Summary:**

This paper addresses the problem of compositional generalization in supervised vision models by introducing a formal compositional similarity index **c** i.e. the maximum number of shared factors between any training and test sample. Authors propose an orthotopic evaluation procedure that generates compositional OOD splits via iterative projection and pruning. The authors validate this framework by training models across different datasets and empirically confirm a “difficulty ladder” from extrapolation (c=0) to in-distribution (c=I). They also introduce Attribute Invariant Networks (AINs), which enforces invariance to irrelevant attribute transformations. Overall, this work offers an interesting analysis and simple approach to improve compositional generalization in vision models.

**Questions:**

1. Can the authors show that orthotopic splits and AINs can work on natural‐image datasets (e.g., COCO, ImageNet) without explicit factor annotations, and how well do they transfer to richer, uncontrolled data?
2. How can your framework and AINs be adapted when generative‐factor labels are noisy, incomplete, or unavailable. Can c be estimated or optimized under such weak or semi-supervised settings?
3. How did the empirical study of the compositional similarity index **c** and the “difficulty ladder” inform your specific architectural choices in AIN?

**Ethical Concerns:**

["NO or VERY MINOR ethics concerns only"]

**Final Justification:**

After considering the rebuttal and discussion, I am increasing my score.

Resolved issues: The authors clarified how the empirical analysis directly informed the design of AINs, which strengthens the link between the study and the proposed method. They also ran additional experiments with noisy labels and missing factors, which demonstrate robustness of the orthotopic evaluation. I agree with the authors that synthetic datasets are particularly well-suited to reveal failures under controlled conditions, and I therefore no longer view their use as a limitation of this work. I was also initially concerned about overlap with Okawa et al. (NeurIPS 2023), but the authors’ clarified the distinctions in scope and contributions.

Remaining concerns: While AINs are more efficient than ED, their linear scaling with the number of attributes could still hinder use in domains with very large factor counts. In terms of novelty, there is still some overlap in high-level idea with Okawa et al.

Overall: The paper is technically solid, makes a clear and useful contribution, and has potential to influence future work.

**Limitations:**

No, the authors don't explicitly discuss the limitation of their analysis and method.

**Paper Formatting Concerns:**

No paper formatting concerns.

**Quality:**

3

**Strengths And Weaknesses:**

## Strengths

1. **Interesting Formulation** - The paper provides a rigorous, unified formalism for compositional evaluation by defining 𝑐 and offering an orthotopic evaluation algorithm that projects and prunes data across all 𝑐-dimensional subspaces, it avoids the combinatorial blow-up.
2. **Scale of Experimentation** - Over 5,000 training experiments on diverse architectures and six benchmarks lends strong statistical support to the paper’s findings.
3. **Insightful Findings** - The empirical results verify that no model generalizes at c=0 (extrapolation) but nearly all succeed at c=I (in-distribution), with a monotonic accuracy ascent in-between, validating the proposed “difficulty ladder”.
4. **Proposed Model** - The AIN architecture is a simple approach that enforces attribute-wise invariance in gradients, it promotes disentanglement without the expensive parameter cost of ED models. Moreover, AIN achieves substantial OOD gains over monolithic backbones with minimal parameter overhead.

## Weaknesses

1. **Disconnect between analysis and method design** - The paper's first part’s detailed empirical study of compositional difficulty does not clearly motivate specific choices in AIN’s architecture. It's unclear which insights from the analysis or the “difficulty ladder” directly informed the design choices in AINs.
2. **Synthetic datasets only** - All evaluations use controlled, synthetic benchmarks (dSprites, I-RAVEN, Shapes3D, CLEVR, Cars3D), leaving the question open whether orthotopic splits and AINs transfer to complex natural-image data (e.g., COCO, ImageNet) with richer variability and no annotations for individual factors.
3. **Fully supervised assumption** - Relies on known, clean generative‐factor labels. In many domains these are latent or noisy, and the paper does not address how to estimate or adapt c under such cases.
4. **Scalability of attribute count** - Although AINs have far lower overhead than ED models, parameter and computational costs still depend on the number of attributes, which may become prohibitively large for real-world images involving many factors.

---

> ### Author Rebuttal · Authors · 2025-07-31
>
> Thank you for your positive feedback on our work!
> We sincerely appreciate your recognition of the empirical setup (an aspect also highlighted by reviewers qPCT and bTmd) as well as the insightful findings it enabled.
> We are also pleased that you found both our broader investigation into compositional generalization and the specific AIN architecture to be meaningful contributions to the field, a sentiment echoed in various ways by the other reviewers as well.
> Furthermore, we appreciate the constructive observations and questions raised in the Weaknesses and Questions sections, which we will address in the following responses.
>
> ---
>
> **W1, Q3: How did the empirical study inform architectural choices in AIN?**
>
> In the first part of the paper, we formalize the orthotopic evaluation framework and leverage it to perform a large-scale evaluation of different models.
> One of the main observations that emerged is that ED is consistently the best-performing model in every dataset and step of the proposed ladder of compositional generalization, often improving the compositional generalization over monolithic models by large margins.
> The reason behind this behavior is that ED trains independent architectures for each attribute, thus achieving high accuracy in compositional tasks.
> However, this comes at the cost of lots of parameters (as explained in Lines 247-256)
>
> This latter limitation of ED-style architectures inspired the design of AIN architectures. In particular, we asked ourselves: is it possible to achieve levels of compositional generalization comparable to ED without incurring its significant parameter overhead (i.e., retaining a model size which is closer to monolithic architectures)?
> In practice, we show that this is possible in AINs.
> How? The encoder $h_i$ is like ED (each encoder is attribute-specific and independent of other attributes), while the meta-model is like in efficient monolithic models and shared between all attributes.
>
> We agree that the transition between the two parts of the paper could be improved; we will include a summary of this discussion in the final version of the paper.
>
> ---
>
> **W2, Q1: Use of synthetic datasets.**
>
> The use of synthetic datasets is, to this date, the de facto standard in the field of compositional generalization in vision [paper references 5-10, 32-42] for different reasons:
>
> - It offers the possibility to operate in a fully controlled environment, limiting the impact of exogenous factors on the empirical observations while having the possibility to stress-test compositionality.
> - SOTA architectures still struggle to effectively model the compositional nature of the generative factors in these simple, synthetic datasets, despite their simplistic nature (as our evaluation shows).
>
> Furthermore, we remark that the synthetic datasets used in our work are already on the more complex side of the spectrum of datasets normally used (compared, for instance, to simpler datasets like 2D Gaussian Bump Generation). One of the datasets that was used in the evaluation, MPI3D-real, is effectively a real-world image dataset (extremely controlled, as required condition for a principled evaluation, but real). CLEVR, despite being synthetic, also takes into account different non-trivial factors in the generation of the images (light, perspective, etc.).
>
> However, the extension of the proposed orthotopic evaluation framework to natural-image datasets would be straightforward, as long as categorical labels for the factors for which we want to measure compositionality are available (note that these will _always_ be necessary to properly evaluate compositionality, see also the next response).
> Unfortunately, setting up experiments for natural‐image datasets like COCO or ImageNet would require more time than that available to us during the rebuttal period, other than being outside of the scope of this work for the reasons mentioned before.
> To make a first step in this direction (also hinted by reviewer tLy1), we design and run an additional experimental validation on the proposed orthotopic evaluation and ladder of compositional generalization to test their robustness under more realistic conditions (noise on the labels, unlabeled generative factors).
> Considering the limited time frame available for the experiments, we modified one of the experiments (on MPI3d-real) by perturbing 2 attribute labels at random in 10% of the training samples to simulate noisy annotations (i.e., re-introducing $\epsilon_y$ in the experimental setup).
> Additionally, we also removed the annotations of two task-relevant factors (color, x-position) from the training data, effectively morphing them from task-relevant ($G$) to task-irrelevant ($O$) generative factors.
> With these changes, we aim to achieve a more realistic, yet fully controllable, setting with unknown latent generative factors and random label noise, as would be the case in a real-world dataset.
>
> The results of these experiments are reported in the following table.
>
> |**Model**|**c=0**|**c=1**|**c=2**|**c=3**|
> |-|-|-|-|-|
> | Convnext-small | 0.0 | 32.9| 40.1| 46.9|
> | ResNet50       | 0.0 | 38.1| 57.8| 75.2|
> | Swin-tiny      | 0.0 | 29.6| 45.6| 49.0|
> | AIN            | 0.0 | 46.8| 67.6| 83.7|
> | ED             | 0.0 | 71.1| 67.6| 83.2|
>
> We can observe that, even in this noisy setting closer to real-world scenarios our main results hold: the relation between compositional similarity and evaluation difficulty, the superiority of architectures that explicitly incorporate some inductive bias toward compositionality in their architecture (e.g., ED and AIN), and the strict increase in generalization accuracy achieved by ED and AINs compared to monolithic architectures.
>
> ---
>
> **W3, Q2: Fully supervised assumption.**
>
> We agree to some extent that the assumption of fully known generative factors is rather strong.
> However, we argue that:
>
> - This is a fairly standard and formally sound assumption in most of the previous works on compositional generalization [e.g., paper references 5-10, 32-42]. Having prior knowledge of at least a subset of the generative factors (or, more generally, of the concepts of which we are testing the composition) is quite essential to evaluate compositional generalization, which would otherwise become an ill-posed problem without them.
>
> - The use of the generative factors labels as a direct training signal for the models allows us to operate in a fully controlled environment, limiting the impact of exogenous factors on the empirical observations while having the possibility to stress-test compositionality.
> This setting should represent the easiest experimental setup for deep neural networks, which are provided with all the necessary and sufficient knowledge to solve the task.
> In this light, our results are quite surprising as they show that even in this extremely simple setting, state-of-the-art networks fail.
> Therefore, before diving into more complex scenarios (e.g., relaxing the assumption of knowing the generative factors), we argue that increasing our understanding of this failure and finding possible remedies that help alleviate it still represents a needed and valuable contribution to the field.
>
> To ablate the presence of unlabelled, latent factors, we excluded from the experiment presented in the previous point 2 of the original generative factors in MPI3D from the labels available to the models.
> The results show that the presence of these kinds of latent factors does not significantly influence the outcomes of our evaluation.
>
> ---
>
> **W4: Scalability of attribute count.**
>
> While it's true that AINs still scale linearly with the number of attributes, they remain far more efficient than ED models and far more accurate than monolithic models.
> This balance between accuracy and efficiency makes AIN the de facto Pareto optimal between SOTA models and applicable in the majority of applications (the overhead of the disentangled encoder modules would not represent an issue, depending on the size of the models, when the number of generative factors is in the order of hundreds or even thousands).
> Exploring sub-linear alternatives to AINs could, however, be an interesting line of research for future work.
>
> ---
>
> **Limitations**
>
> We add the following Limitation section:
>
> > The methods explored in this work -- orthotopic evaluation and AINs -- rely on the assumption that (at least a subset of) the generative factors of variation are accessible. This is analogous to how evaluating classification accuracy or training supervised models requires access to ground-truth labels. Additionally, while AINs, like ED, scale linearly with the number of attributes, they do so with a significantly lower constant factor, making them substantially more efficient in practice. Finally, we emphasize that neither the attribute-invariant gradient updates used by AINs nor the explicit disentanglement enforced by ED guarantee perfect disentanglement or complete independence among attributes. Rather, they serve as effective mechanisms to promote these properties.

---

> ### Comment · Reviewer_QWmU · 2025-08-02
>
> Thank you for the rebuttal and for running additional experiments. The clarifications helped address some of my earlier concerns.
>
> I found the explanation of how the empirical analysis directly inspired the design of AINs to be helpful. Making this connection clearer in the paper would strengthen the overall narrative. I also appreciate the extra noisy-label experiment, which demonstrates that your framework and AINs remain effective under more realistic conditions. That said, I continue to view the dependence on synthetic datasets and full supervision as an important limitation for broader applicability. While I understand this is consistent with current practice in the field, the paper would benefit from a more explicit discussion about transfer to complex, uncontrolled datasets (e.g., ImageNet, COCO). Overall, the rebuttal strengthens my confidence in the technical soundness of the work.

---

> ### Author Response · Authors · 2025-08-03
>
> Thanks for acknowledging our rebuttal! We are glad that the additional explanation on the connection between the empirical analysis and the design of AINs was helpful; it will indeed be included in the final version of the manuscript. We are also pleased that the additional experiment with noisy labels increased your confidence in the overall experimental setting.
>
> On the other hand, we understand your point on the use of synthetic datasets and full supervision as possible limiting factors to the applicability of the findings and methods. However, we remark that:
>
> - **Recent studies** [1] **have already shown that the transfer from synthetic to real in very similar settings** (disentanglement learning) is possible, effectively increasing our confidence in arguing that the transfer to complex, uncontrolled datasets could likely work in our setting as well.
>
> - **Using synthetic datasets to prove failure is, in our opinion, a stronger result than using natural datasets.** While in natural datasets the failure might be the result of an interplay between different causal factors or be attributed to completely different causes, this is not the case for synthetic datasets. The failure only happens _because_ of the compositional nature of the evaluation.
>
> - **The proposed evaluation framework and AINs can directly generalize to natural datasets without requiring any extension or adaptation.** As you point out, however, they both critically depend on annotations. One possible avenue to generalize these methods beyond a fully-annotated dataset could be the way paved by **label-free approaches** [2, 3]: using multimodal language models (instead of human annotators) to a) find the relevant factors of variation and b) annotate the input samples according to these factors.
>
> We agree with you on the importance of discussing these additional topics in the manuscript; for this reason, we plan to include a summary of them in the final version of the paper.
>
> [1] Transferring disentangled representations: bridging the gap between synthetic and real images, Dapueto et al. NeurIPS 2024.
>
> [2] Label-free Concept Bottleneck Models, Oikarinen et al., ICLR 2023.
>
> [3] Language in a Bottle: Language Model Guided Concept Bottlenecks for Interpretable Image Classification, Yang et al. CVPR 2023.

---

> > ### Comment · Reviewer_QWmU · 2025-08-05
> >
> > I acknowledge the points made on the use of synthetic datasets. I particularly agree that synthetic datasets are valuable for revealing failures under controlled conditions, and in this sense I do not consider their reliance on them a limitation of the paper. While extending to natural datasets remains valuable future work, the current scope of the work is sufficient and appropriate.
> >
> >
> > Also, I was initially concerned about the similarity to Okawa et al. (NeurIPS 2023). While the two works share high-level goals, the authors’ response to another reviewer clarified the distinctions in scope and contributions. I encourage the authors to incorporate this comparison explicitly in the final related work section.

---

> > > ### Author Response · Authors · 2025-08-06
> > >
> > > Thank you for eventually agreeing with us about the use of synthetic datasets in our work. We completely agree with you on the fact that extending our work to natural datasets represents valuable material for future work.
> > > We will include a summary of this fruitful discussion (along with the comparison with Okawa et al. 23 in related work) in the final version of the paper.

---

### Official Review · Reviewer_tLy1 · 2025-07-02

**Clarity:** 3
**Significance:** 3
**Originality:** 2
**Rating:** 4
**Confidence:** 3

**Summary:**

Compositional generalization is a key aspect for improving the generalization capability of AI models, but it remains underexplored in how to design principled and scalable evaluation protocols. This makes it challenging to develop models that can systematically generalize to unseen combinations of known concepts. To address these challenges, this paper introduces a unified framework for evaluating compositional generalization, featuring a compositional similarity index and the orthotopic split generation algorithm, which enables efficient and tractable creation of OOD splits with controllable difficulty. Extensive experiments on various vision tasks demonstrate the effectiveness of this evaluation framework. In addition, the paper proposes Attribute Invariant Network (AIN) — a new neural architecture that enforces attribute invariance by design, achieving strong compositional generalization with significantly less parameter overhead compared to fully disentangled models.

**Questions:**

- **Q1.** In Algorithm 1, is the ${X_\text{train}}$ in line 5 meant to be ${X_\text{train}^{C}}$?
- **Q2.** How might the orthotopic evaluation ladder and the compositional similarity index extend to domains where task-relevant generative factors are latent or only partially observable? Would the proposed framework remain tractable and robust under these more realistic conditions?

**Ethical Concerns:**

["NO or VERY MINOR ethics concerns only"]

**Final Justification:**

The authors have provided thorough explanations for each question, which significantly helped clarify the paper's contributions. In particular, I appreciate the clarification regarding the assumption of fully known generative factors. I would also like to offer some additional comments in response to the rebuttal.

Overall, the rebuttal clarified key concerns and strengthened my confidence in the paper’s contributions. I have accordingly raised my score.

**Limitations:**

Some of the limitations are discussed in the Weaknesses and Questions sections.

**Paper Formatting Concerns:**

I don't see any major formatting issues in this paper.

**Quality:**

2

**Strengths And Weaknesses:**

### **Strengths**

- **S1.** The paper clearly motivates the practical importance of compositional generalization and identifies concrete shortcomings in existing evaluation protocols.
- **S2.** It introduces the compositional similarity index to systematically control task difficulty, along with an efficient orthotopic split generation algorithm that enables scalable and controllable evaluation settings.
- **S3.** The experimental results demonstrate that the proposed evaluation framework is well-designed and aligns closely with the authors’ stated motivation.
- **S4.** The proposed AIN architecture achieves strong compositional generalization while requiring significantly fewer additional parameters compared to fully disentangled models.

### **Weaknesses**

- **W1.** The paper assumes that the task-relevant generative factors are fully known in advance. However, in real-world scenarios, it is often difficult to identify these factors, and AINs require labeled generative factors for training. This may significantly limit their practical applicability.
- **W2.** There has been a growing body of research on compositional generalization in vision, including recent work on object-centric representation learning that also considers attribute disentanglement [1, 2]. A more thorough discussion comparing AIN to these related directions would strengthen the paper.
- **W3.** ED and AIN adopt a modular network structure that implicitly assumes each module specializes in a specific generative factor. Analyzing how well each module corresponds to the specific generative factors in practice would provide valuable insights and help clarify how AIN achieves its compositional behavior.
- **W4.** Although the paper mentions training over 5,000 state-of-the-art models multiple times, the detailed results and concrete outcomes of these large-scale experiments are not clearly presented.

[1] Neural Systematic Binder, ICLR 2023

[2] Neural Language of Thought Models, ICLR 2024

---

> ### Author Rebuttal · Authors · 2025-07-31
>
> Thanks for your insightful review of our work!
> We are grateful for your positive feedback on the validity of the evaluation framework and AIN architecture, closely aligned with that of all the other reviewers.
> We also appreciate the constructive observations and comments featured in the Weaknesses and Questions sections; we will address them individually in the following lines.
>
> ---
> **W1: Known generative factors assumption.**
>
> We agree to some extent that the assumption of fully known generative factors is rather strong.
> However, we argue that:
>
> - This is a fairly standard and formally sound assumption in most of the previous works on compositional generalization [e.g., paper references 5-10, 32-42]. Having prior knowledge of at least a subset of the generative factors (or, more generally, of the concepts of which we are testing the composition) is quite essential to evaluate compositional generalization, which would otherwise become an ill-posed problem without them.
>
> - The use of the generative factors labels as a direct training signal for the models allows us to operate in a fully controlled environment, limiting the impact of exogenous factors on the empirical observations while having the possibility to stress-test compositionality.
> This setting should represent the easiest experimental setup for deep neural networks, which are provided with all the necessary and sufficient knowledge to solve the task.
> In this light, our results are quite surprising as they show that even in this extremely simple setting, state-of-the-art networks fail.
> Therefore, before diving into more complex scenarios (e.g., relaxing the assumption of knowing the generative factors), we argue that increasing our understanding of this failure and finding possible remedies that help alleviate it still represents a needed and valuable contribution to the field.
>
> - AINs are a valuable contribution that represents the best solution (Pareto optimal) in scenarios requiring smaller models and high compositional generalization. Furthermore, the failure of monolithic architectures represents an indication that an additional inductive bias is necessary for the model to be able to generalize well. ED models integrate this bias by disentangling the parameters between parameters, but are inefficient and unscalable. AINs retain the same inductive bias, but apply it to only a fraction of the parameters of the network. In future work, the number and depth of the attribute-specific modules could also be inferred dynamically (as a kind of fast-weight programmer) and possibly extended to semi-supervised or unsupervised settings.
>
> ---
>
> **W2: Relations with object-centric attribute disentanglement.**
>
> Thank you for pointing out related works on object-centric representation learning; we agree that this line of work has made significant strides in compositional generalization.
>
> Firstly, SysBinder and NLoTM are mostly concerned with single-object compositional generalization, while those systems are mainly designed for multi-object factor decomposition.
> The most relevant part (for our purpose) of their architecture is the factor binding module, which is extremely similar to the two variants of the ED architecture investigated in this work.
> This module is characterized by different RNN (GRU+MLP) that have different _attribute-specific_ parametrization (shared among slots).
> These separate sub-networks use discrete codebooks in the case of NLoTM (conceptually equivalent to our ED-FPE model) or continuous representations in the case of SysBinder (conceptually equivalent to our ED-Linear model).
> Considering this functional equivalence between their factor-binding module and our ED architecture, we omitted these two baselines from a direct comparison.
>
> However, in light of this similarity, we believe that the same lessons and observations gained with AINs as efficient ED alternatives could directly transfer to their setup.
> In practice, these two directions could benefit from each other and, at some point, be integrated into a single framework: a slot-based model where the factor decomposition is performed by AIN-style models to reduce the parameter overhead while guaranteeing strong disentanglement and compositional generalization.
>
> We will add these important references and expand the related discussions in the final version of the paper.
>
> ---
>
> **W3: How do AIN/ED modules correspond with generative factors?**
>
> ED and AIN do not _assume_ the specialization of each module in a specific attribute; rather, this is integrated implicitly in their architecture through specific design choices.
> In ED, each attribute-specific DNN is trained on a single attribute and is structurally separated from the others.
> In AIN, on the other hand, the independence of the attribute-specific components is guaranteed by the invariances in gradient updates (Theorem 4.2, which theoretically explains "how AIN achieves its compositional behavior").
>
> If possible, could you clarify what exactly is meant by "analyzing how well each module corresponds to the specific generative factors in practice"?
> Is there any specific experiment that we could add to the paper to better clarify this?
>
> ---
>
> **W4: Unclear detailed results and outcomes of large-scale experiments.**
>
> The detailed results of the 5,000 training runs are summarized in Table 1 (for AINs) and Figure 3 (for all the other models), and the most important and concrete takeaways (answering our key research questions R1-R3) are discussed in Sections 4.2 and 3.3, respectively.
> However, we also include much more fine-grained results (for different $c$ values, datasets, models, and even attribute-wise) in Appendices C and D.
>
> Is there any particular paragraph/outcome that is unclear? And why?
>
> ---
>
> **Q1: Algorithm 1, is $X_{train}$ meant to be $X_{train}^C$?**
>
> No, $X_{train}$ is correct. In Line 5, the algorithm prunes out of the training set those samples that belong to the evaluation split because they contain a certain combination of generative factors.
> This is effectively achieved by removing from the training set (set difference) the intersection between the projected training set $X_{proj}$ with the result of `exclusion`.
> The latter computes some attribute-wise threshold according to the framework's parameters (e.g., percentage of the space excluded from the training, position of the excluded orthotopes, etc.) and returns the samples that violate those thresholds.
>
> For instance, consider the simple dataset with three generative factors, $(g_1,g_2,g_3)$, defined as `X = [(0, 3, 2),(2, 4, 0),(3, 4, 2),(1, 1, 1),(1, 3, 2)]`.
>
> Assume that the projection subspace is $(g_1,g_2)$, and the thresholds in this space are $t_1=2,t_2=1$.
> Then, `exclude` would return all possible evaluation samples (given by the cartesian product of the values $g_i\geq t_i$).
> Then, $X_{proj} \cap \text{exclusion}(s) = \{ (2, 4, 0), (3, 4, 2)\}$, which would be removed from the training data (and automatically become part of its complement, the evaluation data).
>
> The confusion regarding $X_{train}$ might stem from the fact that, right now, the behavior of `exclusion` is not sufficiently documented in the manuscript.
> We will adjust this and include more details in a final version of the paper, in order to improve the readability and clarity of the split generation pseudocode.
>
> ---
>
> **Q2: How eval framework extend to settings with latent generative factors?**
>
> As also mentioned in the response to W1, evaluating compositional generalization in domains where the task-relevant generative factors are latent or only partially observable is not possible, as the evaluation problem would become ill-posed.
> Some workarounds might be possible, for instance, when the solution of the downstream task strictly requires learning compositional representations of the input (without shortcuts or memorization tricks of any kind).
> In such cases, the performance on the task could represent a reliable indicator of a compositionality behavior of the model.
> However, this is outside of the scope of this work.
>
> The second part of your question is indeed interesting and inspired us (along with other prompts from QWmU01) to design an additional experiment to test the robustness of orthotopic evaluation under more realistic conditions (noise on the labels, unlabeled generative factors).
> Considering the limited time frame available for the experiments, we modified one of the experiments (on MPI3d-real, a dataset that already consists of real pictures) by perturbing 2 attribute labels at random in 10% of the training samples to simulate noisy annotations (i.e., re-introducing $\epsilon_y$ in the experimental setup).
> Additionally, we also removed the annotations of two task-relevant factors (color, x-position) from the training data, effectively morphing them from task-relevant ($G$) to task-irrelevant ($O$) generative factors.
> With these changes, we aim to achieve a more realistic, yet fully controllable, setting with unknown latent generative factors and random label noise, as would be the case in a real-world dataset.
>
> The results of these experiments are reported in the following table.
>
> |**Model**|**c=0**|**c=1**|**c=2**|**c=3**|
> |-|-|-|-|-|
> | Convnext-small | 0.0 | 32.9| 40.1| 46.9|
> | ResNet50       | 0.0 | 38.1| 57.8| 75.2|
> | Swin-tiny      | 0.0 | 29.6| 45.6| 49.0|
> | AIN            | 0.0 | 46.8| 67.6| 83.7|
> | ED             | 0.0 | 71.1| 67.6| 83.2|
>
> We can observe that, even in this noisy setting closer to real-world scenarios, our main results hold: the relation between compositional similarity and difficulty, the superiority of architectures that explicitly incorporate some inductive bias toward compositionality (e.g., ED and AIN), and the strict increase in generalization accuracy achieved by ED and AINs compared to monolithic architectures.
>
> ---
>
> Once again, thanks for the constructive feedback and comments on our work.
> We are available for any further questions or clarifications.

---

> > ### Comment · Reviewer_tLy1 · 2025-08-03
> >
> > Thank you for the rebuttal and detailed clarifications!
> >
> > The authors have provided thorough explanations for each question, which significantly helped clarify the paper's contributions. In particular, I appreciate the clarification regarding the assumption of fully known generative factors. I would also like to offer some additional comments in response to the rebuttal.
> >
> > **Regarding W3**:
> >
> > W3 raised a concern about whether each module of ED or AIN specializes in a specific generative factor. Based on the authors' response, I now agree that this point does not constitute a weakness of the paper.
> >
> > Moreover, while AIN does not assume module-wise specialization by design, I believe its architecture may still encourage such specialization. From my perspective, a promising way to investigate this would be through a variant of the DCI framework [2], referred to as *block-level DCI* [1,3], where the output of each module can be treated as a block-level representation. Such an approach could help empirically validate AIN’s contribution to disentangled representation learning more concretely.
> >
> > **Regarding W4**:
> >
> > After reviewing the rebuttal, I now understand that the phrase “training more than 5000 models” refers to multiple training runs using the same backbone architecture under different experimental settings, rather than using 5000 distinct backbone models as initially interpreted. As suggested in the rebuttal, explicitly using the term “training runs” would help avoid potential misinterpretations by readers.
> >
> > The rebuttal clarified key concerns and strengthened my confidence in the paper’s contributions. I have accordingly raised my score.
> >
> > [1] Neural Systematic Binder, ICLR 2023
> >
> > [2] A Framework for the Quantitative Evaluation of Disentangled Representations, 2018
> >
> > [3] Attention-based Iterative Decomposition for Tensor Product Representation, ICLR 2024

---

> > > ### Author Response · Authors · 2025-08-04
> > >
> > > Thanks for acknowledging our rebuttal and raising your score! We are pleased that our arguments clarified your concerns and strengthened your confidence in the paper's contribution.
> > > To reply to your additional comments:
> > >
> > > - **W3**: Thanks for the pointers to block-level DCI, it is indeed a very interesting measure to empirically validate the disentanglement in AIN's representations! We will look into it to strengthen our analysis of AINs even further.
> > >
> > > - **W4**: You are right, we now understand why our statement could be confusing in the first place. We will adjust the final manuscript accordingly.

---

### Note · Authors · 2025-08-12

We sincerely thank all reviewers for their feedback, which has helped us strengthen the clarity and rigor of our paper, as well as for their engagement throughout the process.

We are grateful for the wide acknowledgment of the strength of our empirical setup [QWmU, qPCT, bTmd], the validity and potential of our evaluation framework and AIN blueprint [bTmd, tLy1, QWmU], and the meaningful contributions of our broader investigation into compositional generalization [bTmd, QWmU, qPCT].

We also note that all the concerns initially raised by the reviewers were fruitfully addressed during the rebuttal phase. In particular, we:

- Discussed the relation with relevant previous works, in particular Okawa et al. (highlighted by reviewer bTmd) and attribute factorization in object-centric representation learning (highlighted by tLy1).

- Clarified the role of synthetic, labeled datasets in our experiments, whose value was eventually acknowledged by QWmU and tLy1.

- Solved a minor inconsistency in our definition of the compositional similarity index, spotted by reviewer qPCT.

These, along with other minor improvements discussed with the reviewers, will be incorporated into the final version of the manuscript.

---

### Decision · Program_Chairs · 2025-09-17

**Decision:**

Accept (poster)

**Comment:**

This paper introduces a scalable framework for evaluating compositional generalization, conducts an extensive empirical study involving thousands of training runs, and proposes Attribute Invariant Networks (AINs) as an efficient architecture that advances the Pareto frontier in this space. Reviewers agreed on the strengths of the work, highlighting the rigor and clarity of the evaluation framework, the impressive scale of the experiments, and the practical value of AINs in balancing accuracy and efficiency. Some concerns were raised, including reliance on synthetic datasets, the assumption of fully known generative factors, and the need for clearer connections between the empirical analysis and the model design. However, the rebuttal successfully clarified these points, addressed overlaps with prior work, and provided additional experiments under noisy conditions, which strengthened confidence in the contributions. Overall, the consensus is that the paper makes a solid and timely contribution to the study of compositional generalization, and I recommend acceptance.